# Split representation in celestial holography

Chi-Ming Chang[a,b,c], Reiko Liu[a], Wen-Jie Ma[c,a]

[a] *Yau Mathematical Sciences Center (YMSC), Tsinghua University, Beijing, 100084, China*

[b] *School of Natural Sciences, Institute for Advanced Study, Princeton, NJ 08540, USA*

[c] *Beijing Institute of Mathematical Sciences and Applications (BIMSA), Beijing, 101408, China*

`cmchang@tsinghua.edu.cn`, `reiko@tsinghua.edu.cn`, `wenjia.ma@bimsa.cn`

## Abstract

We develop a split representation for celestial amplitudes in celestial holography, by cutting internal lines of Feynman diagrams in Minkowski space. More explicitly, the bulk-to-bulk propagators associated with the internal lines are expressed as a product of two boundary-to-bulk propagators with a coinciding boundary point integrated over the celestial sphere. Applying this split representation, we compute the conformal partial wave and conformal block expansions of celestial four-point functions of massless scalars and photons on the Euclidean celestial sphere. In the $t$-channel massless scalar amplitude, we observe novel intermediate exchanges of staggered modules in the conformal block expansion.

# 1  Introduction

Celestial holography is a duality between a $d+2$-dimensional quantum field theory or quantum gravity in an asymptotic Minkowski space and a hypothetical $d$-dimensional celestial conformal field theory (CCFT) on the celestial sphere at null infinity [1–3]. More concretely, scattering amplitudes in the "bulk" theory when expanding in the conformal primary basis, that manifests the $d$-dimensional conformal symmetry ($d+2$-dimensional Lorentz symmetry), are conjectured to correspond to correlation functions in the "boundary" theory [1].

Besides conformal symmetry, perhaps the most important property that a CCFT should obey is locality. On the level of correlation functions, locality means that singularities arise exclusively when operators coincide and are captured by the operator product expansions (OPEs). Indeed, the analysis of the celestial OPEs of gluons and gravitons shows that the coinciding point singularities on the celestial sphere correspond to the colinear singularities of the scattering amplitudes [4]. However, it is unknown whether all the singularities of the celestial amplitudes are of this type, and it is well-known that generally singularities can occur at configurations where no pair of incoming or outgoing particles are colinear [5].

To examine the singularity structure of celestial amplitudes, it is convenient to fully utilize the conformal symmetry by expanding them in terms of conformal partial waves or conformal blocks. The conformal partial wave and conformal block expansions of the celestial four-point massless scalar amplitudes were studied in [6–9] using the completeness relation of the conformal partial waves. However, such a method is difficult to generalize to higher points due to the complicated integrals over the space of conformal cross ratios. To address this problem, we draw lessons from the anti-de Sitter (AdS) holography, where a powerful technique of computing AdS Witten diagrams and their conformal block expansions is the split representation of the AdS bulk-to-bulk propagators [10, 11]. In this paper, we develop a novel split representation for computing celestial amplitudes.

In perturbation theory, celestial amplitudes can be computed in a way similar to the computation of Witten diagrams in AdS using a set of Feynman rules conveniently formulated in position space (see a review in Section 3.1). In a Feynman diagram computing a celestial amplitude, an external line attaching a "boundary point" from the celestial sphere at the null infinity to a "bulk point" in Minkowski space is a bulk-to-boundary propagator that is identical to the conformal primary wave function. An internal line connecting two bulk points is a bulk-to-bulk propagator, which is the Fourier transform of the usual Feynman propagator from momentum space to position space. The internal and external lines are joined at bulk vertices, which are given by the interaction terms in the bulk action, and integrated over the entire Minkowski space.

To derive a split representation for the bulk-to-bulk propagator, we divide the Fourier

integral in momentum space into integrals over the regions inside and outside the light cone of an observer at the origin ($p = 0$) as shown in Figure 1. The regions inside the past or future light cones are foliated by $d + 1$-dimensional Euclidean anti-de Sitter (EAdS) slices, and the region outside the light cones is foliated by $d + 1$-dimensional de Sitter (dS) slices. We then separately apply the EAdS and dS split representations (see Section 2) for the part of the bulk-to-bulk propagator on the EAdS and dS slices. After putting things together, the final result is a split representation for the bulk-to-bulk propagator into a product of two bulk-to-boundary propagators with a common boundary point integrated over the celestial sphere (see (4.9) in Section 4).

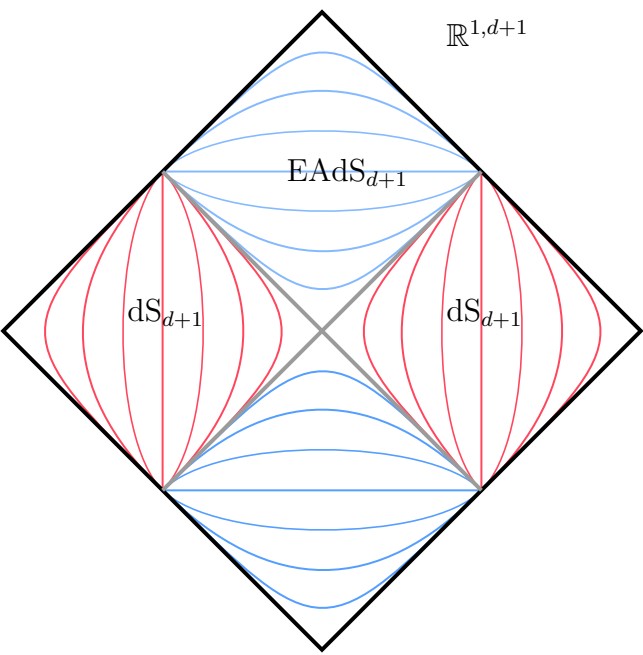

Figure 1: Hyperbolic slicing of momentum space $\mathbb{R}^{1,d+1}$. Each blue curve is a EAdS$_{d+1}$, and each left-right pair of red curves is a dS$_{d+1}$.

Applying our split representation to the internal propagators of a celestial amplitude reduces it into an integral of products of lower point amplitudes. Repeating the above step, we could reduce a higher point celestial amplitude to three-point amplitudes and obtain a conformal partial wave or conformal block expansion. We demonstrate this in various examples of four-point amplitudes (see Section 5), including the $s$-channel massless scalar amplitude, the $s$-channel Compton amplitude, the $t$-channel massless scalar amplitude, and the contact scalar amplitude. It is important to note that our split representation naturally produces expansions in terms of conformal blocks on the Euclidean celestial sphere. In particular, for $d = 2$, we achieve an expansion of the $t$-channel massless scalar amplitude in terms of the SL$(2, \mathbb{C})$ blocks. This is in contrast to the expansion of the same amplitude in

terms of $\mathrm{SL}(2,\mathbb{R}) \times \mathrm{SL}(2,\mathbb{R})$ blocks [7], which are the conformal blocks on the Lorentzian celestial torus. In both cases, there are extra contributions to the expansion beyond the conformal blocks of unitary Verma modules. Notably, In the $\mathrm{SL}(2,\mathbb{C})$ block expansion, we find chiral staggered modules, contrasting with the light-ray states identified in the $\mathrm{SL}(2,\mathbb{R}) \times \mathrm{SL}(2,\mathbb{R})$ block expansion [7].

The rest of this paper is organized as follows. Section 2 and Appendix D derive the split representations in EAdS and dS spaces. Section 3 reviews the computation of celestial amplitudes in perturbation theory using Feynman diagrams and Feynman rules involving the bulk-to-boundary and bulk-to-bulk propagators. Section 4 presents our main result, the split representation for the scalar bulk-to-bulk propagator. Section 5 applies the split representation to various celestial four-point amplitudes. Section 6 ends with some concluding remarks.

# 2 Split representations in EAdS and dS spaces

For the EAdS spacetime, the (scalar) bulk-to-bulk propagator $G_\Delta(X, X')$ is related to the product of two bulk-to-boundary propagators $K_\Delta(X, P)K_{\widetilde{\Delta}}(X', P)$ with the common point $P$ integrated over the boundary [10, 12–14]. This is called the split representation of the scalar propagator, and is a special case of the split representation of the Dirac delta function, *i.e.* the resolution of identity on the space of normalizable functions on $\mathrm{EAdS}_{d+1}$,

$$\delta(X, X') = \frac{1}{2\pi i} \int_{\frac{d}{2}}^{\frac{d}{2}+i\infty} \frac{d\Delta}{\mu(\Delta)} \int dP \, K_\Delta(X, P) K_{\widetilde{\Delta}}(X', P), \qquad (2.1)$$

where $X, X'$ are bulk points, $P$ is a boundary point parametrized by $P = \left(\frac{1+x^2}{2}, x, \frac{1-x^2}{2}\right), x \in \mathbb{R}^d$, $\widetilde{\Delta}$ is the shadow weight $\widetilde{\Delta} = d - \Delta$, $K_\Delta(X, P)$ is the bulk-to-boundary propagator,

$$K_\Delta(X, P) = (-2X \cdot P)^{-\Delta}, \qquad (2.2)$$

and $\mu^{-1}(\Delta)$ is called the Plancherel measure,

$$\mu(\Delta) = \frac{\pi^d \Gamma(\Delta - \frac{d}{2}) \Gamma(\widetilde{\Delta} - \frac{d}{2})}{\Gamma(\Delta) \Gamma(\widetilde{\Delta})}. \qquad (2.3)$$

The idea behinds the split representation is as follows.

By the spectrum theorem, for a manifold $M$ and a self-adjoint differential operator $D$ on $M$, given an orthnormal basis $\{e_i(x) : i \in I\}$ of $D$: $De_i = \lambda_i e_i$, the resolution of identity

$$\delta(x, x') = \sum_{i \in I} e_i^*(x') e_i(x), \qquad (2.4)$$

is equivalent to the completeness of the basis. The propagator $G$ is the operator inverse of $D$: $DG = \delta$, and

$$G(x, x') = \sum_{i \in I} \frac{1}{\lambda_i} e_i^*(x') e_i(x). \tag{2.5}$$

If further $M$ admits a transitive symmetry $G$, i.e. $M = G/H$, then different eigenspaces of $D$ carry inequivalent representations of $G$ labeled by $i$ with the same $\lambda_i$.

For EAdS$_{d+1}$, choosing the differential operator as the Laplacian, the eigenspaces are unitary principal series representations $\mathcal{E}_\Delta$ of the isometry group $SO(d+1, 1)$. For any boundary point $P$, the bulk-to-boundary propagator $K_\Delta(X, P)$ is an eigenfunction of the Laplacian, and the integration over $P$ resolves the degeneracy of the eigenspaces.

The appearance of the bulk-to-boundary propagator $K_\Delta(X, P)$ in the split representation (2.1) is quite natural from the perspective of harmonic analysis, see Appendix D. For a normalizable function $f(X)$ on EAdS, the Fourier transform $\mathcal{F}_\Delta$ is

$$\mathcal{F}_\Delta[f(X)](P) = \int dX \, K_\Delta(X, P) f(X), \tag{2.6}$$

and for a normalizable function $f(P)$ on the boundary, the Poisson transform $\mathcal{F}_\Delta^\dagger$ is

$$\mathcal{F}_\Delta^\dagger[f(P)](X) = \int dP \, K_{\tilde{\Delta}}(X, P) f(P). \tag{2.7}$$

Then the composition maps $\mathcal{P}_\Delta = \mathcal{F}_\Delta^\dagger \mathcal{F}_\Delta$ are projectors that decompose $f(X)$ according to the unitary principal series representations $\mathcal{E}_\Delta$,

$$\mathcal{P}_\Delta[f(X')](X) = \int dX' \, \phi_\Delta(X, X') f(X'), \tag{2.8}$$

where the integral kernel $\phi_\Delta(X, X')$ is called the spherical function on EAdS,

$$\phi_\Delta(X, X') = \int dP \, K_\Delta(X, P) K_{\tilde{\Delta}}(X, P). \tag{2.9}$$

The inversion formula ensures the decomposition is complete,

$$f(X) = \frac{1}{2\pi i} \int_{\frac{d}{2}}^{\frac{d}{2}+i\infty} \frac{d\Delta}{\mu(\Delta)} \, \mathcal{P}_\Delta[f(X)], \tag{2.10}$$

and is equivalent to the resolution of identity (2.1). The physical interpretations are as follows: the spherical function is the Wightman function of the free scalar; the Poisson transform is the Euclidean counterpart of the HKLL reconstruction of the free scalar [15, 16]; the Fourier transform is the Dobrev intertwining relation [17], which motivates the split representation on EAdS [12].

The preceding discussions inspire the question: is there a similar relation to (2.1) on the dS spacetime? The answer is affirmative with two new features: the quantity $X \cdot P$ is indefinite for $X \in \mathrm{dS}_{d+1}$ and $K_\Delta(X, P)$ in (2.2) is ill-defined; the spectrum of the Laplacian contains both continuous and discrete parts. The split representation in dS is

$$
\begin{aligned}
\delta(X, X') = {} & \frac{1}{2\pi i} \sum_{\epsilon=0,1} \int_{\frac{d}{2}}^{\frac{d}{2}+i\infty} \frac{d\Delta}{2\mu(\Delta)} \int dP \, K_{\Delta,\epsilon}(X, P) K_{\widetilde{\Delta},\epsilon}(X', P) \\
& + \alpha_d \sum_{(\Delta,\epsilon)\in D_{\mathrm{dS}}} \mathrm{Res} \, \frac{1}{2\mu(\Delta)} \int dP \, K_{\Delta,\epsilon}(X, P) K_{\widetilde{\Delta},\epsilon}(X', P).
\end{aligned}
\tag{2.11}
$$

where $\alpha_d = 1$ for odd $d$ and $\alpha_d = \frac{1}{2}$ for even $d$, the analog of the bulk-to-boundary propagator is

$$
K_{\Delta,\epsilon}(X, P) = |-2X \cdot P|^{-\Delta,\epsilon} \equiv |-2X \cdot P|^{-\Delta} \mathrm{sgn}^\epsilon(-2X \cdot P),
\tag{2.12}
$$

and the discrete spectrum is

$$
D_{\mathrm{dS}} = \{(\Delta \in \mathbb{Z}, \epsilon): \Delta > \frac{d}{2} \text{ and } \epsilon = 1 + d + \Delta \bmod \mathbb{Z}_2\}.
\tag{2.13}
$$

The derivation requires the harmonic analysis on the symmetric spaces, and we leave it to the appendices. In Appendix C we review the spherical function method in the harmonic analysis. In Appendix D we derive the split representations on EAdS/dS from the spherical function method. The result (2.11) is not new, but appears in different forms in the old literature [18–22].

# 3 A brief review of celestial amplitudes

## 3.1 Feynman rules for celestial amplitudes

In a $d$-dimensional celestial CFT, celestial amplitudes are scattering amplitudes expanded on the basis of conformal primary wavefunctions [1, 23] instead of the usual plane-waves. For instance, given an amputated amplitude $\mathcal{M}(X_j)$ of massless scalars in the position space, we obtain a celestial scalar amplitude $\mathcal{A}(x_i)$ by changing the basis as the integral

$$
\mathcal{A}(x_i) = \left(\prod_{j=1}^{k+n} \int d^{d+2}X_j\right) \left(\prod_{j=1}^{k} \phi^+_{\Delta_j}(x_j; X_j)\right) \left(\prod_{j=k+1}^{k+n} \phi^-_{\Delta_j}(x_j; X_j)\right) \mathcal{M}(X_i),
\tag{3.1}
$$

where $\phi^\pm_\Delta(x; X)$ are the incoming $(+)$ and outgoing $(-)$ massless scalar conformal primary wave functions with the conformal dimension $\Delta$. Since the conformal primary basis connects a bulk point $X$ in the $(d+2)$-dimensional Minkowski space and a boundary point $x$ on the

$d$-dimensional celestial sphere, we will also call it a **bulk-to-boundary propagator**. We will review the explicit forms of the bulk-to-boundary propagators in the next subsection.

In perturbation theory, the amplitude $\mathcal{M}(X_j)$ can be computed by Feynman rules. The propagators and interaction vertices are simply given by Fourier transforms of those in the momentum space. For instance, a scalar propagator in the position space, connecting two bulk points $X_1$ and $X_2$, is given by[1]

$$\mathcal{K}_m(X_1, X_2) = \int \frac{d^{d+2}p}{(2\pi)^{d+2}} \frac{i}{p^2 + m^2 - i\epsilon} e^{-ip \cdot (X_1 - X_2)} , \qquad (3.2)$$

which satisfies the Green's function equation

$$(\partial_X^2 + m^2)\mathcal{K}_m(X, Y) = i\delta^{(d+2)}(X - Y) . \qquad (3.3)$$

We will call it a **bulk-to-bulk propagator**.

We will focus on tree-level amplitudes with scalars and photons. For example, the $2 \to 2$ contact celestial scalar amplitude is given by

$$\mathcal{A}^{\Delta_i}_{1_0^0 + 2_0^0 \to 3_0^0 + 4_0^0}(x_i) = \int d^{d+2}X \, \phi^+_{\Delta_1}(x_1, X)\phi^+_{\Delta_2}(x_2, X)\phi^-_{\Delta_3}(x_3, X)\phi^-_{\Delta_4}(x_4, X) . \qquad (3.4)$$

We have introduced the notation

$$i_m^s \qquad \text{for } i\text{-th, mass } m, \text{ helicity/spin } s \text{ particle.} \qquad (3.5)$$

Another example is the $s$-channel tree-level celestial amplitude with two incoming and two outgoing massless scalars and a massive scalar exchange. Using the bulk-to-bulk and bulk-to-boundary propagators, this amplitude can be computed by

$$_s\mathcal{A}^{\Delta_i}_{1_0^0 + 2_0^0 \to 3_0^0 + 4_0^0}(x_i) = \int d^{d+2}X_1 d^{d+2}X_2 \left(\prod_{i=1}^{2} \phi^+_{\Delta_i}(x_i, X_1)\right)\mathcal{K}_m(X_1, X_2)\left(\prod_{i=3}^{4} \phi^-_{\Delta_i}(x_i, X_2)\right) . \quad (3.6)$$

Similarly, the $t$-channel tree-level $2 \to 2$ celestial scalar amplitude is

$$_t\mathcal{A}^{\Delta_i}_{1_0^0 + 2_0^0 \to 3_0^0 + 4_0^0}(x_i) = \int d^{d+2}X_1 d^{d+2}X_2$$
$$\times \phi^+_{\Delta_1}(x_1, X_1)\phi^+_{\Delta_2}(x_2, X_2)\mathcal{K}_m(X_1, X_2)\phi^-_{\Delta_3}(x_3, X_1)\phi^-_{\Delta_4}(x_4, X_2) . \qquad (3.7)$$

Finally, we consider the $s$-channel Compton amplitude. In the plane-wave basis, the amplitude in four-dimensional Minkowski space is

$$\mathcal{M}_{\mu\nu} = (2q_{1\mu} + q_{2\mu})\frac{1}{(q_1 + q_2)^2 - i\epsilon}(2q_{3\nu} + q_{4\nu}) . \qquad (3.8)$$

---

[1]We use the most positive metric in this paper, *i.e.* $g_{\mu\nu} = \text{diag}(-1, +1, +1, \cdots, +1)$.

The corresponding celestial amplitude is

$$
{}_s\mathcal{A}^{\Delta_i}_{1^0_0+2^-_0\to3^0_0+4^-_0}(x_i)
$$
$$
= \frac{1}{(2\pi)^8}\int d^4X_1 d^4X_2 \left[2\partial^\mu_{X_1}\phi^+_{\Delta_1}(x_1,X_1)\phi^+_{\Delta_2;\mu-}(x_2,X_1) + \phi^+_{\Delta_1}(x_1,X_1)\partial^\mu_{X_1}\phi^+_{\Delta_2;\mu-}(x_2,X_1)\right] \quad (3.9)
$$
$$
\times \mathcal{K}_{m=0}(X_1,X_2)\left[2\partial^\nu_{X_2}\phi^-_{\Delta_3}(x_3,X_2)\phi^-_{\Delta_4;\nu-}(x_4,X_2) + \phi^-_{\Delta_3}(x_3,X_2)\partial^\nu_{X_2}\phi^-_{\Delta_4;\nu-}(x_4,X_2)\right],
$$

where $\phi^\pm_{\Delta;\mu a}$ is the bulk-to-boundary photon propagator.

## 3.2 Bulk-to-boundary propagators

The bulk-to-boundary propagators (conformal primary wave functions) $\phi^\pm_{\Delta,m}(x;X)$ and $\phi^\pm_\Delta(x;X)$ for massive and massless scalars with mass $m$ are given by

$$
\phi^\pm_{\Delta,m}(x;X) = \int \frac{d^{d+1}\widehat{p}'}{\widehat{p}'^0}\frac{1}{(-\widehat{q}\cdot\widehat{p}')^\Delta}e^{\pm im\widehat{p}'\cdot X},
$$
$$
\phi^\pm_\Delta(x;X) = \int_0^{+\infty} d\omega\, \omega^{\Delta-1}e^{\pm i\omega\widehat{q}\cdot X - \epsilon\omega} = N^\pm_\Delta\frac{1}{(-\widehat{q}\cdot X\mp i\epsilon)^\Delta}, \quad (3.10)
$$

where $N^\pm_\Delta$ is a constant factor given by

$$
N^\pm_\Delta = (\mp i)^\Delta\Gamma[\Delta]. \quad (3.11)
$$

$q^\mu$ and $\widehat{q}$ are massless and massive on-shell momenta with the parameterizations[2]

$$
q^\mu(\omega,x) = \omega\widehat{q}^\mu(x) = \omega(1+x^2, 2\vec{x}, 1-x^2) = 2\omega P^\mu,
$$
$$
p^\mu(m,y,x) = m\widehat{p}^\mu(y,x) = \frac{m}{2y}(1+y^2+x^2, 2\vec{x}, 1-y^2-x^2). \quad (3.12)
$$

There is another set of bulk-to-boundary propagators given by the shadow transform of the wave functions in (3.10). As shown in [23], the shadow transform of $\phi^\pm_{\Delta,m}(x;X)$ is simply $\phi^\pm_{d-\Delta,m}(x;X)$. The shadow transformation of $\phi^\pm_\Delta(x;X)$ is

$$
\widetilde{\phi}^\pm_\Delta(x;X) = \frac{\Gamma[d-\Delta]}{\pi^{\frac{d}{2}}\Gamma[\frac{d}{2}-\Delta]}\int d^dx'\frac{\phi^\pm_{d-\Delta}(x';X)}{|x-x'|^{2\Delta}} = \frac{N^\pm_{d-\Delta}}{N^\pm_\Delta}(-X^2)^{\Delta-\frac{d}{2}}\phi^\pm_\Delta(x;X). \quad (3.13)
$$

As we will see later in Section 4, in the split representation, the bulk-to-bulk propagator would also factorize into bulk-to-boundary propagators with imaginary mass, $im$. Hence, we

---

[2]We note that the coordinates $P$ defined in Section 2 is related to $\widehat{q}$ by $\widehat{q} = 2P$.

introduce the bulk-to-boundary propagators for tachyonic scalar as[3]

$$\phi_{\Delta,im,\epsilon}(x;X) = \int \frac{d^{d+1}\widehat{k}'}{|\widehat{k}'+|} \frac{1}{|\widehat{q} \cdot \widehat{k}'|^\Delta} \mathrm{sgn}^\epsilon(\widehat{q} \cdot \widehat{k}') e^{-im\widehat{k}' \cdot X} \tag{3.14}$$

with $k'^2 = m^2$ and $\epsilon = 0, 1$. Note that there is no distinction between incoming and outgoing for tachyons. The bulk-to-boundary propagators $\phi_{\Delta,im,\epsilon}(x;X)$ are conformally covariant and satisfy the tachyonic Klein-Gordon equation $(\partial_X^2 + m^2)\phi_{\Delta,im,\epsilon}(x;X) = 0$.

The bulk-to-boundary propagator (conformal primary wavefunctions) for massless spin-one particles is [23, 24]

$$\begin{aligned}
\phi_{\mu a}^{\Delta,\pm}(\widehat{q};X) &= \frac{\partial_a \widehat{q}_\mu}{(-\widehat{q} \cdot X \mp i\epsilon)^\Delta} + \frac{\widehat{q}_\mu \partial_a \widehat{q} \cdot X}{(-\widehat{q} \cdot X \mp i\epsilon)^{\Delta+1}} \\
&\equiv V_{\mu a}^{\Delta,\pm}(x,X) + W_{\mu a}^{\Delta,\pm}(x,X) \, .
\end{aligned} \tag{3.15}$$

The shadow transform of $\phi_{\mu a}^{\Delta,\pm}$ is related to itself and we have

$$\widetilde{\phi}_{\mu a}^{\Delta,\pm} = (-X^2)^{\Delta-1}\phi_{\mu a}^{\Delta,\pm} = \frac{(-X^2)^{\Delta-1}\partial_a \widehat{q}^\mu}{(-\widehat{q} \cdot X \mp i\epsilon)^\Delta} + \frac{(-X^2)^{\Delta-1}\widehat{q}^\mu \partial_a \widehat{q} \cdot X}{(-\widehat{q} \cdot X \mp i\epsilon)^{\Delta+1}} \, . \tag{3.16}$$

Up to a pure gauge, $\phi_{\mu a}^{\Delta,\pm}(\widehat{q};X)$ can be written as a Mellin transform of a plane-wave as

$$\phi_{\mu a}^{\Delta,\pm}(\widehat{q};X) = \frac{\Delta-1}{\Delta N_\Delta^\pm} \partial_a \widehat{q}^\mu \int_0^\infty d\omega \, \omega^{\Delta-1} e^{\pm i\omega \widehat{q} \cdot X - \epsilon\omega} + \partial_\mu \alpha_a^{\Delta,\pm} \, . \tag{3.17}$$

# 4 Split representation in CCFT

We are now ready to derive the split representation for the bulk-to-bulk propagator $\mathcal{K}_m(X_1, X_2)$ in (3.2) in celestial holography, by combining the split representations in EAdS and dS developed in Section 2 (with more details in Appendix D). As discussed in the introduction, we divide the Fourier integral of the bulk-to-bulk propagator (3.2) into integrals over the regions $p^2 > 0$ and $p^2 < 0$ as

$$\mathcal{K}_m(X_1, X_2) = \mathcal{K}_m^+(X_1, X_2) + \mathcal{K}_m^-(X_1, X_2) \, , \tag{4.1}$$

where $\mathcal{K}_m^+(X_1, X_2)$ and $\mathcal{K}_m^-(X_1, X_2)$ are given by

$$\mathcal{K}_m^+(X_1, X_2) = \int_{p^2>0} \frac{d^{d+2}p}{(2\pi)^{d+2}} \frac{i}{p^2 + m^2} e^{-ip \cdot (X_1 - X_2)} \, , \tag{4.2}$$

$$\mathcal{K}_m^-(X_1, X_2) = \int_{p^2<0} \frac{d^{d+2}p}{(2\pi)^{d+2}} \frac{i}{p^2 + m^2} e^{-ip \cdot (X_1 - X_2)} \, . \tag{4.3}$$

---

[3]One may also define the tachyonic basis using $e^{+im\widehat{k}' \cdot X}$ instead of $e^{-im\widehat{k}' \cdot X}$ in (3.14). It is easy to check that this definition gives $(-1)^\epsilon \phi_{\Delta,im,\epsilon}(x;X)$.

Let us first compute $\mathcal{K}_m^-$. We define $p^\mu \equiv M\widehat{p}^\mu$ with $\widehat{p}^2 = -1$ and $\widehat{p}^0 > 0$. This leads to

$$\mathcal{K}_m^-(X_1, X_2) = \int_{-\infty}^{+\infty} \frac{dM}{(2\pi)^{d+2}} \frac{i|M|^{d+1}}{-M^2 + m^2} \int \frac{d^{d+1}\widehat{p}_1}{\widehat{p}_1^0} \int \frac{d^{d+1}\widehat{p}_2}{\widehat{p}_2^0} \widehat{p}_2^0 \delta^{(d+1)}(\widehat{p}_1 - \widehat{p}_2) e^{-iM\widehat{p}\cdot(X_1 - X_2)} ,$$

(4.4)

where we inserted a delta function. Using the resolution of identity (2.1) in $\text{EAdS}_{d+1}$ we can rewrite $\mathcal{K}_m^-$ as

$$\mathcal{K}_m^-(X_1, X_2) = \frac{1}{2\pi i} \int_0^{+\infty} \frac{dM}{(2\pi)^{d+2}} \frac{iM^{d+1}}{-M^2 + m^2} \int_{\frac{d}{2}}^{\frac{d}{2}+i\infty} \frac{d\Delta}{\mu(\Delta)}$$

$$\times \int d^d x \left[ \phi_{\Delta,M}^-(x; X_1)\phi_{d-\Delta,M}^+(x; X_2) + \phi_{\Delta,M}^+(x; X_1)\phi_{d-\Delta,M}^-(x; X_2) \right] .$$

(4.5)

Next, let us consider $\mathcal{K}_m^+$. We can define $k = R\widehat{k}$ with $R > 0$, $\widehat{k}^2 = 1$. The invariant measure becomes

$$\int_{p^2 > 0} d^{d+2}p = \int_0^{+\infty} dR\, R^{d+1} \int \frac{d^{d+1}\widehat{k}}{|\widehat{k}^+|} ,$$

(4.6)

where we introduced the notation $d^{d+1}\widehat{k} \equiv d\widehat{k}^+ d\widehat{k}_1 \cdots d\widehat{k}_d$ and $\widehat{k}^+ = \widehat{k}^0 + \widehat{k}^{d+1}$. As a result, $\mathcal{K}_m^+(X_1, X_2)$ can be written as

$$\mathcal{K}_m^+ = \int_0^{+\infty} \frac{dR}{(2\pi)^{d+2}} \frac{iR^{d+1}}{R^2 + m^2} \int \frac{d^{d+1}\widehat{k}_1}{|\widehat{k}_1^+|} \int \frac{d^{d+1}\widehat{k}_2}{|\widehat{k}_2^+|} |\widehat{k}_2^+| \delta(\widehat{k}_1^+ - \widehat{k}_2^+) \delta^{(d)}(\widehat{k}_1 - \widehat{k}_2) e^{-iR\widehat{k}_1\cdot(X_1 - X_2)} .$$

(4.7)

Using the resolution identity (2.11) in $\text{dS}_{d+1}$ we can re-write $\mathcal{K}_m^+$ as

$$\mathcal{K}_m^+ = \frac{1}{2} \int_0^\infty \frac{dR}{(2\pi)^{d+2}} \frac{iR^{d+1}}{R^2 + m^2} \left( \frac{1}{2\pi i} \sum_{\epsilon=0,1} \int_{\frac{d}{2}}^{\frac{d}{2}+i\infty} \frac{(-1)^\epsilon d\Delta}{\mu(\Delta)} \int d^d x\, \phi_{\Delta,iR,\epsilon}(x; X_1)\phi_{d-\Delta,iR,\epsilon}(x; X_2) \right.$$

$$\left. + \alpha_d \sum_{(\Delta,\epsilon)\in D_{\text{ds}}} \text{Res} \frac{(-1)^\epsilon}{\mu(\Delta)} \int d^d x\, \phi_{\Delta,iR,\epsilon}(x; X_1)\phi_{d-\Delta,iR,\epsilon}(x; X_2) \right) ,$$

(4.8)

where $\phi_{\Delta,iR,\epsilon}$ are the tachyon wave functions introduced in (3.14).

Putting everything together, we thus conclude that the bulk-to-bulk propagator $\mathcal{K}_m(X_1, X_2)$

has the following split representation[4]

$$
\begin{aligned}
\mathcal{K}_m&(X_1, X_2) \\
&= \frac{1}{2\pi i} \int_0^{+\infty} \frac{dM}{(2\pi)^{d+2}} \frac{iM^{d+1}}{-M^2+m^2} \int_{\frac{d}{2}}^{\frac{d}{2}+i\infty} \frac{d\Delta}{\mu(\Delta)} \int d^d x \left( \phi_{\Delta,M}^- \phi_{d-\Delta,M}^+ + \phi_{\Delta,M}^+ \phi_{d-\Delta,M}^- \right) \\
&\quad + \frac{1}{2} \int_0^\infty \frac{dM}{(2\pi)^{d+2}} \frac{iM^{d+1}}{M^2+m^2} \left( \frac{1}{2\pi i} \sum_{\epsilon=0,1} \int_{\frac{d}{2}}^{\frac{d}{2}+i\infty} \frac{(-1)^\epsilon d\Delta}{\mu(\Delta)} \int d^d x \phi_{\Delta,iM,\epsilon} \phi_{d-\Delta,iM,\epsilon} \right. \\
&\quad \left. + \alpha_d \sum_{(\Delta,\epsilon)\in D_{\mathrm{dS}}} \mathrm{Res} \frac{(-1)^\epsilon}{\mu(\Delta)} \int d^d x \phi_{\Delta,iM,\epsilon} \phi_{d-\Delta,iM,\epsilon} \right) .
\end{aligned} \tag{4.9}
$$

From the above formula, we see that the bulk-to-bulk propagator in Minkowski space can be thought of as a superposition of a product of two massive (tachyonic) bulk-to-boundary propagators over a range of real (imaginary) mass $(i)M$. Specifically, the first line in (4.9) is the contribution from propagating a massive scalar with mass $M$, while the last two lines are the contribution from propagating a tachyonic scalar with imaginary mass $iM$.

# 5   Examples

Let us apply the split representation (4.9) to the various celestial amplitudes introduced in Section 3.1, and find the conformal partial wave expansions for these celestial amplitudes.

## 5.1   $s$-channel massless scalar four-point amplitude

Let us consider the amplitude (3.6), which is a $s$-channel tree-level $2 \to 2$ celestial scalar amplitude with a massive scalar exchange. We note that in this case, the internal momentum $p = q_1 + q_2$ satisfies $p^2 \leqslant 0$. Thus only the first line in (4.9) contributes. Using the split representation, we find that

$$
{}_s\mathcal{A}_{1_0+2_0\to3_0+4_0}^{\Delta_i} = \frac{1}{2\pi i} \int_0^\infty \frac{dM}{(2\pi)^{d+2}} \frac{iM^{d+1}}{-M^2+m^2} \int_{\frac{d}{2}}^{\frac{d}{2}+i\infty} \frac{d\Delta}{\mu(\Delta)} \int d^d x_0 \, \mathcal{A}_{1_0+2_0\to0_M^0}^{\Delta_1,\Delta_2,\Delta} \mathcal{A}_{0_M^0\to3_0+4_0}^{d-\Delta,\Delta_3,\Delta_4} ,
\tag{5.1}
$$

where $\mathcal{A}_{1_0+2_0\to0_M^0}^{\Delta_1,\Delta_2,\Delta}$ and $\mathcal{A}_{0_M^0\to3_0+4_0}^{d-\Delta,\Delta_3,\Delta_4}$ are three-point celestial amplitudes, given by

$$
\mathcal{A}_{1_0+2_0\to0_M^0}^{\Delta_1,\Delta_2,\Delta} = \int d^{d+2}X_1 \, \phi_{\Delta_1}^+(x_1, X_1)\phi_{\Delta_2}^+(x_2, X_1)\phi_{\Delta,M}^-(x_0, X_1) ,
\tag{5.2}
$$

---

[4]A similar split representation of the bulk-to-bulk propagator was also studied previously in [25]. However, the contributions from the discrete part in (2.11) were missing.

$$\mathcal{A}^{d-\Delta,\Delta_3,\Delta_4}_{0^0_M \to 3^0_0+4^0_0} = \int d^{d+2}X_2\, \phi^+_{d-\Delta,M}(x_0,X_2)\phi^-_{\Delta_3}(x_3,X_2)\phi^-_{\Delta_4}(x_4,X_2)\,. \tag{5.3}$$

After performing the $X_i$ integrals, they take the standard form of scalar three-point correlation functions in CFTs, *i.e.*

$$\mathcal{A}^{\Delta_1,\Delta_2,\Delta}_{1^0_0+2^0_0\to 0^0_M} = \frac{C^{\Delta_1,\Delta_2,\Delta_3}_{1^0_0+2^0_0\to 3^0_M}}{|z_{12}|^{\Delta_1+\Delta_2-\Delta_3}|z_{13}|^{\Delta_1-\Delta_2+\Delta_3}|z_{23}|^{\Delta_2+\Delta_3-\Delta_1}}\,, \tag{5.4}$$

$$\mathcal{A}^{\Delta_3,\Delta_1,\Delta_2}_{3^0_M\to 1^0_0+2^0_0} = \frac{C^{\Delta_3,\Delta_1,\Delta_2}_{3^0_M\to 1^0_0+2^0_0}}{|z_{12}|^{\Delta_1+\Delta_2-\Delta_3}|z_{13}|^{\Delta_1-\Delta_2+\Delta_3}|z_{23}|^{\Delta_2+\Delta_3-\Delta_1}}\,. \tag{5.5}$$

Here, the three-point coefficients $C^{\Delta_1,\Delta_2,\Delta_3}_{1^0_0+2^0_0\to 3^0_M}$ and $C^{\Delta_3,\Delta_1,\Delta_2}_{3^0_M\to 1^0_0+2^0_0}$ are

$$C^{\Delta_1,\Delta_2,\Delta_3}_{1^0_0+2^0_0\to 3^0_M} = C^{\Delta_3,\Delta_1,\Delta_2}_{3^0_M\to 1^0_0+2^0_0} = \frac{M^{\Delta_1+\Delta_2-d-2}\Gamma[\frac{\Delta_{12}+\Delta_3}{2}]\Gamma[\frac{-\Delta_{12}+\Delta_3}{2}]}{2^{\Delta_1+\Delta_2}\Gamma[\Delta_3]}\,, \tag{5.6}$$

where we defined $\Delta_{ij}\equiv\Delta_i-\Delta_j$. Substituting (5.4) into (5.1) then leads to following the conformal partial wave expansion of $_s\mathcal{A}^{\Delta_i}_{1^0_0+2^0_0\to 3^0_0+4^0_0}(x_i)$

$$_s\mathcal{A}^{\Delta_i}_{1^0_0+2^0_0\to 3^0_0+4^0_0} = \frac{1}{2\pi i}\int_0^\infty \frac{dM}{(2\pi)^{d+2}}\frac{iM^{d+1}}{-M^2+m^2}\int_{\frac{d}{2}}^{\frac{d}{2}+i\infty}\frac{d\Delta}{\mu(\Delta)}C^{\Delta_1,\Delta_2,\Delta}_{1^0_0+2^0_0\to 0^0_M}C^{d-\Delta,\Delta_3,\Delta_4}_{0^0_M\to 3^0_0+4^0_0}\Psi^{\Delta_{12},\Delta_{34}}_{\Delta,J=0}\,, \tag{5.7}$$

which agrees with the results in [6] for $d=1$ and in [26] for $d=2$.

## 5.2  *s*-channel Compton amplitude

Let us apply the split representation to the *s*-channel celestial Compton amplitude (3.9).

For simplicity, we will focus on two-dimensional CCFT, *i.e.* we choose $d=2$. We stress here that the helicity $+$ or $-$ in this paper indicates the helicity in four-dimensional Minkowski space. For conformal primary basis, the helicity in two-dimensional CCFT coincides with the helicity [**spin?**] in four-dimensional Minkowski space. On the other hand, for shadow conformal primary basis, the helicity in two-dimensional CCFT gets flipped compared with the helicity in four-dimensional Minkowski space. This is because the shadow transformation flips the helicity.

Applying integration by parts on the integral in (3.9) and using the Lorentz gauge condition $\partial^\mu_X\phi^\pm_{\Delta;\mu a}(x,X)=0$, we find

$$_s\mathcal{A}^{\Delta_i}_{1^0_0+2^-_0\to 3^0_0+4^-_0} = \frac{4}{(2\pi)^8}\int d^4X_1 d^4X_2 \partial^\mu_{X_1}\phi^+_{\Delta_1}(x_1,X_1)\phi^+_{\Delta_2;\mu-}(x_2,X_1)$$
$$\times\, \mathcal{K}_{m=0}(X_1,X_2)\partial^\nu_{X_2}\phi^-_{\Delta_3}(x_3,X_2)\phi^-_{\Delta_4;\nu-}(x_4,X_2)\,. \tag{5.8}$$

With the help of split representation (4.9), $_s\mathcal{A}^{\Delta_i}_{1^0_0+2^-_0\to3^0_0+4^-_0}(x_i)$ can be written as

$$_s\mathcal{A}^{\Delta_i}_{1^0_0+2^-_0\to3^0_0+4^-_0}(x_i) = -\frac{4}{2\pi i}\int_0^\infty \frac{dM}{(2\pi)^{12}}\frac{iM^3}{-M^2+i\epsilon}$$
$$\times\int_1^{1+i\infty}d\Delta\frac{(\Delta-1)^2}{\pi^2}\int d^2x_0\mathcal{A}^{\Delta_1,\Delta_2,\Delta}_{1^0_0+2^-_0\to0^0_M}\mathcal{A}^{2-\Delta,\Delta_3,\Delta_4}_{0^0_M\to3^0_0+4^-_0}\,, \tag{5.9}$$

where the three-point celestial amplitudes $\mathcal{A}^{\Delta_1,\Delta_2,\Delta}_{1^0_0+2^-_0\to0^0_M}$ and $\mathcal{A}^{2-\Delta,\Delta_3,\Delta_4}_{0^0_M\to3^0_0+4^-_0}$ are

$$\mathcal{A}^{\Delta_1,\Delta_2,\Delta}_{1^0_0+2^-_0\to0^0_M} = \int d^4X_1\partial^\mu_{X_1}\phi^+_{\Delta_1}(x_1,X_1)\phi^+_{\Delta_2;\mu-}(x_2,X_1)\phi^-_{\Delta,M}(x_0,X_1)\,, \tag{5.10}$$

and

$$\mathcal{A}^{2-\Delta,\Delta_3,\Delta_4}_{0^0_M\to3^0_0+4^-_0} = \int d^4X_2\partial^\nu_{X_2}\phi^-_{\Delta_3}(x_3,X_2)\phi^-_{\Delta_4;\nu-}(x_4,X_2)\phi^+_{2-\Delta,M}(x_0,X_2)\,. \tag{5.11}$$

To compute $\mathcal{A}^{\Delta_1,\Delta_2,\Delta}_{1^0_0+2^-_0\to0^0_M}$, we use (3.15) to reach

$$\mathcal{A}^{\Delta_1,\Delta_2,\Delta}_{1^0_0+2^-_0\to0^0_M} = \frac{\Delta_1N^+_{\Delta_1}}{N^+_{\Delta_1+1}}\int d^4X_1\phi^+_{\Delta_1+1}(x_1,X_1)\phi^-_{\Delta,M}(x_0,X_1)$$
$$\times\widehat{q}^\mu_1(x_1)\left[V^+_{\Delta_2;\mu-}(z_2,X_1)+W^+_{\Delta_2;\mu-}(z_2,X_1)\right]\,. \tag{5.12}$$

The term involves $V^+_{\Delta_2;\mu-}(z_2,X_1)$ is given by

$$\frac{\Delta_1N^+_{\Delta_1}}{N^+_{\Delta_1+1}}\int d^4X_1\phi^+_{\Delta_1+1}(z_1,X_1)\widehat{q}^\mu_1V^+_{\Delta_2;\mu-}(z_2,X_1)\phi^-_{\Delta,M}(z_0,X_1)$$
$$= \frac{2\Delta_1N^+_{\Delta_1}}{N^+_{\Delta_1+1}N^+_{\Delta_2}}z_{12}\int d^4X_1\phi^+_{\Delta_1+1}(z_1,X_1)\phi^+_{\Delta_2}(z_2,X_1)\phi^-_{\Delta,M}(z_0,X_1) \tag{5.13}$$
$$= \frac{2\Delta_1N^+_{\Delta_1}}{N^+_{\Delta_1+1}N^+_{\Delta_2}}z_{12}\mathcal{A}^{\Delta_1+1,\Delta_2,\Delta}_{1^0_0+2^0_0\to0^0_M}\,.$$

Moreover we note that the term involves $W^+_{\Delta_2;\mu-}(z_2,X_1)$ can be computed from

$$\frac{\Delta_1N^+_{\Delta_1}}{N^+_{\Delta_1+1}}\int d^4X_1\phi^+_{\Delta_1+1}(z_1,X_1)\widehat{q}^\mu_1W^+_{\Delta_2;\mu-}(z_2,X_1)\phi^-_{\Delta,M}(z_0,X_1)$$
$$= \frac{-2\Delta_1N^+_{\Delta_1}}{\Delta_2N^+_{\Delta_1+1}N^+_{\Delta_2}}|z_{12}|^2\partial_{\bar{z}_2}\int d^4X_1\phi^+_{\Delta_1+1}(z_1,X_1)\phi^+_{\Delta_2}(z_2,X_1)\phi^-_{\Delta,M}(z_0,X_1) \tag{5.14}$$
$$= \frac{-2\Delta_1N^+_{\Delta_1}}{\Delta_2N^+_{\Delta_1+1}N^+_{\Delta_2}}|z_{12}|^2\partial_{\bar{z}_2}\mathcal{A}^{\Delta_1+1,\Delta_2,\Delta}_{1^0_0+2^0_0\to0^0_M}\,.$$

With the help of (5.4) and (5.6), we get

$$\mathcal{A}^{\Delta_1,\Delta_2,\Delta_3}_{1_0^0+2_0^-\to 3_M^0} = \frac{C^{\Delta_1,\Delta_2,\Delta_3}_{1_0^0+2_0^-\to 3_M^0}}{z_{12}^{h_1+h_2-h_3}z_{13}^{h_1-h_2+h_3}z_{23}^{h_2+h_3-h_1}\bar{z}_{12}^{\bar{h}_1+\bar{h}_2-\bar{h}_3}\bar{z}_{13}^{\bar{h}_1-\bar{h}_2+\bar{h}_3}\bar{z}_{23}^{\bar{h}_2+\bar{h}_3-\bar{h}_1}} \ , \tag{5.15}$$

where $C^{\Delta_1,\Delta_2,\Delta_3}_{1_0^0+2_0^-\to 3_M^0}$ is

$$C^{\Delta_1,\Delta_2,\Delta_3}_{1_0^0+2_0^-\to 3_M^0} = \frac{(2\pi)^4\Delta_1 N^+_{\Delta_1}M^{\Delta_1+\Delta_2-\Delta_3-1}\Gamma[\frac{\Delta_{12}+\Delta_3+1}{2}]\Gamma[\frac{-\Delta_{12}+\Delta_3+1}{2}]}{2^{\Delta_1+\Delta_2-2}N^+_{\Delta_1+1}N^+_{\Delta_2}\Delta_2\Gamma[\Delta_3]} \ . \tag{5.16}$$

and the conformal weights are $(h_1,\bar{h}_1) = (\Delta_1/2,\Delta_1/2)$, $(h_2,\bar{h}_2) = ((\Delta_2-1)/2,(\Delta_2+1)/2)$, and $(h_3,\bar{h}_3) = (\Delta_3/2,\Delta_3/2)$. As a result, the celestial amplitude $_s\mathcal{A}^{\Delta_i}_{1_0^0+2_0^-\to 3_0^0+4_0^-}(z_i)$ takes the form as

$$\begin{aligned}
_s\mathcal{A}^{\Delta_i}_{1_0^0+2_0^-\to 3_0^0+4_0^-} =& \frac{-4}{2\pi i}\int_0^{+\infty}\frac{dM}{(2\pi)^{12}}\frac{iM^3}{-M^2+i\epsilon}\\
&\times\int_1^{1+i\infty}d\Delta\frac{(\Delta-1)^2}{\pi^2}C^{\Delta_1,\Delta_2,\Delta_3}_{1_0^0+2_0^-\to 3_M^0}C^{2-\Delta,\Delta_3,\Delta_4}_{0_M^0\to 3_0^0+4_0^-}\Psi^{h_{12},h_{34};\bar{h}_{12},\bar{h}_{34}}_{\Delta,J=0} \ .
\end{aligned}\tag{5.17}$$

## 5.3 $t$-channel massless scalar four-point amplitude

Let us apply the split representation (4.9) to the $t$-channel celestial scalar amplitude (3.7). We note that in this case, the internal momentum $p = q_1 - q_2$ satisfies $p^2 \geqslant 0$. Thus only the last two lines in (4.9) contribute. We find

$$\begin{aligned}
&_t\mathcal{A}^{\Delta_i}_{1_0^0+2_0^0\to 3_0^0+4_0^0}(x_i)\\
&= \frac{1}{2}\int_0^\infty\frac{dM}{(2\pi)^{d+2}}\frac{iM^{d+1}}{M^2+m^2}\left[\frac{1}{2\pi i}\sum_{\epsilon=0,1}\int_{\frac{d}{2}}^{\frac{d}{2}+i\infty}\frac{d\Delta}{\mu(\Delta)}\int d^dx_0\ \mathcal{A}^{\Delta_1,\Delta_3,\Delta}_{1_0^0\to 3_0^0+0_{iM,\epsilon}^0}\mathcal{A}^{d-\Delta,\Delta_2,\Delta_4}_{0_{iM,\epsilon}^0+2_0^0\to 4_0^0}\right.\\
&\quad \left.+ \alpha_d\sum_{(\Delta,\epsilon)\in D_{\mathrm{dS}}}\mathrm{Res}\left(\frac{1}{\mu(\Delta)}\int d^dx_0\mathcal{A}^{\Delta_1,\Delta_3,\Delta}_{1_0^0\to 3_0^0+0_{iM,\epsilon}^0}\mathcal{A}^{d-\Delta,\Delta_2,\Delta_4}_{0_{iM,\epsilon}^0+2_0^0\to 4_0^0}\right)\right] \ .
\end{aligned}\tag{5.18}$$

Here, $\mathcal{A}^{\Delta_1,\Delta_3,\Delta}_{1_0^0\to 3_0^0+0_{iM,\epsilon}^0}$ and $\mathcal{A}^{d-\Delta,\Delta_2,\Delta_4}_{0_{iM,\epsilon}^0+2_0^0\to 4_0^0}$ are three-point celestial amplitudes involving two massless scalars and one tachyonic scalar with imaginary mass $iM$:

$$\mathcal{A}^{\Delta_1,\Delta_3,\Delta}_{1_0^0\to 3_0^0+0_{iM,\epsilon}^0} = \int d^{d+2}X_1\phi^+_{\Delta_1}(x_1,X_1)\phi^-_{\Delta_3}(x_3,X_1)\phi_{\Delta,iM,\epsilon}(x_0,X_1) \ , \tag{5.19}$$

and

$$\mathcal{A}^{d-\Delta,\Delta_2,\Delta_4}_{0_{iM,\epsilon}^0+2_0^0\to 4_0^0} = (-1)^\epsilon\int d^{d+2}X_2\phi^+_{\Delta_2}(x_2,X_2)\phi_{d-\Delta,iM,\epsilon}(x_0,X_2)\phi^-_{\Delta_4}(x_4,X_2) \ , \tag{5.20}$$

where the integrals are evaluated in Appendix E. Using the results from Appendix E, we find that $_t\mathcal{A}^{\Delta_i}_{1_0^0+2_0^0\to3_0^0+4_0^0}(x_i)$ can be written as

$$
\begin{aligned}
&_t\mathcal{A}^{\Delta_i}_{1_0^0+2_0^0\to3_0^0+4_0^0}(x_i)\\
&= \frac{1}{2}\int_0^\infty \frac{dM}{(2\pi)^{d+2}}\frac{iM^{d+1}}{M^2+m^2}\left[\frac{1}{2\pi i}\sum_{\epsilon=0,1}\int_{\frac{d}{2}}^{\frac{d}{2}+i\infty}\frac{d\Delta}{\mu(\Delta)}C^{\Delta_1,\Delta_3,\Delta}_{1_0^0\to3_0^0+0^0_{iM,\epsilon}}\,C^{d-\Delta,\Delta_2,\Delta_4}_{0^0_{iM,\epsilon}+2_0^0\to4_0^0}\,\Psi^{\Delta_{13},\Delta_{24}}_{\Delta,J=0}\right.\\
&\quad\left.+\,\alpha_d\sum_{(\Delta,\epsilon)\in D_{\mathrm{dS}}}\mathrm{Res}\left(\frac{1}{\mu(\Delta)}C^{\Delta_1,\Delta_3,\Delta}_{1_0^0\to3_0^0+0^0_{iM,\epsilon}}\,C^{d-\Delta,\Delta_2,\Delta_4}_{0^0_{iM,\epsilon}+2_0^0\to4_0^0}\,\Psi^{\Delta_{13},\Delta_{24}}_{\Delta,J=0}\right)\right],
\end{aligned}
\tag{5.21}
$$

where $C^{\Delta_1,\Delta_3,\Delta}_{1_0^0\to3_0^0+0^0_{iM,\epsilon}}$ and $C^{d-\Delta,\Delta_2,\Delta_4}_{0^0_{iM,\epsilon}+2_0^0\to4_0^0}$ are three-point coefficients appearing in $\mathcal{A}^{\Delta_1,\Delta_2,\Delta}_{1_0^0\to3_0^0+0^0_{iM,\epsilon}}$ and $\mathcal{A}^{d-\Delta,\Delta_2,\Delta_4}_{0^0_{iM,\epsilon}+2_0^0\to4_0^0}$, given by

$$
\begin{aligned}
C^{\Delta_1,\Delta_3,\Delta}_{1_0^0\to3_0^0+0^0_{iM,\epsilon}}&=\frac{M^{\Delta_1+\Delta_3-d-2}\Gamma[1-\Delta]\Gamma[\frac{\Delta-\Delta_{13}}{2}]\Gamma[\frac{\Delta+\Delta_{13}}{2}]}{2^{\Delta_1+\Delta_3-1}\pi}\\
&\quad\times\left(\cos[\frac{\Delta}{2}\pi]\sin[\frac{\Delta_{13}}{2}\pi]\right)^\epsilon\left(\sin[\frac{\Delta}{2}\pi]\cos[\frac{\Delta_{13}}{2}\pi]\right)^{1-\epsilon},
\end{aligned}
\tag{5.22}
$$

and

$$
\begin{aligned}
C^{d-\Delta,\Delta_2,\Delta_4}_{0^0_{iM,\epsilon}+2_0^0\to4_0^0}&=\frac{M^{\Delta_2+\Delta_4-d-2}\Gamma[1-d+\Delta]\Gamma[\frac{d-\Delta-\Delta_{42}}{2}]\Gamma[\frac{d-\Delta+\Delta_{42}}{2}]}{2^{\Delta_2+\Delta_4-1}\pi}\\
&\quad\times\left(\cos[\frac{d-\Delta}{2}\pi]\sin[\frac{\Delta_{42}}{2}\pi]\right)^\epsilon\left(\sin[\frac{d-\Delta}{2}\pi]\cos[\frac{\Delta_{42}}{2}\pi]\right)^{1-\epsilon}.
\end{aligned}
\tag{5.23}
$$

Extracting the $M$-dependence in the three-point coefficients, we can compute the $M$-integral, giving

$$
\begin{aligned}
_t\mathcal{A}^{\Delta_i}_{1_0^0+2_0^0\to3_0^0+4_0^0}&=\frac{i\pi m^{\beta-d}}{4(2\pi)^{d+2}}\csc[\frac{\pi}{2}(d-\beta)]\left[\frac{1}{2\pi i}\sum_{\epsilon=0,1}\int_{\frac{d}{2}}^{\frac{d}{2}+i\infty}\frac{d\Delta}{\mu(\Delta)}C^{\Delta_1,\Delta_3,\Delta}_{1_0^0\to3_0^0+0^0_{i,\epsilon}}\,C^{d-\Delta,\Delta_2,\Delta_4}_{0^0_{i,\epsilon}+2_0^0\to4_0^0}\,\Psi^{\Delta_{13},\Delta_{24}}_{\Delta,J=0}\right.\\
&\quad\left.+\,\alpha_d\sum_{(\Delta,\epsilon)\in D_{\mathrm{dS}}}\mathrm{Res}\left(\frac{1}{\mu(\Delta)}C^{\Delta_1,\Delta_3,\Delta}_{1_0^0\to3_0^0+0^0_{i,\epsilon}}\,C^{d-\Delta,\Delta_2,\Delta_4}_{0^0_{i,\epsilon}+2_0^0\to4_0^0}\,\Psi^{\Delta_{13},\Delta_{24}}_{\Delta,J=0}\right)\right],
\end{aligned}
\tag{5.24}
$$

where we defined $\beta=\sum_{i=1}^4\Delta_i-4$.

We can see that the $t$-channel four-point celestial amplitudes $_t\mathcal{A}^{\Delta_i}_{1_0^0+2_0^0\to3_0^0+4_0^0}(x_i)$ directly factorize into two three-point celestial amplitudes involving one tachyon. As a result, we find that apart from the Plancherel measure $\mu^{-1}(\Delta)$ the coefficient in the conformal partial wave expansion (5.24) is factorized into a product of two three-point coefficients. To further get the conformal block expansion, we use the following expression of the conformal partial wave $\Psi^{\Delta_{13},\Delta_{24}}_{\Delta,J}$:

$$
\Psi^{\Delta_{13},\Delta_{24}}_{\Delta,J}(x_i)=I_{13-24}(x_i)\left(K^{\Delta_2,\Delta_4}_{d-\Delta,J}G^{\Delta_{13},\Delta_{24}}_{\Delta,J}(u,v)+K^{\Delta_1,\Delta_3}_{\Delta,J}G^{\Delta_{13},\Delta_{24}}_{d-\Delta,J}(u,v)\right),
\tag{5.25}
$$

where $I_{13-24}(x_i)$ is the kinematic factor that encodes the scaling behaviour and $G^{\Delta_{13},\Delta_{24}}_{\Delta,J}(u,v)$ are the $t$-channel conformal blocks, which are functions of conformal cross-ratios

$$u = \frac{x_{13}^2 x_{24}^2}{x_{12}^2 x_{34}^2}, \qquad v = \frac{x_{14}^2 x_{23}^2}{x_{12}^2 x_{34}^2}, \tag{5.26}$$

and the coefficients $K^{\Delta_1,\Delta_2}_{\Delta,J}$ are given by

$$K^{\Delta_1,\Delta_2}_{\Delta,J} = \left(-\frac{1}{2}\right)^J \frac{\pi^{\frac{d}{2}} \Gamma[\Delta - \frac{d}{2}] \Gamma[\Delta + J - 1] \Gamma[\frac{d-\Delta+\Delta_{12}+J}{2}] \Gamma[\frac{d-\Delta-\Delta_{12}+J}{2}]}{\Gamma[\Delta-1]\Gamma[d-\Delta+J]\Gamma[\frac{\Delta+\Delta_{12}+J}{2}]\Gamma[\frac{\Delta-\Delta_{12}+J}{2}]}. \tag{5.27}$$

To get the conformal block expansion, we should first learn the pole structures of coefficients appearing in the partial wave expansion (5.24). The existence of trigonometric functions in the three-point coefficients makes the pole structures different for even and odd $d$. When $d$ is odd, $i.e.$, $d = 2s + 1$ with $s = 0, 1, 2, \ldots$, we get

$$\frac{1}{\mu(\Delta)} C^{\Delta_1,\Delta_3,\Delta}_{1^0_0 \to 3^0_0 + 0^0_{i,\epsilon}} C^{d-\Delta,\Delta_2,\Delta_4}_{0^0_{i,\epsilon} + 2^0_0 \to 4^0_0} K^{\Delta_2,\Delta_4}_{d-\Delta,0}$$
$$= \frac{(-1)^s \Gamma[2s+1-\Delta]\Gamma[\Delta-2s]\Gamma[\frac{\Delta-\Delta_{13}}{2}]\Gamma[\frac{\Delta+\Delta_{13}}{2}]\Gamma[\frac{\Delta-\Delta_{42}}{2}]\Gamma[\frac{\Delta+\Delta_{42}}{2}]}{2^{\beta+3}\pi^{\frac{2s+3}{2}}\Gamma[\Delta]\Gamma[\Delta-s-\frac{1}{2}]} \tag{5.28}$$
$$\times \left(\sin[\frac{\Delta_{13}}{2}\pi]\sin[\frac{\Delta_{42}}{2}\pi]\right)^\epsilon \left(\cos[\frac{\Delta_{13}}{2}\pi]\cos[\frac{\Delta_{42}}{2}\pi]\right)^{1-\epsilon},$$

and

$$\frac{1}{\mu(\Delta)} C^{\Delta_1,\Delta_3,\Delta}_{1^0_0 \to 3^0_0 + 0^0_{i,\epsilon}} C^{d-\Delta,\Delta_2,\Delta_4}_{0^0_{i,\epsilon} + 2^0_0 \to 4^0_0} K^{\Delta_1,\Delta_3}_{\Delta,0}$$
$$= \frac{(-1)^s \Gamma[\Delta-2s]\Gamma[\frac{2s+1-\Delta-\Delta_{13}}{2}]\Gamma[\frac{2s+1-\Delta+\Delta_{13}}{2}]\Gamma[\frac{2s+1-\Delta-\Delta_{42}}{2}]\Gamma[\frac{2s+1-\Delta+\Delta_{42}}{2}]}{2^{\beta+3}\pi^{\frac{2s+3}{2}}\Gamma[s-\Delta+\frac{1}{2}]} \tag{5.29}$$
$$\times \left(\sin[\frac{\Delta_{13}}{2}\pi]\sin[\frac{\Delta_{42}}{2}\pi]\right)^\epsilon \left(\cos[\frac{\Delta_{13}}{2}\pi]\cos[\frac{\Delta_{42}}{2}\pi]\right)^{1-\epsilon}.$$

Thus for odd $d$, enclosing the contour in the first line of (5.24) into right-hand $\Delta$-plane only pick up simple poles at $\Delta = 2s + 1 + n$ with $n = 0, 1, 2, \cdots$. Also, it can be checked that only simple poles contribute to the discrete series in the second line of (5.24) for odd $d$ since conformal blocks only have simple poles in odd $d$.

On the other hand, when $d$ is even, $i.e.$, $d = 2s$ with $s = 1, 2, \ldots$, we get

$$\frac{1}{\mu(\Delta)} C^{\Delta_1,\Delta_3,\Delta}_{1^0_0 \to 3^0_0 + 0^0_{i,\epsilon}} C^{d-\Delta,\Delta_2,\Delta_4}_{0^0_{i,\epsilon} + 2^0_0 \to 4^0_0} K^{\Delta_2,\Delta_4}_{d-\Delta,0}$$
$$= \frac{(-1)^{s+1-\epsilon}\Gamma[2s-\Delta]\Gamma[1-\Delta]\Gamma[1-2s+\Delta]\Gamma[\frac{\Delta-\Delta_{13}}{2}]\Gamma[\frac{\Delta+\Delta_{13}}{2}]\Gamma[\frac{\Delta-\Delta_{42}}{2}]\Gamma[\frac{\Delta+\Delta_{42}}{2}]}{2^{\beta+2}\pi^{s+2}\Gamma[\Delta-s]} \tag{5.30}$$
$$\times \left(\cos[\frac{\Delta}{2}\pi]^2\sin[\frac{\Delta_{13}}{2}\pi]\sin[\frac{\Delta_{42}}{2}\pi]\right)^\epsilon \left(\sin[\frac{\Delta}{2}\pi]^2\cos[\frac{\Delta_{13}}{2}\pi]\cos[\frac{\Delta_{42}}{2}\pi]\right)^{1-\epsilon},$$

and

$$\frac{1}{\mu(\Delta)} C^{\Delta_1,\Delta_3,\Delta}_{1^0_0 \to 3^0_0 + 0^0_{i,\epsilon}} C^{d-\Delta,\Delta_2,\Delta_4}_{0^0_{i,\epsilon}+2^0_0 \to 4^0_0} K^{\Delta_1,\Delta_3}_{\Delta,0}$$
$$= \frac{(-1)^{s+1-\epsilon}\Gamma[\Delta]\Gamma[1-\Delta]\Gamma[1-2s+\Delta]\Gamma[\frac{2s-\Delta-\Delta_{13}}{2}]\Gamma[\frac{2s-\Delta+\Delta_{13}}{2}]\Gamma[\frac{2s-\Delta-\Delta_{42}}{2}]\Gamma[\frac{2s-\Delta+\Delta_{42}}{2}]}{2^{\beta+2}\pi^{s+2}\Gamma[s-\Delta]}$$
$$\times \left( \cos[\frac{\Delta}{2}\pi]^2 \sin[\frac{\Delta_{13}}{2}\pi] \sin[\frac{\Delta_{42}}{2}\pi] \right)^\epsilon \left( \sin[\frac{\Delta}{2}\pi]^2 \cos[\frac{\Delta_{13}}{2}\pi] \cos[\frac{\Delta_{42}}{2}\pi] \right)^{1-\epsilon} .$$

$$(5.31)$$

Thus for even $d$, enclosing the contour in the first line of (5.24) into right-hand $\Delta$-plane pick up both simple poles at $\Delta = 1, 2, \cdots, 2s-1$ and double poles at $\Delta = 2s + n$ with $n = 0, 1, 2, \cdots$. Moreover, it can be checked that both simple poles and double poles contribute to the discrete series in the second line of (5.24) for even $d$ since conformal blocks have double poles in even $d$.

In the next two subsections, we will focus on $d = 1$ and $d = 2$ and derive the conformal block expansions of $_t\mathcal{A}^{\Delta_i}_{1^0_0 + 2^0_0 \to 3^0_0 + 4^0_0}(x_i)$.

### 5.3.1 $d = 1$

We first focus on the amplitude in three-dimensional Minkowski spacetime, with a one-dimensional celestial circle. Setting $s = 0$ in (5.28) and (5.29) and enclosing the contour into the right-hand $\Delta$-plane, the continuous part can be written as the form of conformal block expansion. The exchange operators are scalars and have conformal dimension $\Delta = 1 + n$ with $n = 0, 1, 2, \cdots$. The corresponding coefficients are

$$\frac{i\pi m^{\beta-1}}{4(2\pi)^3} \sec[\frac{\beta\pi}{2}] \frac{(-1)^n \Gamma[\frac{1+n-\Delta_{13}}{2}]\Gamma[\frac{1+n+\Delta_{13}}{2}]\Gamma[\frac{1+n-\Delta_{24}}{2}]\Gamma[\frac{1+n+\Delta_{24}}{2}]}{2^{\beta+3}\pi^{\frac{3}{2}}n!\Gamma[n+\frac{1}{2}]} \cos[\frac{\Delta_{13}-\Delta_{42}}{2}\pi] . \quad (5.32)$$

Expanding the conformal partial wave in the discrete part as a sum of conformal block and shadow conformal block, we find that the conformal block part contributes to the conformal block expansions with the same exchange operators. The corresponding coefficients are

$$\frac{i\pi m^{\beta-1}}{4(2\pi)^3} \sec[\frac{\beta\pi}{2}] \sin[\frac{\Delta_{13}+n}{2}\pi] \sin[\frac{\Delta_{42}+n}{2}\pi]$$
$$\times \frac{(-1)^{n+1}\Gamma[\frac{1+n-\Delta_{13}}{2}]\Gamma[\frac{1+n+\Delta_{13}}{2}]\Gamma[\frac{1+n-\Delta_{24}}{2}]\Gamma[\frac{1+n+\Delta_{24}}{2}]}{2^{\beta+3}\pi^{\frac{3}{2}}n!\Gamma[n+\frac{1}{2}]} . \quad (5.33)$$

Moreover, using the fact that

$$\text{Res}_{\Delta=1+n}G_{1-\Delta}^{\Delta_{13},\Delta_{24}} = -\frac{\Gamma[1+n-\Delta_{13}]\Gamma[1+n-\Delta_{42}]}{2\Gamma[2n+2]\Gamma[n+1]\Gamma[-n-\Delta_{13}]\Gamma[-n-\Delta_{42}]}G_{1+n}^{\Delta_{13},\Delta_{24}} \tag{5.34}$$

$$= -\frac{2^{4n+1}\Gamma[\frac{1+n-\Delta_{13}}{2}]\Gamma[\frac{2+n-\Delta_{13}}{2}]\Gamma[\frac{1+n-\Delta_{42}}{2}]\Gamma[\frac{2+n-\Delta_{42}}{2}]}{\Gamma[2n+1]\Gamma[2n+2]\Gamma[\frac{-n-\Delta_{13}}{2}]\Gamma[\frac{1-n-\Delta_{13}}{2}]\Gamma[\frac{-n-\Delta_{42}}{2}]\Gamma[\frac{1-n-\Delta_{42}}{2}]}G_{1+n}^{\Delta_{13}\Delta_{24}} \,,$$

we find that the shadow conformal block part contributes to the conformal block expansions with the same exchange operators. The corresponding coefficients are

$$\frac{i\pi m^{\beta-1}}{4(2\pi)^3}\sec[\frac{\beta\pi}{2}]\cos[\frac{\Delta_{13}+n}{2}\pi]\cos[\frac{\Delta_{42}+n}{2}\pi]$$

$$\times \frac{(-1)^n\Gamma[\frac{1+n-\Delta_{13}}{2}]\Gamma[\frac{1+n+\Delta_{13}}{2}]\Gamma[\frac{1+n-\Delta_{42}}{2}]\Gamma[\frac{1+n+\Delta_{42}}{2}]}{2^{\beta+3}\pi^{\frac{3}{2}}n!\Gamma[n+\frac{1}{2}]} \,, \tag{5.35}$$

where we used the identity

$$\Gamma[z]\Gamma[z+\frac{1}{2}] = 2^{1-2z}\pi^{\frac{1}{2}}\Gamma[2z] \,. \tag{5.36}$$

Adding all three contributions together, we get the following conformal block expansions

$$_t\mathcal{A}_{1_0^0+2_0^0\to3_0^0+4_0^0}^{\Delta_i}(x_i) = I_{13-24}\sum_{n=0}^{+\infty}\mathcal{C}_{1+n}^{\Delta_i}G_{1+n}^{\Delta_{13},\Delta_{24}}(\chi) \,, \tag{5.37}$$

where the conformal cross-ratio $\chi = \frac{z_{13}z_{24}}{z_{12}z_{34}}$ and $\mathcal{C}_{1+n}^{\Delta_i}$ is given by

$$\mathcal{C}_{1+n}^{\Delta_i} = \frac{i\pi m^{\beta-1}}{2^{\beta+4}(2\pi)^3}\sec[\frac{\beta\pi}{2}]\frac{(-1)^n2^{2n}\Gamma[\frac{1+n-\Delta_{13}}{2}]\Gamma[\frac{1+n-\Delta_{42}}{2}]}{\Gamma[2n+1]\Gamma[\frac{1-n-\Delta_{13}}{2}]\Gamma[\frac{1-n-\Delta_{42}}{2}]} \,. \tag{5.38}$$

In the Appendix F, we compute the conformal block expansion of $_t\mathcal{A}_{1_0^0+2_0^0\to3_0^0+4_0^0}^{\Delta_i}$ in one dimensional CCFT by using the alpha-space approach and find the results agree with (5.38).

### 5.3.2 $d = 2$

Now, we consider the amplitude in four-dimensional Minkowski spacetime, with a two-dimensional celestial sphere. When $d = 2$, after defining $f_\epsilon^{\Delta_i,\Delta}$ as

$$f_\epsilon^{\Delta_i,\Delta} = \frac{(-1)^\epsilon\Gamma[\frac{\Delta-\Delta_{13}}{2}]\Gamma[\frac{\Delta+\Delta_{13}}{2}]\Gamma[\frac{\Delta-\Delta_{24}}{2}]\Gamma[\frac{\Delta+\Delta_{24}}{2}]}{2^{\beta+2}\pi^3}$$

$$\times \left(\cos[\frac{\Delta}{2}\pi]^2\sin[\frac{\Delta_{13}}{2}\pi]\sin[\frac{\Delta_{42}}{2}\pi]\right)^\epsilon\left(\sin[\frac{\Delta}{2}\pi]^2\cos[\frac{\Delta_{13}}{2}\pi]\cos[\frac{\Delta_{42}}{2}\pi]\right)^{1-\epsilon} \,, \tag{5.39}$$

we have

$$\frac{1}{\mu(\Delta)}C_{1_0^0\to3_0^0+0_{i,\epsilon}^0}^{\Delta_1,\Delta_3,\Delta}C_{0_{i,\epsilon}^0+2_0^0\to4_0^0}^{2-\Delta,\Delta_2,\Delta_4}K_{2-\Delta,0}^{\Delta_2,\Delta_4} = \Gamma[2-\Delta]\Gamma[1-\Delta]f_\epsilon^{\Delta_i,\Delta} \,, \tag{5.40}$$

and

$$\frac{1}{\mu(\Delta)} C^{\Delta_1,\Delta_3,\Delta}_{1^0_0\to3^0_0+0^0_{i,\epsilon}} C^{2-\Delta,\Delta_2,\Delta_4}_{0^0_{i,\epsilon}+2^0_0\to4^0_0} = \Gamma[\Delta]\Gamma[\Delta-1]f^{\Delta_i,2-\Delta}_\epsilon \ . \tag{5.41}$$

Enclosing the contour into the right-hand $\Delta$-plane leads to [5]

$$
{}_t\mathcal{A}^{\Delta_i}_{1^0_0+2^0_0\to3^0_0+4^0_0}(x_i)
$$
$$
= \frac{i\pi m^{\beta-2}}{4(2\pi)^4}\csc[\frac{\pi\beta}{2}]\Bigg[f^{\Delta_i,1}_0 G^{\Delta_{13},\Delta_{24}}_{\frac{1}{2},\frac{1}{2}} - \frac{1}{2}\sum_{n=0}^{+\infty}\mathrm{Res}_{\Delta=2+n}\bigg(\Gamma[2-\Delta]\Gamma[1-\Delta]f^{\Delta_i,\Delta}_{n+1\,\mathrm{mod}\,2}\, G^{\Delta_{13},\Delta_{24}}_{\frac{\Delta}{2},\frac{\Delta}{2}}\bigg)
$$
$$
-\frac{1}{2}\sum_{n=0}^{+\infty}\mathrm{Res}_{\Delta=-n}\bigg(\Gamma[2-\Delta]\Gamma[1-\Delta]f^{\Delta_i,\Delta}_{n+1\,\mathrm{mod}\,2}\, G^{\Delta_{13},\Delta_{24}}_{\frac{\Delta}{2},\frac{\Delta}{2}}\bigg)\Bigg]\ , \tag{5.42}
$$

where we used the fact that

$$
\sum_{n=1}^{+\infty}\mathrm{Res}_{\Delta=1+n}\bigg(\Gamma[\Delta]\Gamma[\Delta-1]f^{\Delta_i,2-\Delta}_{n\,\mathrm{mod}\,2}\, G^{\Delta_{13},\Delta_{24}}_{\frac{2-\Delta}{2},\frac{2-\Delta}{2}}\bigg)
$$
$$
= -\sum_{n=0}^{+\infty}\mathrm{Res}_{\Delta=-n}\bigg(\Gamma[2-\Delta]\Gamma[1-\Delta]f^{\Delta_i,\Delta}_{n+1\,\mathrm{mod}\,2}\, G^{\Delta_{13},\Delta_{24}}_{\frac{\Delta}{2},\frac{\Delta}{2}}\bigg)\ . \tag{5.43}
$$

We note that the conformal block $G^{\Delta_{13},\Delta_{24}}_{\frac{\Delta}{2},\frac{\Delta}{2}}$ itself has double poles located at $\Delta=-n$. Near these poles, the conformal block $G^{\Delta_{13},\Delta_{24}}_{\frac{\Delta}{2},\frac{\Delta}{2}}$ can be expanded as (G.31). This leads to

$$
-\frac{1}{2}\sum_{n=0}^{+\infty}\mathrm{Res}_{\Delta=-n}\bigg[\Gamma[2-\Delta]\Gamma[1-\Delta]f^{\Delta_i,\Delta}_{n+1\,\mathrm{mod}\,2}\, G^{\Delta_{13},\Delta_{24}}_{\frac{\Delta}{2},\frac{\Delta}{2}}\bigg] \tag{5.44}
$$
$$
= -\frac{1}{2}\sum_{n=0}^{+\infty}\Bigg[\frac{f^{\Delta_i,n+2}_{n+1\,\mathrm{mod}\,2}}{n!(n+1)!}\bigg((B^{h_i}_n + \widetilde{B}^{h_i}_{-n} - \psi(1+n) - \psi(2+n))G^{\Delta_{13},\Delta_{24}}_{\frac{n+2}{2},\frac{n+2}{2}} + \frac{\partial G^{\Delta_{13},\Delta_{24}}_{\frac{\Delta}{2},\frac{\Delta}{2}}}{\partial\Delta}\bigg|_{\Delta=n+2}\bigg)
$$
$$
+\Gamma[n+1](-\frac{n}{2}-h_{13})_{n+1}(-\frac{n}{2}+h_{24})_{n+1}f^{\Delta_i,-n}_{n+1\,\mathrm{mod}\,2}\bigg(G^{\Delta_{13},\Delta_{24}}_{\mathrm{sta},-\frac{n}{2},\frac{n}{2}+1} + G^{\Delta_{13},\Delta_{24}}_{\mathrm{sta},\frac{n}{2}+1,-\frac{n}{2}}\bigg)\Bigg]\ ,
$$

where we used the fact that

$$(-\frac{n}{2}-h_{13})^2_{n+1}(-\frac{n}{2}+h_{24})^2_{n+1}f^{\Delta_i,-n}_{n+1\,\mathrm{mod}\,2} = f^{\Delta_i,n+2}_{n+1\,\mathrm{mod}\,2}\ . \tag{5.45}$$

The constant $\widetilde{B}^{h_i}_{-n}$ is

$$\widetilde{B}^{h_i}_{-n} = \frac{\psi(\frac{-n}{2}-h_{13}) + \psi(\frac{-n}{2}+h_{13}) + \psi(\frac{-n}{2}-h_{24}) + \psi(\frac{-n}{2}+h_{24})}{2}\ , \tag{5.46}$$

---

[5]We use the notation $G^{\Delta_{13},\Delta_{24}}_{h,\bar{h}}$ to denote the two dimensional $t$-channel conformal block with external conformal dimension $\Delta_i$ and spin $J=0$ and internal weight $(h,\bar{h})$.

where $\psi(x)$ is the digamma function. On the other hand, we have

$$-\frac{1}{2}\sum_{n=0}^{+\infty}\mathrm{Res}_{\Delta=2+n}\left[\Gamma[2-\Delta]\Gamma[1-\Delta]f_{n+1\,\mathrm{mod}\,2}^{\Delta_i,\Delta}\,G_{\frac{\Delta}{2},\frac{\Delta}{2}}^{\Delta_{13},\Delta_{24}}\right] \tag{5.47}$$

$$=-\frac{1}{2}\sum_{n=0}^{+\infty}\frac{f_{n+1\,\mathrm{mod}\,2}^{\Delta_i,n+2}}{n!(n+1)!}\left((-\widetilde{B}_{2+n}^{h_i}+\psi(1+n)+\psi(2+n))G_{\frac{n+2}{2},\frac{n+2}{2}}^{\Delta_{13},\Delta_{24}}-\left.\frac{\partial G_{\frac{\Delta}{2},\frac{\Delta}{2}}^{\Delta_{13},\Delta_{24}}}{\partial\Delta}\right|_{\Delta=n+2}\right).$$

As a result, we finally get that

$$_t\mathcal{A}_{1_0^-+2_0^-\to3_0^-+4_0^-}^{\Delta_i}(x_i) \tag{5.48}$$

$$=\frac{i\pi m^{\beta-2}}{4(2\pi)^4}\csc[\frac{\pi\beta}{2}]\Bigg\{f_0^{\Delta_i,1}G_{\frac{1}{2},\frac{1}{2}}^{\Delta_{13},\Delta_{24}}-\frac{1}{2}\sum_{n=0}^{+\infty}\left[\frac{f_{n+1\,\mathrm{mod}\,2}^{\Delta_i,n+2}}{n!(n+1)!}(B_n^{h_i}+\widetilde{B}_{-n}^{h_i}-\widetilde{B}_{2+n}^{h_i})G_{\frac{n+2}{2},\frac{n+2}{2}}^{\Delta_{13},\Delta_{24}}\right.$$

$$\left.+\Gamma[n+1](-\frac{n}{2}-h_{13})_{n+1}(-\frac{n}{2}+h_{24})_{n+1}f_{n+1\,\mathrm{mod}\,2}^{\Delta_i,-n}\left(G_{\mathrm{sta},-\frac{n}{2},\frac{n}{2}+1}^{\Delta_{13},\Delta_{24}}+G_{\mathrm{sta},\frac{n}{2}+1,-\frac{n}{2}}^{\Delta_{13},\Delta_{24}}\right)\right]\Bigg\},$$

where the conformal blocks $G_{\mathrm{sta},-\frac{n}{2},\frac{n}{2}+1}^{\Delta_{13},\Delta_{24}}$ and $G_{\mathrm{sta},\frac{n}{2}+1,-\frac{n}{2}}^{\Delta_{13},\Delta_{24}}$ associated with the staggered module are defined in (A.3) and (A.6).

We note that in the conformal block expansion of $_t\mathcal{A}_{1_0^-+2_0^-\to3_0^-+4_0^-}^{\Delta_i}$ there are scalar exchanges with conformal dimension $\Delta=n$ for $n=1,2,3,\cdots$. These scalar exchanges have also been observed in the conformal block expansion of $_t\mathcal{A}_{1_0^-+2_0^-\to3_0^-+4_0^-}^{\Delta_i}$ in Klein space [7]. Besides the scalar exchanges, the staggered modules, which are absent in the Klein space, appear in the conformal block expansion of $_t\mathcal{A}_{1_0^-+2_0^-\to3_0^-+4_0^-}^{\Delta_i,m}$ in Minkowski space. As we have shown in Appendix A, the staggered modules are generated by two operators $\mathcal{O}^1$ and $\mathcal{O}^2$. The operator $\mathcal{O}^1$, which has conformal dimension one and integer spin, is primary, while the operator $\mathcal{O}^2$, which has conformal dimension $n+2$ and spin-zero, is neither primary nor descendant. The physical correspondence of operators $\mathcal{O}^1$ and $\mathcal{O}^2$ in Minkowski space is obscure. One possible interpretation is that $\mathcal{O}^1$ and $\mathcal{O}^2$ may correspond to soft particles and their Goldstone bosons in the Minkowski space. We will leave this in the future work [27].

## 5.4  Contact scalar four-point amplitude

The split representation can also be used to compute the conformal partial wave expansion of four-point contact amplitude. We take massless scalar four-point contact amplitude as an example. The corresponding celestial amplitude $_c\mathcal{A}_{1_0^-+2_0^-\to3_0^-+4_0^-}^{\Delta_i}$ is

$$_c\mathcal{A}_{1_0^-+2_0^-\to3_0^-+4_0^-}^{\Delta_i}=\int d^4X\,\phi_{\Delta_1}^+(x_1,X)\phi_{\Delta_2}^+(x_2,X)\phi_{\Delta_1}^-(x_3,X)\phi_{\Delta_4}^-(x_1,X)\,. \tag{5.49}$$

Using the fact that

$$\partial_X^2 \mathcal{K}_{m=0}(X,Y) = i\delta^{(4)}(X-Y) , \tag{5.50}$$

we find that $_c\mathcal{A}^{\Delta_i}_{1_0^0+2_0^0\to3_0^0+4_0^0}$ can be re-written as

$$
_c\mathcal{A}^{\Delta_i}_{1_0^0+2_0^0\to3_0^0+4_0^0} = -i\int d^4X_1 d^4X_2 \phi^+_{\Delta_1}(x_1,X_1)\phi^+_{\Delta_2}(x_2,X_1)
$$
$$
\times \partial_{X_1}^2 \mathcal{K}_{m=0}(X_1,X_2)\phi^-_{\Delta_3}(x_3,X_2)\phi^-_{\Delta_4}(x_4,X_2) . \tag{5.51}
$$

Integrating by parts and using the equation of motion $\partial_X^2 \phi^\pm_\Delta(x,X) = 0$ leads to

$$
_c\mathcal{A}^{\Delta_i}_{1_0^0+2_0^0\to3_0^0+4_0^0} = -2i\int d^4X_1 d^4X_2 \partial_{X_1\mu}\phi^+_{\Delta_1}(x_1,X_1)\partial^\mu_{X_1}\phi^+_{\Delta_2}(x_2,X_1)
$$
$$
\times \mathcal{K}_{m=0}(X_1,X_2)\phi^-_{\Delta_3}(x_3,X_2)\phi^-_{\Delta_4}(x_4,X_2)
$$
$$
= \frac{2i\Delta_1\Delta_2 N^+_{\Delta_1}N^+_{\Delta_2}}{N^+_{\Delta_1+1}N^+_{\Delta_2+1}} x_{12}^2 \; _s\mathcal{A}^{\Delta_1+1,\Delta_2+1,\Delta_3,\Delta_4}_{1_0^0+2_0^0\to3_0^0+4_0^0}\bigg|_{m=0} . \tag{5.52}
$$

(5.52) relates the four-point contact diagrams to the $s$-channel amplitude. Using (5.7), we get

$$
_c\mathcal{A}^{\Delta_i}_{1_0^0+2_0^0\to3_0^0+4_0^0} \tag{5.53}
$$
$$
= \frac{-2i\Delta_1\Delta_2 N^+_{\Delta_1}N^+_{\Delta_2}}{N^+_{\Delta_1+1}N^+_{\Delta_2+1}(2\pi)^{d+2}}\delta(\beta-d+2)\int_{\frac{d}{2}}^{\frac{d}{2}+i\infty} \frac{d\Delta}{\mu(\Delta)} C^{\Delta_1+1,\Delta_2+1,\Delta}_{1_0^0+2_0^0\to0_1^0} C^{d-\Delta,\Delta_3,\Delta_4}_{0_1^0\to3_0^0+4_0^0} \Psi^{\Delta_{12},\Delta_{34}}_{\Delta,J=0} .
$$

We can also relate the four-point contact diagrams to the $t$-channel amplitude by noting that

$$
_c\mathcal{A}^{\Delta_i}_{1_0^0+2_0^0\to3_0^0+4_0^0} = -i\int d^4X_1 d^4X_2 \phi^+_{\Delta_1}(x_1,X_1)\phi^-_{\Delta_3}(x_3,X_1)
$$
$$
\times \partial_{X_1}^2 \mathcal{K}_{m=0}(X_1,X_2)\phi^+_{\Delta_2}(x_2,X_2)\phi^-_{\Delta_4}(x_4,X_2) . \tag{5.54}
$$

Integrating by parts and using the equation of motion $\partial_X^2 \phi^\pm_\Delta(x,X) = 0$ leads to

$$
_c\mathcal{A}^{\Delta_i}_{1_0^0+2_0^0\to3_0^0+4_0^0} = \frac{2i\Delta_1\Delta_3 N^+_{\Delta_1}N^-_{\Delta_3}}{N^+_{\Delta_1+1}N^-_{\Delta_3+1}} x_{13}^2 \; _t\mathcal{A}^{\Delta_1+1,\Delta_2,\Delta_3+1,\Delta_4}_{1_0^0+2_0^0\to3_0^0+4_0^0}\bigg|_{m=0} . \tag{5.55}
$$

With the help of (5.21), we get

$$
_c\mathcal{A}^{\Delta_i}_{1_0^0+2_0^0\to3_0^0+4_0^0} \tag{5.56}
$$
$$
= \frac{-\Delta_1\Delta_3 N^+_{\Delta_1}N^-_{\Delta_3}}{N^+_{\Delta_1+1}N^-_{\Delta_3+1}(2\pi)^{d+1}}\delta(\beta-d+2)\Bigg[\frac{1}{2\pi i}\sum_{\epsilon=0,1}\int_{\frac{d}{2}}^{\frac{d}{2}+i\infty} \frac{d\Delta}{\mu(\Delta)} C^{\Delta_1+1,\Delta_3+1,\Delta}_{1_0^0\to3_0^0+0_{i,\epsilon}^0} C^{d-\Delta,\Delta_2,\Delta_4}_{0_{i,\epsilon}^0+2_0^0\to4_0^0} \Psi^{\Delta_{13},\Delta_{24}}_{\Delta,J=0}
$$
$$
+ \alpha_d\sum_{(\Delta,\epsilon)\in D_{\mathrm{dS}}} \mathrm{Res}\bigg(\frac{1}{\mu(\Delta)} C^{\Delta_1+1,\Delta_3+1,\Delta}_{1_0^0\to3_0^0+0_{i,\epsilon}^0} C^{d-\Delta,\Delta_2,\Delta_4}_{0_{i,\epsilon}^0+2_0^0\to4_0^0} \Psi^{\Delta_{13},\Delta_{24}}_{\Delta,J=0}\bigg)\Bigg] .
$$

# 6   Conclusion

In this paper, we derived the split representation (4.9) of the Feynman propagator in Minkowski space. Roughly, the Feynman propagator in Minkowski space can be written as a product of a pair of (tachyonic) massive conformal primary bases with conformal dimensions $\Delta$ and $d - \Delta$, and with the common boundary point, the parameter $\Delta$ and the (imaginary) mass integrated.

Using the split representation, we computed the conformal partial wave expansion of various celestial amplitudes, including the $s$-channel massless scalar celestial amplitude (5.7), the $s$-channel celestial Compton amplitude (5.17), the $t$-channel massless scalar celestial amplitude (5.24), and the contact four-point amplitude (5.53) and (5.56). We found that the split representation leads to the conformal partial wave expansions of these amplitudes. For the four-point tree-level celestial amplitudes, apart from the Plancherel measure the coefficient in the conformal partial wave expansion is factorized into a product of two three-point coefficients involving massive or/and tachyonic scalars. Without much effort, one can verify that similar factorizations also occur in the conformal partial wave expansion of higher-point tree-level celestial amplitudes.

With the conformal partial wave expansion in hand, we obtained the conformal block expansion (5.48) of the $t$-channel massless scalar celestial amplitude in two-dimensional Euclidean celestial CFT. Interestingly, we found that a new class of modules — the chiral staggered modules — appear in the spectrum of the conformal block expansion. The staggered modules are extensions of Verma modules and contain operators that are neither primary nor descendent. It is noteworthy that staggered modules also appear in Carrollian CFTs [28], and one can explore if there are possible relations of the staggered modules in celestial and Carrollian CFTs.

Although we used the conformal primary basis given by a Mellin transform on the plane waves for massless particles in this paper, the split representation does not depend on the choice of the conformal primary basis for external operators. Recently, the celestial amplitudes of massless particles have been studied using shadow conformal primary basis, see e.g. [26, 29, 30] and the references therein. Our computations can be easily generalized to these cases.

Our work leads to many directions for future research. Firstly, the split representation (4.9) can be used to compute the conformal block expansion of more general celestial amplitudes such as those of higher points and/or with loops.[6] As mentioned in the introduction, studying the pole structures of general celestial amplitudes is significantly important for un-

---

[6]Recent progress on the computation of two-dimensional $M$-point global conformal blocks can be found in [31–33] and references therein.

derstanding locality in celestial holography. In addition, by taking the three-colinear limit in the higher-point scattering amplitudes followed by a Mellin transform, the authors of [34] showed that the celestial OPE between two particles contains a term with a branch cut which is conjectured to describe multi-particle states. Armed with the split representation, it would be interesting to study this branch cut by taking the OPE limit directly in the conformal block expansion of higher-point celestial amplitudes.

Secondly, it would be of great interest to study the bulk correspondence of the staggered module we have found. Since the conformal dimensions of the two operators in the staggered module are integers, one possibility is that the two operators may correspond to soft particles and their Goldstone modes in the Minkowski space. Another observation is that the conformal primary basis may not be suitable for transforming the neither primary nor descendant operator in the staggered module to certain bulk operator, and it is necessary to find a new basis to resolve this obstacle.

Another avenue is to generalize the split representation in our paper to spinning Feynman propagators. In this paper we only derived the split representation for the spinless Feynman propagator. In order to compute the celestial amplitudes involving spinning propagators, it would be important to derive the split representations for spinning Feynman propagators. Based on the split representations for spinning Feynman propagators, one can study the conformal block expansion of celestial amplitudes with exchanged photons and gravitons, and possibly find a new spectrum in the celestial holography.

Finally, one may generalize the split representation in our paper to Klein space. The Klein space can be foliated by $AdS_3/\mathbb{Z}$, and similar to the dS spacetime, the resolution of identity on $AdS_3/\mathbb{Z}$ contains both continuous and discrete parts [22]. Conformal block expansion of celestial massless $t$-channel amplitude in Klein space has been obtained in [7] by using the alpha-space approach. We expect that the split representation can provide a more efficient way to get the conformal block expansions for celestial amplitudes in Klein space.

# Acknowledgements

We thank Prahar Mitra and Tianqing Zhu for the inspiring discussions. CC is grateful to the organizers of the Kickoff Workshop of the Simons Collaboration on Celestial Holography. CC is partly supported by National Key R&D Program of China (NO. 2020YFA0713000).

# A    Staggered modules of $2d$ global conformal algebra

The staggered modules appear in the study of logarithmic CFTs with Virasoro conformal symmetry [35–38] and Carrollian CFTs [28, 39–41]. Unlike the Verma modules, they are generically non-unitary and contain neither primary nor descendant operators. In this appendix we discuss the staggered modules for the $1d/2d$ global conformal symmetry.

In one dimensional CFTs, for $\Delta = -\frac{n}{2}$, $n \in \mathbb{N}$, we find the following staggered module generated by two operators $\mathcal{O}^1$ and $\mathcal{O}^2$, which satisfy

$$
\begin{aligned}
L_0|\mathcal{O}^1\rangle &= \Delta|\mathcal{O}^1\rangle, & L_1|\mathcal{O}^1\rangle &= 0, \\
L_0|\mathcal{O}^2\rangle &= (1-\Delta)|\mathcal{O}^2\rangle, & L_1|\mathcal{O}^2\rangle &= aL_{-1}^{-2\Delta}|\mathcal{O}^1\rangle.
\end{aligned}
\tag{A.1}
$$

The actions on the descendants $|n,1\rangle := L_{-1}^n|\mathcal{O}^1\rangle, |n,2\rangle := L_{-1}^n|\mathcal{O}^2\rangle$ are

$$
\begin{aligned}
L_0|n,1\rangle &= (\Delta+n)|n,1\rangle, \\
L_1|n,1\rangle &= n(2\Delta+n-1)|n-1,1\rangle, \\
L_0|n,2\rangle &= (1-\Delta+n)|n,2\rangle, \\
L_1|n,2\rangle &= n(-2\Delta+n+1)|n-1,2\rangle + a|n-2\Delta,1\rangle.
\end{aligned}
\tag{A.2}
$$

It is easy to check that there is a gauge redundancy of $|\mathcal{O}^2\rangle$: the actions (A.1) and (A.2) are invariant under the redefinition $|\mathcal{O}^2\rangle \to |\mathcal{O}^2\rangle + bL_{-1}^{1-2\Delta}|\mathcal{O}^1\rangle$.

The primary state $|\mathcal{O}^1\rangle$ generates the submodule, while $|\mathcal{O}^2\rangle$ is neither a primary state nor a descendant of any other state. The action of $L_0$ is diagonal and the operator mixing only happens for $L_1|n,2\rangle$. As a result, the action of the Casimir element is not diagonal: $\mathcal{C}|n,1\rangle = 0$ and $\mathcal{C}|n,2\rangle + a|n-2\Delta+1,1\rangle = 0$, or equivalently $\mathcal{C}^2|n,2\rangle = 0$.

The four-point conformal block $G_{\mathrm{sta},-\frac{n}{2}}^{\Delta_{13},\Delta_{24}}(\chi)$ with external scalar primaries and the exchanged staggered module can be computed by either solving the Casimir equations or using the operator product expansion [27]. The results are given by

$$
\begin{aligned}
G_{\mathrm{sta},-\frac{n}{2}}^{\Delta_{13},\Delta_{24}}(\chi) &= \sum_{k=0}^n \frac{(-1)^k(-\frac{n}{2}-\Delta_{13})_k(-\frac{n}{2}+\Delta_{24})_k(n-k)!}{n!\Gamma[k+1]}\chi^{k-\frac{n}{2}} \\
&+ \sum_{s=0}^n \sum_{k=0}^\infty \frac{(-1)^n(-\frac{n}{2}-\Delta_{13})_{k+n+1}(-\frac{n}{2}+\Delta_{24})_{k+n+1}}{n!\Gamma[k+n+2]\Gamma[k+1](k+1+s)}\chi^{k+\frac{n+2}{2}}.
\end{aligned}
\tag{A.3}
$$

In two dimensional CFTs, by the factorization $\mathfrak{so}(2,2) \simeq \mathfrak{so}(2,1) \times \mathfrak{so}(2,1)$, we define two chiral staggered modules with two operators $\mathcal{O}^1$ and $\mathcal{O}^2$. In the first chiral staggered module, the operator $\mathcal{O}^1$ is a primary operator with $(h,\overline{h}) = (-\frac{n}{2}, \frac{n+2}{2})$ for $n \in \mathbb{N}$,

$$
\begin{aligned}
L_0|\mathcal{O}^1\rangle &= -\frac{n}{2}|\mathcal{O}^1\rangle, & \overline{L}_0|\mathcal{O}^1\rangle &= \frac{n+2}{2}|\mathcal{O}^1\rangle \\
L_1|\mathcal{O}^1\rangle &= \overline{L}_1|\mathcal{O}^1\rangle = 0,
\end{aligned}
\tag{A.4}
$$

and the operator $\mathcal{O}^2$ is neither primary nor descendant,

$$
\begin{aligned}
L_0|\mathcal{O}^2\rangle &= \frac{n+2}{2}|\mathcal{O}^2\rangle , \quad \overline{L}_0|\mathcal{O}^2\rangle = \frac{n+2}{2}|\mathcal{O}^2\rangle \\
L_1|\mathcal{O}^2\rangle &= aL_{-1}^n|\mathcal{O}^1\rangle , \quad \overline{L}_1|\mathcal{O}^2\rangle = 0.
\end{aligned}
\tag{A.5}
$$

Then the operator $\mathcal{O}^1$ has conformal dimension $\Delta = 1$ and spin $J = -n$, while $\mathcal{O}^2$ has conformal dimension $\Delta = 1 + n$ and spin $J = 0$.

In a similar way, one can define the second chiral staggered module by switching $L_n$ and $\overline{L}_n$ in (A.4) and (A.5), and the operator $\mathcal{O}^1$ has conformal dimension $\Delta = 1$ and spin $J = n$, while $\mathcal{O}^2$ has conformal dimension $\Delta = 1 + n$ and spin $J = 0$.

Armed with (A.3), the conformal blocks $G^{\Delta_{13},\Delta_{24}}_{\mathrm{sta},-\frac{n}{2},\frac{n+2}{2}}(\chi,\overline{\chi})$ and $G^{\Delta_{13},\Delta_{24}}_{\mathrm{sta},\frac{n+2}{2},-\frac{n}{2}}(\chi,\overline{\chi})$ with external scalar primaries and exchanged chiral staggered module are

$$
\begin{aligned}
G^{\Delta_{13},\Delta_{24}}_{\mathrm{sta},-\frac{n}{2},\frac{n}{2}+1}(\chi,\overline{\chi}) &= G^{h_{13},h_{24}}_{\mathrm{sta},-\frac{n}{2}}(\chi)G^{\overline{h}_{13},\overline{h}_{24}}_{\frac{n}{2}+1}(\overline{\chi}) , \\
G^{\Delta_{13},\Delta_{24}}_{\mathrm{sta},\frac{n}{2}+1,-\frac{n}{2}}(\chi,\overline{\chi}) &= G^{h_{13},h_{24}}_{\frac{n}{2}+1}(\chi)G^{\overline{h}_{13},\overline{h}_{24}}_{\mathrm{sta},-\frac{n}{2}}(\overline{\chi}) ,
\end{aligned}
\tag{A.6}
$$

where $h_i = \overline{h}_i = \frac{\Delta_i}{2}$ for external scalars.

# B    Tempered distributions

In this appendix we summarize the different basis of homogeneous (tempered) distributions, see e.g. [42]. There are two independent distributional solutions of the functional equation $f(\lambda x) = \lambda^a f(x)$, $\lambda > 0$ on the real line $\mathbb{R}$:

$$
x_+^a = \theta(x)|x|^a, \qquad x_-^a = \theta(-x)|x|^a,
\tag{B.1}
$$

and we can also recombine them as

$$
|x|^{a,0} := |x|^a = x_+^a + x_-^a, \qquad |x|^{a,1} := |x|^a \operatorname{sgn}(x) = x_+^a - x_-^a,
\tag{B.2}
$$

where $\theta(x)$ is the Heaviside step function and $\operatorname{sgn}(x)$ is the sign function. The redundant superscript 0 is to stress the companion of $|x|^{a,0}$ to $|x|^{a,1}$.

The above distributions are actually meromorphic functions of $a \in \mathbb{C}$, and there can be simple poles at $a = -1, -2, \ldots$ with residues proportional to the delta distributions $\delta^{(n)}(x)$. To cancel the poles we can either multiply them by Gamma functions, or choose suitable linear combinations. The first approach leads to the regularization of distributions [42],

see also the appendices of [43, 44]. The following normalized distributions are holomorphic functions of $a \in \mathbb{C}$,

$$\frac{1}{\Gamma(a+1)}x_+^a, \quad \frac{1}{\Gamma(a+1)}x_-^a, \quad \frac{1}{\Gamma(\frac{a+1}{2})}|x|^{a,0}, \quad \frac{1}{\Gamma(\frac{a+2}{2})}|x|^{a,1}. \tag{B.3}$$

and the values at the removable poles are listed in table 2.

The second approach is related to boundary values of meromorphic functions, *i.e.* $i\epsilon$-prescription. The basis is $(x \pm i\epsilon)^a$ and the relations to other bases are

$$(x+i\epsilon)^a = x_+^a + e^{ia\pi}x_-^a = \frac{1}{2}(1+e^{ia\pi})|x|^{a,0} + \frac{1}{2}(1-e^{ia\pi})|x|^{a,1}, \tag{B.4}$$

$$(x-i\epsilon)^a = x_+^a + e^{-ia\pi}x_-^a = \frac{1}{2}(1+e^{-ia\pi})|x|^{a,0} + \frac{1}{2}(1-e^{-ia\pi})|x|^{a,1}. \tag{B.5}$$

The poles at $a = -1, -2, \ldots$ get canceled due to the factor $e^{ia\pi} = (-1)^a$, hence $(x \pm i\epsilon)^a$ are holomorphic functions with respect to $a$, see table 2. The inverse relations are

$$|x|^{a,0} = \frac{(x+i\epsilon)^a + e^{i\pi a}(x-i\epsilon)^a}{1+e^{i\pi a}}, \qquad |x|^{a,1} = \frac{(x+i\epsilon)^a - e^{i\pi a}(x-i\epsilon)^a}{1-e^{i\pi a}}. \tag{B.6}$$

For higher dimensions, the spherical-symmetric solution of $f(\lambda x) = \lambda^a f(x)$ on $\mathbb{R}^d$ is $|x|^a$. In the spherical coordinates $x = r\hat{x}$, $\hat{x} \in \mathrm{S}^{d-1}$, the action of $|x|^a$ on a test function is $(|x|^a, f(x)) = \int_{\mathrm{S}^{d-1}} d\hat{x} \int dr\, r^{a+d-1} f(r\hat{x})$, hence the possible poles are at $a = -d-n$, $n \in \mathbb{N}$.

| scaling | $a = -2$ | $-2 < a < -1$ | $a = -1$ | $\cdots$ |
|---|---|---|---|---|
| parity-even | $\frac{1}{x^2}$ | $|x|^a$ | $\delta(x) = \epsilon|x|^{-1+2\epsilon}$ | $\cdots$ |
| parity-odd | $\delta'(x) = -\epsilon|x|^{-2+2\epsilon,1}$ | $|x|^{a,1}$ | $\frac{1}{x}$ | $\cdots$ |
| half-line | / | $x_+^a, x_-^a$ | / | $\cdots$ |
| $i\epsilon$-prescription | $\frac{1}{(x\pm i\epsilon)^2}$ | $(x \pm i\epsilon)^a$ | $\frac{1}{x\pm i\epsilon}$ | $\cdots$ |

Table 1: Three bases of homogeneous distributions on $\mathbb{R}$.

| distributions | values $(n \in \mathbb{N})$ | removed poles |
|---|---|---|
| $\frac{1}{\Gamma(a+1)} x_+^a$ | $\delta^{(n)}(x)$ | $a = -1 - n$ |
| $\frac{1}{\Gamma(a+1)} x_-^a$ | $(-1)^n \delta^{(n)}(x)$ | $a = -1 - n$ |
| $\frac{1}{\Gamma(\frac{a+1}{2})} |x|^{a,0}$ | $\frac{(-1)^n n!}{(2n)!} \delta^{(2n)}(x)$ | $a = -1 - 2n$ |
| $\frac{1}{\Gamma(\frac{a+2}{2})} |x|^{a,1}$ | $\frac{(-1)^{n+1} n!}{(2n+1)!} \delta^{(2n+1)}(x)$ | $a = -2 - 2n$ |
| $(x + i\epsilon)^a$ | $x^{-n-1} - i\pi \frac{(-1)^n}{n!} \delta^{(n)}(x)$ | $a = -1 - n$ |
| $(x - i\epsilon)^a$ | $x^{-n-1} + i\pi \frac{(-1)^n}{n!} \delta^{(n)}(x)$ | $a = -1 - n$ |
| $\frac{1}{\Gamma(\frac{a+d}{2})} |x|^a$ on $\mathbb{R}^d$ | $\frac{(-1)^n \pi^{\frac{d}{2}}}{2^{2n} \Gamma(\frac{d}{2}+n)} \Box^n \delta(x)$ | $a = -d - 2n$ |

Table 2: Regularized distributions holomorphic to $a$.

# C   Harmonic analysis on semisimple symmetric spaces

This appendix serves as the mathematical background of the Appendix D. We provide a comprehensive review of the harmonic analysis on the semisimple symmetric spaces, see e.g. [45–47]. We first introduce the semisimple symmetric spaces and their quasi-regular representations, then sketch the spherical function method of decomposing quasi-regular representations, along with the Fourier transforms and the inversion formulas. To get intuitions, the reader can consider the semisimple symmetric space $G/H$ as the sphere $\mathrm{S}^2 = \mathrm{SO}(3)/\mathrm{SO}(2)$.

The notations and conventions in this appendix are listed below:

- $G$ - a semisimple Lie group with finite center[7];

- $K$ - the maximally compact subgroup of $G$;

- $H$ - a closed subgroup of $G$;

- $G/H$ - the homogeneous space of the pair $(G, H)$;

- $\pi$ - a unitary irreducible representation (abbr. irrep) of $G$ on a Hilbert space $\mathcal{H}_\pi$.

- $\rho$ - the quasi-regular representation on the Hilbert space $L^2(G/H)$;

- $\widetilde{G}$ - the unitary dual - the set of inequivalent unitary irreps of $G$;

- $\widetilde{G}^0$ - the tempered unitary dual - the set of inequivalent unitary irreps in $L^2(G)$;

- $\widetilde{G}^H$ - the set of inequivalent unitary irreps in $L^2(G/H)$.

The inner product $\langle x|y \rangle$ or $(x, y)$ is anti-linear with respect to the first argument. The distributions are tempered in the sense of Schwartz. Besides, it is nontrivial to generalize various concepts and tools from finite-dimensional representations to infinite-dimensional, and we shall ignore the technical details of functional analysis and apply the final results directly.

## C.1   Basic ingredients

In this appendix we recall the necessary concepts of the harmonic analysis on the semisimple symmetric spaces.

---

[7]This finiteness condition excludes the Lorentzian conformal group $\widetilde{\mathrm{SO}}(d, 2)$. As the universal covering of $\mathrm{SO}(d, 2)$, $\widetilde{\mathrm{SO}}(d, 2)$ is nonlinear, *i.e.* not a subgroup of any $\mathrm{GL}(N)$.

**Motivation.** Back to the discussion of Euclidean inversion formula [48–51], the correlation function $f(x_1) := \langle \mathcal{O}(x_1) \ldots \rangle$ lives on the homogeneous space $\mathrm{S}^d = G/P$, where $G = \mathrm{SO}(d+1, 1)$ is the Euclidean conformal group and $P$ is the one-point stabilizer subgroup including dilatation, rotations and special conformal transformations. Then for sufficiently nice $f(x_1)$ we can lift it to a function $f(x) \in L^2(G)$ invariant under the right action of $P$, and decompose $f(x)$ by the Plancherel theorem of $G$,

$$L^2(G) \simeq \int_{\widetilde{G}^0}^{\oplus} d\pi \, \mathcal{H}_\pi, \implies f(x) = \sum_J \int \frac{d\Delta}{N(\Delta, J)} I(\Delta, J) \Psi_{\Delta, J}(x) + \text{discrete part.} \quad \text{(C.1)}$$

Here $\widetilde{G}^0 \subset \widetilde{G}$ is the tempered unitary dual, $N^{-1}(\Delta, J)$ is the Plancherel measure on $\widetilde{G}^0$ ensuring that the isomorphism is an isometry, $\Psi_{\Delta, J}(x)$ is the conformal partial wave and $I(\Delta, J) = (\Psi_{\Delta, J}(x), f(x))$ is the inversion function. According to the dimension, the Plancherel measure of the conformal group may contain both continuous and discrete parts, corresponding to the principal and discrete series representations.

Actually, the lifting procedure is quite technical and can fail for homogeneous spaces, e.g. the dS spacetimes. To overcome this difficulty, the harmonic analysis of Lie groups is generalized to that of homogeneous spaces, and the aim is to decompose quasi-regular representations on homogeneous spaces as direct integrals of unitary irreps. Without further conditions this is still an open problem. One of the best-understood cases is the reductive symmetric spaces of Harish-Chandra class, for which the harmonic analysis has been systematically established in [52–54].

**Semisimple symmetric space.** For a closed subgroup $H \subset G$, the homogeneous space $G/H$ is identified as the left coset space $G/H = \{gH : g \in G\}/\sim$. The origin $e$ of $G/H$ is the coset of identity $eH$, and $H$ is the stabilizer subgroup (i.e. little group) around this origin. When there is an involutive diffeomorphism of $G$ fixing $H$ invariant, there exists a unique $G$-invariant pseudo-Riemannian metric on $G/H$ with constant sectional curvature, and $G/H$ is called a symmetric space. At this step there is no need to require the semisimpleness of $G$, and if $G$ is semisimple indeed, $G/H$ is called a semisimple symmetric space.

For simplicity, we only consider the semisimple symmetric spaces satisfying one of the following conditions:

1. compact case: $G$ and $H$ are compact, e.g. the spheres;

2. Riemannian case: $G$ is non-compact and $H = K$ is maximally compact in $G$, e.g. the Euclidean AdS spaces;

3. non-Riemannian rank-one case: $G$ and $H$ are non-compact and the rank of $G/H$ is one, e.g. the dS spacetimes;

4. group case: $G$ is of the form $G' \times G'$ and $H$ is the diagonal subgroup $H \simeq G'$, e.g. $\text{AdS}_3 / \mathbb{Z} = \text{SL}(2, \mathbb{R})$.

These cases belong to the reductive symmetric spaces of Harish-Chandra class mentioned above, and are broad enough to include the spheres and the real hyperbolic spaces.

We explain the third and fourth cases further. For the third case, the rank-one condition is a technical condition to simplify the later discussion. This is equivalent to the uniqueness of $G$-left-invariant differential operators on $G/H$, *i.e.* the Laplacian operator, which helps the decomposition of $L^2(G/H)$. It also has an equivalent geometric explanation: for any two pairs of points $(x_1, x_2)$ and $(y_1, y_2)$ on $G/H$ with equal geodesic distances $d(x_1, x_2) = d(y_1, y_2)$, there exists an isometry $\iota$ sending one pair to the other, $\iota(x_1) = y_1$, $\iota(x_2) = y_2$.

For the fourth case, the group $G$ itself can be considered as the homogeneous space $G \times G / G_{\mathrm{d}}$, where $G_{\mathrm{d}} = \{(g, g) : g \in G\}$ is the diagonal normal subgroup. For the quotient map $\mathrm{d} : (g_1, g_2) \in G \times G \mapsto g_1 g_2^{-1} \in G$, the kernel is $\ker \mathrm{d} = G_{\mathrm{d}}$ and the image $\operatorname{im} \mathrm{d} = G$ is identified with the quotient space $G \times G / G_{\mathrm{d}}$. Then the map $\mathrm{d}$ induces an action of $G \times G$ on $G$ as $(g_1, g_2) \cdot g = g_1 g g_2^{-1}$. The quasi-regular representation of $G \times G / G_{\mathrm{d}}$ captures the left and right regular representations of $G$ simultaneously, and its decomposition recovers the Plancherel formula of $G$. And in this case, the spherical functions introduced later, correspond to the characters of $G$. The physical relevant example is

$$\text{AdS}_3 / \mathbb{Z} \simeq \text{SL}(2, \mathbb{R}) \simeq \text{SL}(2, \mathbb{R})_{\text{left}} \times \text{SL}(2, \mathbb{R})_{\text{right}} / \text{SL}(2, \mathbb{R})_{\text{diag}}. \tag{C.2}$$

**Quasi-regular representation.** The quasi-regular representation of a homogeneous space $G/H$ is a generalization of the regular representation of $G$. Choosing a $G$-invariant measure $d\mu$ on $G/H$, the Hilbert space $\mathcal{H}_\rho = L^2(G/H)$ equipped with the inner product $(f, g) = \int_{G/H} d\mu \, f^*(x) g(x)$ is a unitary representation of $G$, and the action of $G$ is given by $g \cdot f(x) = f(g^{-1} x)$ for $g \in G$, $f(x) \in \mathcal{H}_\rho$.[8] This is called a quasi-regular representation $\rho$ of $G$ associated with $(G/H, d\mu)$. For a semisimple symmetric space $G/H$ there exists a unique $G$-invariant measure, and $\rho$ is referred as "the" (quasi-)regular representation associated with $G/H$.

Another useful viewpoint of $\rho$ is the induced representation $\operatorname{Ind}_H^G 1$ from the trivial one of $H$ to $G$. For a representation $\pi$ of $H$ on $\mathcal{H}_\pi$, the induced representation $\operatorname{Ind}_H^G \pi$ contains $H$-right-covariant functions from $G$ to $\mathcal{H}_\pi$,

$$\operatorname{Ind}_H^G \pi = \{f : G \to \mathcal{H}_\pi, \, f(gh^{-1}) = \pi(h)f(g), \, \forall h \in H\}, \tag{C.3}$$

and the action of $G$ is given by $(g \cdot f)(x) = f(g^{-1} x)$. Then choosing $\pi$ as the trivial one, $\operatorname{Ind}_H^G \pi$ contains $H$-right-invariant functions on $G$, equivalently, functions on $G/H$. The

normalizable condition of $f(x) \in L^2(G/H)$ is achieved by refining (C.3) to the $L^2$-induction.

**Direct integral.** For noncompact noncommutative Lie groups, the unitary irreps are all infinite-dimensional Hilbert spaces, and it is necessary to extend the direct sum $\bigoplus_{i \in I}$ on an index set $I$ to the direct integral $\int_M^\oplus d\mu$ on a measure space $(M, d\mu)$. The idea of the direct integral is as follows.

In the Fourier analysis of the abelian group $\mathbb{R}$, the regular representation $\rho$ on $L^2(\mathbb{R})$ is given by $\rho(c)f(x) = f(x-c)$ for $c \in \mathbb{R}$. The unitary irrep $\pi_p$ on a one-dimensional space $\mathcal{H}_p$ is given by $\pi_p(c)v = e^{ipc}v$ for $v \in \mathcal{H}_p$ and $c \in \mathbb{R}$. Then $\rho$ is decomposed into the direct integral $L^2(\mathbb{R}) = \int_\mathbb{R}^\oplus \frac{dp}{2\pi}\pi_p$, where $\{\pi_p : p \in \widetilde{\mathbb{R}} \simeq \mathbb{R}\}$ is the set of one-dimensional unitary irreps. At the level of functions, for any $f(x) \in L^2(\mathbb{R})$ we have the pair of Fourier transforms

$$f(x) = \int_\mathbb{R} \frac{dp}{2\pi}\, \widetilde{f}(p)e^{ipx}, \quad \text{and} \quad \widetilde{f}(p) = \int_\mathbb{R} dx\, f(x)e^{-ipx}. \tag{C.4}$$

The Fourier component $f_p(x) := \widetilde{f}(p)e^{ipx}$ associated with $\pi_p$ is not square-integrable, $f_p(x) \notin L^2(\mathbb{R})$. Hence for any $p$, $\pi_p$ is not a subrepresentation of $\rho$. Instead of direct sum, $\{\pi_p\}$ behave like densities of subrepresentations and should be integrated together. Roughly, a direct integral of Hilbert spaces $\{\mathcal{H}_\mu : \mu \in M\}$ with measure $d\mu$ contains square-integrable series of vectors $v = \{v_\mu\}$, $\int_M d\mu\, \|v_\mu\|^2 < \infty$, and the inner product is

$$(v_1, v_2) = \int_M d\mu\, (v_{1,\mu}, v_{2,\mu})_\mu. \tag{C.5}$$

Similar to the scattering and bound states of self-adjoint operators, the measure $d\mu$ may contain both continuous part and discrete part, and only the Hilbert spaces in the discrete part are genuine subspaces of the direct integral. The direct sum $\bigoplus_{i \in I}$ can be regarded as a direct integral on $I$ equipped with the counting measure.

**Positive-definite distributions.** To discuss the positive-definiteness of inner products on Hilbert spaces, positive-definite matrices $v_i^* K_{ij} v_j \geqslant 0, \forall v \in V$ are extended to positive-definite distributions. For simplicity we consider distributions on the real line. If for any test function $f(x)$, the integral is positive,

$$(f, f) = \int_\mathbb{R} dx dy\, f^*(x)K(x-y)f(y) \geqslant 0, \tag{C.6}$$

then the distributional kernel $K$ is called positive-definite. Bochner-Schwartz theorem asserts that the Fourier transform of a positive-definite distribution is a positive distribution, and vice versa.

---

[8]If there is no $G$-invariant measure on $G/H$, the alternative is quasi-invariant measure that transforms covariantly under $G$, $d\mu_{gx} = \Delta(g, x)d\mu_x$. Then the action of $G$ is modified by $g \cdot f(x) = \sqrt{\Delta(g, g^{-1}x)}f(g^{-1}x)$ to compensate the Jacobian factor.

The motivation of Bochner-Schwartz theorem is as follows. If $K$ is a function, the definition (C.6) is equivalent to the discretized one:

$$\sum_{i,j}^{N} v_i^* K(x_i - x_j) v_j \geqslant 0 \tag{C.7}$$

for any number of sample points $x_i$ and $v_i$. By the Fourier transform of $K$, the left side of (C.7) can be rewritten as

$$\text{l.h.s} = \sum_{i,j} v_i^* v_j \int_{\mathbb{R}} \frac{dp}{2\pi} \, e^{ip(x_i - x_j)} \widetilde{K}(p) = \int_{\mathbb{R}} \frac{dp}{2\pi} \left| \sum_i e^{-ipx_i} v_i \right|^2 \widetilde{K}(p) \geqslant 0. \tag{C.8}$$

Then the Fourier transform $\widetilde{K}$ is positive by the arbitraryness of $x_i$ and $v_i$.

## C.2   Method of spherical functions for compact $H$

In this appendix we consider the compact and Riemannian cases of semisimple symmetric spaces, and the stabilizer group $H$ is compact. We first introduce the $H$-fixed vector and $H$-spherical function, then the Poisson and Fourier transforms, and finally the completeness relation and the inversion formula.

**$H$-fixed vector.** When $H$ is compact, we can define the averaging operator by the right action of $H$, $\mathcal{P}_H \cdot f(x) = \int_H dh \, f(xh)$. It is a projector from $L^2(G)$ to $L^2(G/H)$. Hence the set $\widetilde{G}^H$ of unitary irreps in $L^2(G/H)$ is a proper subset of the tempered unitary dual $\widetilde{G}^0$ of $L^2(G)$. To determine $\widetilde{G}^H$ we formally apply the Frobenius reciprocity theorem to the quasi-regular representation $\rho = \text{Ind}_H^G 1$: for any unitary irrep $\pi$ of $G$ on $\mathcal{H}_\pi$,

$$\text{Hom}_G(\text{Ind}_H^G 1, \pi) = \text{Hom}_H(1, \text{Res}_H^G \pi), \tag{C.9}$$

where $\text{Hom}_G(\pi_1, \pi_2)$ denotes the vector space of intertwining operators of $G$ from $\pi_1$ to $\pi_2$. The right side of (C.9) can be rephrased as the vector space of $H$-fixed vectors

$$\mathcal{H}_\pi^H := \{v \in \mathcal{H}_\pi : \pi(h)v = v, \forall h \in H\}. \tag{C.10}$$

Hence the necessary and sufficient condition of an irrep $\pi$ being in $\widetilde{G}^H$ is the existence of non-vanishing $H$-fixed vectors, and the multiplicity is equal to the dimension of $\mathcal{H}_\pi^H$:

$$\widetilde{G}^H = \{\pi \in \widetilde{G} : \dim \mathcal{H}_\pi^H \geqslant 1\}. \tag{C.11}$$

**$H$-spherical function.** Another object closely related to the $H$-fixed vector is the $H$-spherical function[9]. The $H$-left-invariant functions on $G/H$ are defined by the condition $f(h^{-1}x) = f(x)$ for $x \in G/H$, $h \in H$. Equivalently, they are $H$-bi-invariant on $G$, $f(h_1^{-1}gh_2) = f(g)$ for $g \in G$, $h_1, h_2 \in H$. For each $H$-fixed vector $v$, due to $\pi(h)v = v$ there is naturally a $H$-bi-invariant function $\phi_v(g)$ defined by

$$\phi_v(g) := (\pi(g)v, v), \text{ for } g \in G, \tag{C.12}$$

and it reduces to a $H$-left-invariant function $\phi_v(x)$ on $G/H$ for $g = xh$. These functions arising from $H$-fixed vectors are called $H$-spherical functions on $G/H$.

The expression (C.12) can be written in a more symmetric form by introducing the two-variable version of $\phi_v(x)$,

$$\phi_v(g_1, g_2) := (\pi(g_1)v, \pi(g_2)v) = (\pi(g_2^{-1}g_1)v, v) = \phi_v(g_2^{-1}g_1), \text{ for } g_1, g_2 \in G. \tag{C.13}$$

Due to $\pi(h)v = v$ and $g_i = x_i h_i$, $\phi_v(g_1, g_2)$ is a function $\phi_v(x_1, x_2)$ on $G/H \times G/H$. Furthermore, $\phi_v(x_1, x_2)$ is $G$-left-invariant, $\phi_v(x_1, x_2) = \phi_v(gx_1, gx_2)$ for $g \in G$.

The relation between $\phi_v(x_1, x_2)$ and $\phi_v(x)$ from (C.13) is $\phi_v(x_1, x_2) = \phi_v(x_2^{-1}x_1)$. This is inconvenient for calculations since the group multiplication $x_2^{-1}x_1$ is nonlinear. A more practical relation is by the geodesic distance. The $H$-left-invariant functions on $G/H$ depend only on the geodesic distance $d(x, e)$ between $x$ and the origin $e$, and the two-variable version is a function of $d(x_1, x_2)$. They are related by left translations of $G$ sending the pair $(x_1, x_2)$ to $(x, e)$. The physical analog is that under the Poincare symmetry, the two-point functions $\langle \phi(x_1)\phi(x_2) \rangle$ depend only on $(x_1 - x_2)^2$.

For convenience, we list the useful properties of the spherical functions without further explanations:

1. they are the images of $H$-fixed vectors under the Poisson transform introduced later;

2. they are eigenfunctions of $G$-left-invariant differential operators on $G/H$;

3. they are positive-definite and provide a canonical basis of all the $H$-left-invariant functions on $G/H$;

4. they satisfy the following integral equation

$$\phi_v(e) \int_H dh\, \phi_v(g_1 h g_2) = \phi_v(g_1)\phi_v(g_2), \text{ for } g_1, g_2 \in G. \tag{C.14}$$

___
[9]In the literature, for compact $H$, $\phi_v(x)$ is called the $H$-/elementary/zonal spherical function, or spherical function for short. According to the context, the terminology "spherical function" may refer to different generalizations of the $H$-spherical functions.

**Poisson and Fourier transforms.** For each $H$-fixed vector $v$ in the irrep $\mathcal{H}_\pi$ with inner product $(\cdot, \cdot)$, we can construct an intertwining operator $\mathcal{F}_v^\dagger$ from $\mathcal{H}_\pi$ to $L^2(G/H)$:

$$\mathcal{F}_v^\dagger : w \in \mathcal{H}_\pi \mapsto (\pi(x)v, w), \text{ for } x \in G/H. \tag{C.15}$$

The image of $\mathcal{H}_\pi$ in $L^2(G/H)$ should be understood as a component of the direct integral. By the Schur lemma and the irreducibility of $\pi$, $\mathcal{F}_v^\dagger$ is an embedding from $\mathcal{H}_\pi$ into $L^2(G/H)$, also called the Poisson transform. By the definition $(\mathcal{F}_v^\dagger[w], f(x)) = (w, \mathcal{F}_v[f(x)])$, the adjoint $\mathcal{F}_v$ of the Poisson transform $\mathcal{F}_v^\dagger$ is the smearing operator

$$\mathcal{F}_v : f(x) \in L^2(G/H) \mapsto \int_{G/H} dx \, f(x)\pi(x)v, \tag{C.16}$$

which is also called the Fourier transform of $f(x)$. Then the projector $\mathcal{P}_v := \mathcal{F}_v^\dagger \mathcal{F}_v$ extracts the $\mathcal{H}_\pi$-part of $f(x) \in L^2(G/H)$:

$$\mathcal{P}_v : f(x) \mapsto \int_{G/H} dy \, f(y)(\pi(x)v, \pi(y)v) = \int_{G/H} dy \, f(y)\phi_v(y^{-1}x). \tag{C.17}$$

In the last equality, we have used the fact that if $f_1 \in L^2(G/H)$ and $f_2$ is spherical, the convolution on $G/H$ is well-defined,

$$(f_1 * f_2)(x) := \int_{G/H} dy \, f_1(y)f_2(y^{-1}x), \text{ for } f_1, f_2 \in L^2(G/H). \tag{C.18}$$

The Poisson transform maps the $H$-fixed vector $v$ to the $H$-spherical function $\phi_v(x)$, and the integral kernel of (C.17) is the two-variable version $\phi_v(x, y)$. The decomposition (C.17) yearns a completeness relation of the set of projectors $\{\mathcal{P}_v\}$.

**Completeness relation.** For the compact and Riemannian cases of semisimple symmetric spaces, the set of spherical functions $\phi_v(x)$ associated with $H$-fixed vectors provides a complete orthogonal basis of $H$-left-invariant functions on $G/H$. For the Dirac delta distribution $\delta(x)$ on $G/H$, from (C.17) we have $\mathcal{P}_v[\delta(x)] = \phi_v(x)$, and the completeness relation can be written as

$$\delta(x) = \int_{\widetilde{G}^{H,c}} |c^c(v)|^{-2} dv \, \phi_v(x) + \sum_{\widetilde{G}^{H,d}} |c^d(v)|^{-2} \phi_v(x), \tag{C.19}$$

where $\widetilde{G}^{H,c}$ and $\widetilde{G}^{H,d}$ are the continuous and discrete parts of $\widetilde{G}^H$ in the decomposition of $L^2(G/H)$. The density can be read off as

$$|c^c(v)|^2 \delta(u, v) = (v, v)^{-1}(\phi_u, \phi_v) \tag{C.20}$$
$$|c^d(v)|^2 \delta_{u,v} = (v, v)^{-1}(\phi_u, \phi_v) \tag{C.21}$$

where the inner product of spherical functions is that of $L^2(G/H)$. From (C.17) and (C.19) the inversion formula of $f(x) \in L^2(G/H)$ is

$$f(x) = \int_{\widetilde{G}^{H,c}} |c^c(v)|^{-2} dv\, \mathcal{P}_v[f(x)] + \sum_{\widetilde{G}^{H,d}} |c^d(v)|^{-2} \mathcal{P}_v[f(x)]. \qquad (C.22)$$

In the literature, $c(v)$ and $|c(v)|^{-2}$ are called the Harish-Chandra $c$-function and the Plancherel measure of $G/H$ respectively.

## C.3   Noncompact $H$

**$H$-fixed distribution.** When $H$ is noncompact, the averaging projector $\mathcal{P}_H$ is divergent, and normalizable functions in $L^2(G/H)$ are no longer normalizable in $L^2(G)$. In result, the irreducible decomposition of $L^2(G/H)$ is not necessarily related to that of $L^2(G)$, and there can be additional irreps in $\widetilde{G}^H$ that are not present in $\widetilde{G}^0$.

This phenomenon occurs for the real hyperbolic space. There is a discrete part in the regular representation $L^2(\mathrm{SO}(p,q))$ if $pq$ is even, corresponding to the discrete series representations. In contrast, there is a discrete part in $L^2(\mathrm{H}_{p,q})$ for any $q \geqslant 2$, and the corresponding irreps are degenerate principal series representations or their quotients. For the dS case $p = 1$, they are scalar complementary and exceptional series representations.

Nevertheless, the construction of irreps in $\widetilde{G}^H$ from $H$-fixed vectors still holds, but they are incomplete to span the whole space $L^2(G/H)$. To describe the additional irreps in $\widetilde{G}^H$ but not in $\widetilde{G}^0$, the concept of $H$-fixed vectors should extend to $H$-fixed distributions. The matrix element $(\pi(x)v, w)$ still makes sense when $w$ is in a dense subspace $\mathcal{H}_{\pi,0} \subset \mathcal{H}_\pi$ such that $\pi(g)\mathcal{H}_{\pi,0} \subset \mathcal{H}_{\pi,0}$, $\forall g \in G$ and $v$ is a distribution in the dual of $\mathcal{H}_{\pi,0}$. The subspace $\mathcal{H}_{\pi,0}$ is usually chosen as the Garding space containing smooth vectors, and the $H$-fixed distributions are defined thereby. For the semisimple symmetric spaces we considered, it turns out that this generalization is sufficient: the spherical functions associated with $H$-fixed distributions provide a complete orthogonal basis, and are in one-to-one correspondence with the irreps in $\widetilde{G}^H$.

**Rank-one condition and Completeness from Laplacian.** In the preceding discussion the completeness of spherical functions was not fully addressed. This problem is usually more difficult, and one approach is the spectral decomposition of the Laplacian differential operator.

For a Lie group $G$, the Casimirs correspond to bi-invariant differential operators $\{\mathcal{D}_i\}$ on $G$ by the exponential map. They act as scalars on the irreps in the tempered unitary dual $\widetilde{G}^0$, and the joint eigenspaces are identified with these irreps up to multiplicity. The

self-adjointness of $\{\mathcal{D}_i\}$ ensures the completeness of the decomposition of the regular representation. In the same spirit, the $G$-left-invariant differential operators on $G/H$ decompose $L^2(G/H)$ into irreps up to multiplicity. For the semisimple symmetric spaces of rank one, there is only one independent $G$-left-invariant differential operator - the Laplacian operator $\mathcal{L}$, and its spectral decomposition provides a complete classification of irreps in $\widetilde{G}^H$.

## C.4   Example: $S^2 = SO(3)/SO(2)$

In this appendix, as a motivating example we apply the formal discussions to the sphere $S^2 = SO(3)/SO(2)$.

There is another equivalent version of the Poisson and Fourier transforms with respect to the right coset space $H\backslash G$,

$$\mathcal{F}_v : f(x) \in L^2(H\backslash G) \mapsto \int_{H\backslash G} dx\, f(x)\pi^\dagger(x)v, \qquad (C.23)$$

$$\mathcal{F}_v^\dagger : w \in \mathcal{H}_\pi \mapsto (\pi^\dagger(x)v, w), \qquad (C.24)$$

$$\mathcal{P}_v : f(x) \mapsto \int_{H\backslash G} dy\, f(y)\phi_v(y^{-1}x). \qquad (C.25)$$

The spherical function is $\phi_v(x) = (v, \pi(x)v)$, *i.e.* the conjugate of $(\pi(x)v, v)$, and the two-variable version is $\phi_v(x, y) = (\pi(y)v, \pi(x)v) = \phi_v(y^{-1}x)$.

For the abelian group $\mathbb{R}$, the Fourier transform (C.16) and the projector (C.17) are actually the conjugate of the usual ones: $\mathcal{F}_v : f(x) \mapsto \int_{\mathbb{R}} dx\, f(x)e^{ipx}$ and $\mathcal{P}_v : f(x) \mapsto \int_{\mathbb{R}} dx'\, f(x')e^{ip(x'-x)}$, c.f. (C.4). The equivalence of the two conventions is because that the set of plane-waves is closed under conjugation. In this sub-appendix only, to match the Fourier analysis on $\mathbb{R}$ (C.4), we use the convention (C.23).

The SO(2)-fixed vector of integer spin representations $\mathcal{H}_j := \{|j, m\rangle : m = -j, \ldots, j\}$ is $|j, 0\rangle$, while the half-integer spin representations contain no nonvanishing SO(2)-fixed vectors. The matrix element $(w_1, \pi(g)w_2)$ is the Wigner D-function

$$D^j_{m_1,m_2}(\theta, \varphi, \phi) = \langle j, m_1|\pi(g)|j, m_2\rangle, \qquad (C.26)$$

then the Poisson transform (C.24) maps $|j, m\rangle$ to the spherical harmonics

$$\mathcal{F}_j^\dagger : |j, m\rangle \mapsto D^j_{0,m}(\phi, \theta, \varphi) = \sqrt{\frac{4\pi}{2j+1}} Y^j_m(\theta, \varphi). \qquad (C.27)$$

Particularly the SO(2)-fixed vector $|j, 0\rangle$ is mapped to the spherical function $\sqrt{\frac{4\pi}{2j+1}} Y^j_0(\theta, \varphi) = P^j(\cos\theta)$, which is independent of $\varphi$.

The Fourier transform and the projector are

$$\mathcal{F}_j : f(\theta, \varphi) \mapsto \sqrt{\frac{4\pi}{2j+1}} \int_{S^2} d\mu \, f(\theta, \varphi) Y_m^{j*}(\theta, \varphi), \tag{C.28}$$

$$\mathcal{P}_j : f(\theta, \varphi) \mapsto \int_{S^2} d\mu' \, f(\theta', \varphi') P_j(\cos\widetilde{\theta}), \tag{C.29}$$

where $d\mu = \sin\theta d\theta d\varphi$ and $\widetilde{\theta}$ is the angle between the two points $(\theta, \varphi)$ and $(\theta', \varphi')$ on $S^2$, explicitly given by

$$\cos\widetilde{\theta} = \cos\theta\cos\theta' + \sin\theta\sin\theta'\cos(\phi - \phi'). \tag{C.30}$$

In (C.29) we have used the addition formula of spherical harmonics,

$$\sum_{m=-j}^{j} \frac{4\pi}{2j+1} Y_m^{j*}(\theta', \varphi') Y_m^j(\theta, \varphi) = P_j(\cos\widetilde{\theta}). \tag{C.31}$$

The left side corresponds to the two-variable spherical function and the right side is the single-variable one.

The complete set of spherical functions is $\{P^j(\cos\theta) : j = 0, 1, \ldots\}$ and the measure is $|c(j)|^2 = \int_{S^2} d\mu \, (P^j(\cos\theta))^2 = \frac{4\pi}{2j+1}$. Now the inversion formula (C.22) can be rewritten as

$$f(\theta, \varphi) = \sum_{j=0}^{\infty} \sum_{m=-j}^{j} f_{j,m} Y_m^j(\theta, \varphi), \quad \text{where} \quad f_{j,m} = \int_{S^2} d\mu \, f(\theta, \varphi) Y_m^{j*}(\theta, \varphi). \tag{C.32}$$

# D   Split representation on EAdS/dS from the harmonic analysis

In this appendix, we derive the split representations on the EAdS$_{d+1}$ and dS$_{d+1}$ from the harmonic analysis. From the formal discussions in Appendix C, the procedure of deriving the split representation contains the following steps:

1. exhaust the unitary irreps containing $H$-fixed vectors/distributions;

2. determine the $H$-fixed vectors and the spherical functions;

3. derive the inversion formula and the split representation.

In the following, the volume of sphere is $\mathrm{vol}\,S^n = \frac{2^n \pi^{n/2} \Gamma(n/2)}{\Gamma(n)}$; the symbol $d = p + q - 2$ is referred to the dimension of CFT, and $\widetilde{\Delta} = d - \Delta$ is the shadow dimension of $\Delta$.

## D.1 Embedding formalism of the real hyperbolic spaces

The real hyperbolic space[10] $H_{p,q} = SO(p,q)/SO(p,q-1)$, $p \geqslant 1$ is a generalization of the usual hyperbolic space $H_{d+1}$ and can be embedded as the hypersurface

$$X \cdot X = -X_1^2 - \cdots - X_p^2 + X_{p+1}^2 + \cdots + X_{p+q}^2 = 1, \tag{D.1}$$

in the ambient space $X \in \mathbb{R}^{p,q}$. This includes AdS, Euclidean AdS, dS and Kleinian dS spaces:

$$\text{AdS}_{d+1}/\mathbb{Z} = H_{d,2}, \quad \text{EAdS}_{d+1} = H_{d+1,1}/\mathbb{Z}_2, \quad \text{dS}_{d+1} = H_{1,d+1}, \quad \text{KdS}_{d+1} = H_{2,d}, \tag{D.2}$$

where the Kleinian dS space is referred as the tachyon mass shell in Kleinian spacetime, just like the dS space can be identified with the tachyon mass shell in Lorentzian spacetime.

The lightcone $P \cdot P = 0$, $P \neq 0$, $P \in \mathbb{R}^{p,q}$ is denoted as $\text{LC}_{p,q}$. Then the asymptotic boundary of $H_{p,q}$ is the projective lightcone $\text{PC}_{p,q} := \text{LC}_{p,q}/\mathbb{R}^+ \simeq S^{p-1} \times S^{q-1}$ by the quotient $P \sim \lambda P, \lambda > 0$. For the Euclidean AdS case, $H_{d+1,1}$, $\text{LC}_{d+1,1}$ and $\text{PC}_{d+1,1}$ contain two connected components. We choose the upper ones $x_{d+2} > 0$ and denote them as $H_{d+1} = \text{EAdS}_{d+1}$, $\text{LC}_{d+1}$ and $\text{PC}_{d+1} \simeq S^d$ respectively.

The coordinate systems are summarized in table 3 and are explained later[11]. The hat notation denotes the spherical coordinates $\widehat{X} \in S^d \subset \mathbb{R}^{d+1}$. Similarly the check notation denotes $\check{X} \in H_d \subset \mathbb{R}^{d,1}$ and the tilde notation denotes $\widetilde{X} \in \text{dS}_d \subset \mathbb{R}^{d,1}$.

| name | $H_{d+1}$ | $\text{dS}_{d+1}$ | $\text{PC}_{d+1}$ |
|---|---|---|---|
| global | $(\cosh t, \widehat{X} \sinh t)$ | $(\sinh t, \widehat{X} \cosh t)$ | $(1, \widehat{P})$ |
| hyperbolic | $(\check{X} \cosh t, \sinh t)$ | $(\check{X} \sinh t, \cosh t)$ | $(\check{P}, \pm 1)$ |
| Poincare | $\left(\frac{1+z^2+x^2}{2z}, \frac{x}{z}, \frac{1-z^2-x^2}{2z}\right)$ | $\left(\frac{1-z^2+x^2}{-2z}, -\frac{x}{z}, \frac{1+z^2-x^2}{-2z}\right)$ | $\left(\frac{1+x^2}{2}, x, \frac{1-x^2}{2}\right)$ |

Table 3: Coordinate systems.

**dS.** The dS spacetime $\text{dS}_{d+1}$ is embedded into $\mathbb{R}^{1,d+1}$ as the one-sheeted hyperboloid

$$-(X^0)^2 + (X^1)^2 + \cdots + (X^{d+1})^2 = 1. \tag{D.3}$$

---

[10]For $p = 0$ it's the $(q-1)$-dimensional sphere. The term "real" is to distinguish from the complex and quaternion hyperbolic spaces: $SU(p,q)/S(U(1) \times U(p,q-1))$, $Sp(p,q)/(Sp(p,q-1) \times Sp(1))$.

[11]In celestial CFT, the common convention of the Poincare coordinates on the lightcone is $(1+x^2, 2x, 1-x^2)$.

The global coordinates of $\mathrm{dS}_{d+1}$ are $X = (\sinh t, \widehat{X}\cosh t)$ for $t \in \mathbb{R}$, $\widehat{X} \in \mathrm{S}^d$, The Poincare coordinates are $(\frac{1-z^2+x^2}{-2z}, -\frac{x}{z}, \frac{1+z^2-x^2}{-2z})$ for $z \neq 0$, $x \in \mathbb{R}^d$. The upper (lower) Poincare patch corresponds to $z < 0$ ($z > 0$), with the future (past) infinity as the limit $z \to 0^-$ ($z \to 0^+$).

We also need the hyperbolic coordinates that divide $\mathrm{dS}_{d+1}$ into three charts, see figure 2,

$$\mathrm{I} \quad (\check{X}\sqrt{T^2-1}, T), \quad \text{for } T = X \cdot X_0 < -1, \tag{D.4}$$

$$\mathrm{II} \quad (\widetilde{X}\sqrt{1-T^2}, T), \quad \text{for } -1 < T < 1, \tag{D.5}$$

$$\mathrm{III} \quad (\check{X}\sqrt{T^2-1}, T), \quad \text{for } T > 1, \tag{D.6}$$

where $X_0 = (0, \ldots, 1)$ and the "radial" coordinate $T$ can be further parametrized into $T = \pm\cosh t = \cos\theta$ in different charts. The geometric meaning is the geodesic flow starting from $X_0$: the points in the third (second) chart can be connected by a single timelike (spacelike) geodesic from $X_0$, while the points in the first chart cannot be connected by any single geodesic starting from $X_0$.

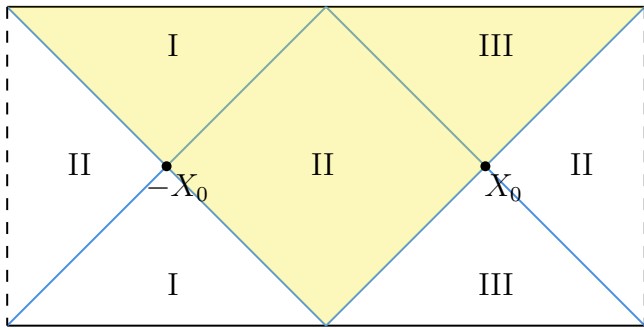

Figure 2: Penrose diagram of hyperbolic and Poincare charts of $\mathrm{dS}_{d+1}$. The upper Poincare chart corresponds to the yellow triangle. The three hyperbolic charts correspond to the region I, II and III. Each point in this diagram is half of $\mathrm{S}^{d-1}$, and the two dashed vertical lines should be identified, which doubles the usual dS Penrose diagram. $-X_0$ is the antipodal point of $X_0$.

**EAdS.** To relate the discussion of $\mathrm{dS}_{d+1}$, we choose the ambient spacetime of $\mathrm{H}_{d+1}$ as $\mathbb{R}^{1,d+1}$ with the most-plus signature, which is contrary to (D.1). The Euclidean AdS is embedded as the upper component of the two-sheeted hyperboloid

$$-(X^0)^2 + (X^1)^2 + \cdots + (X^{d+1})^2 = -1, \; X^0 > 0. \tag{D.7}$$

The global coordinates of $\mathrm{H}_{d+1}$ are $X = (\cosh t, \widehat{X}\sinh t)$ for $t > 0$, $\widehat{X} \in \mathrm{S}^d$. The Poincare coordinates are $(\frac{1+z^2+x^2}{2z}, \frac{x}{z}, \frac{1-z^2-x^2}{2z})$ for $z > 0$, $x \in \mathbb{R}^d$. The hyperbolic coordinates are $(\check{X}\cosh t, \sinh t)$, for $t \in \mathbb{R}$, $\check{X} \in \mathrm{H}_d$.

**Lightcone.** The lightcone $\mathrm{LC}_{1,d+1}$ and the projective one $\mathrm{PC}_{1,d+1}$ satisfy

$$-(P^0)^2 + (P^1)^2 + \cdots + (P^{d+1})^2 = 0, \tag{D.8}$$

and they contain two disconnected components, corresponding to the future and past infinities of the dS spacetime. The upper components are denoted as $\mathrm{LC}_{d+1}$ and $\mathrm{PC}_{d+1}$. The lower component can be accessed by the antipodal map $P \to -P$ for the point $P \in \mathrm{PC}_{d+1}$. In other words, we treat $\mathrm{PC}_{1,d+1}$ as the trivial double cover of $\mathrm{PC}_{d+1}$.

We choose three charts of the projective lightcone $\mathrm{PC}_{d+1}$: $\mathrm{S}^d$ ($P^0 = 1$), $\mathbb{R}^d$ ($P^0 + P^{d+1} = 1$) and $\mathrm{H}_d \amalg \mathrm{H}_d$ ($P^{d+1} = \pm 1$). The coordinates of the first chart are $P = (1, \widehat{P}) = (1, \widehat{x} \sin\theta, \cos\theta)$ for $\widehat{P} \in \mathrm{S}^d$, $\widehat{x} \in \mathrm{S}^{d-1}$, $0 \leqslant \theta \leqslant \pi$. The coordinates of the second are $P = (\frac{1+x^2}{2}, x, \frac{1-x^2}{2})$ for $x \in \mathbb{R}^d$. The last section contains two charts, each of which covers half of $\mathrm{PC}_{d+1}$, and the coordinates are $P = (\widecheck{P}, \pm 1) = (\cosh t, \widehat{x} \sinh t, \pm 1)$ for $P \in \mathrm{H}_d$, $\widehat{x} \in \mathrm{S}^{d-1}$, $t > 0$. The three coordinate systems are related by the conformal compactification $x = \widehat{x} \tan\frac{\theta}{2} = \widehat{x} \tanh^{\pm 1} \frac{t}{2}$.

The integrals on $\mathrm{PC}_{d+1}$ can be lifted onto the $\mathrm{LC}_{d+1}$ in a gauge invariant way, called the conformal integrals in [55]:

$$\int_{\mathrm{LC}_{d+1}} d'P\, f(P) = \int_{\mathrm{S}^d} d\widehat{P}\, f_{\mathrm{S}}(\widehat{P}) = \int_{\mathbb{R}^d} d^d x\, f_{\mathrm{F}}(x) = \int_{\mathrm{H}_d} d\widecheck{P} \left( f_{\mathrm{H}}(\widecheck{P}, 1) + f_{\mathrm{H}}(\widecheck{P}, -1) \right) \tag{D.9}$$

Formally $d'P = \frac{dP}{\mathrm{vol}\,\mathbb{R}}$ is regularized by the volume of the scaling symmetry on $\mathrm{LC}_{d+1}$, and the integrand $f(P)$ must be a homogeneous function with mass dimension $[f] = d$.

**Radial part of Laplacian.** In the preceding we also need the radial part of the Laplacian operator in EAdS/dS spacetime. In the global coordinates of EAdS spacetime, the Laplacian eigenvalue equation is

$$\frac{\partial^2 f}{\partial t^2} + d \coth t \frac{\partial f}{\partial t} + \frac{1}{\sinh^2 t} \Box_{\mathrm{S}^d} f = \Delta(\Delta - d) f. \tag{D.10}$$

The little group at $X_0 = (1, 0, \dots)$ is $H_{X_0} \simeq \mathrm{SO}(d+1)$, and if $f$ is a $H_{X_0}$-left-invariant function on $\mathrm{EAdS}_{d+1}$, the radial part gives the Sturm–Liouville equation

$$\sinh^{-d} t \frac{d}{dt} \left[ \sinh^d t \frac{df}{dt} \right] = \Delta(\Delta - d) f, \tag{D.11}$$

and with $T = -X \cdot X_0 = \cosh t \in (1, \infty)$, the equation is transformed to

$$\left(T^2 - 1\right) f''(T) + (d+1) T f'(T) = \Delta(\Delta - d) f(T). \tag{D.12}$$

In the hyperbolic coordinates of dS spacetime (D.4), the Laplacian eigenvalue equations are

$$-\frac{\partial^2 f}{\partial t^2} - d \coth t \frac{\partial f}{\partial t} + \frac{1}{\sinh^2 t} \Box_{\mathrm{H}_d} f = \Delta(d - \Delta) f, \quad \text{in 1st/3rd chart,} \tag{D.13}$$

$$\frac{\partial^2 f}{\partial \theta^2} + d \cot \theta \frac{\partial f}{\partial \theta} + \frac{1}{\sin^2 \theta} \Box_{\mathrm{dS}_d} f = \Delta(d - \Delta)f, \ \ \text{in 2nd chart.} \tag{D.14}$$

The little group at $X_0 = (0, \ldots, 1)$ is $H_{X_0} \simeq \mathrm{SO}(1, d)$, and interestingly, if $f$ is a $H_{X_0}$-left-invariant function on $\mathrm{dS}_{d+1}$, the radial part is exactly the same as (D.12) with $T = X \cdot X_0 \in \mathbb{R}$. The only difference is that the $T$-coordinate covers $(-\infty, -1), (-1, 1)$ and $(1, \infty)$, corresponding to the three hyperbolic charts (D.4) respectively.

## D.2 Unitary irreducible representations of the Euclidean conformal group

In this section we consider **scalar** unitary irreps of $\mathrm{SO}(d+1, 1)$, which can be cast into three classes[12]: the principal, complementary, and exceptional series, see e.g. [48, 49, 56–58]. The decomposition of quasi-regular representation on the Euclidean AdS space $\mathrm{H}_{d+1} = \mathrm{SO}(d+1, 1)/\mathrm{SO}(d+1)$ includes only continuous part, and the corresponding irreps are the principal series representations. The decomposition on the dS spacetime $\mathrm{dS}_{d+1} = \mathrm{SO}(1, 1+d)/\mathrm{SO}(1, d)$ includes both continuous and discrete parts. The continuous part is still the principal series, and the discrete part can be identified with the complementary and exceptional series.

**Principal series.** The scalar unitary principal series representation $\mathcal{E}_{\Delta = \frac{d}{2} + is, J = 0}$, $s \in \mathbb{R}$ is defined on the space of homogeneous functions $f(\lambda P) = \lambda^{-\Delta} f(P)$, $P \in \mathrm{LC}_{d+1}$, $\lambda > 0$ by the action

$$g \cdot f(P) = f(g^{-1}P), \ \ \text{for} \ g \in \mathrm{SO}(d+1, 1). \tag{D.15}$$

Equivalently, $f$ transforms as a fictitious primary operator $\mathcal{O}(P)$ with complex weight. The homogeneous functions on the lightcone $\mathrm{LC}_{d+1}$ are determined by the values on the projective lightcone $\mathrm{PC}_{d+1}$. Choosing different sections we get the representatives $f_\mathrm{S}$ on $\mathrm{S}^d$ and $f_\mathrm{F}$ on $\mathbb{R}^d$ respectively, and they are related by the Weyl transform $f_\mathrm{S} = (\frac{1+x^2}{2})^\Delta f_\mathrm{F}$. The subscripts labeling sections will be omitted in the following.

For $f_1, f_2 \in \mathcal{E}_\Delta$, the mass dimension of $f_1^*(P) f_2(P)$ is $d$, hence the inner product

$$(f_1, f_2) = \int_{\mathrm{LC}_{d+1}} d'P \, f_1^*(P) f_2(P) \tag{D.16}$$

is invariant under the action (D.15), which justifies the unitarity of $\mathcal{E}_\Delta$.

---

[12]The discrete series representations appear in the discrete part of the regular representation, and are not directly related to the quasi-regular representations. For $d = 1, 3$ they are related to the exceptional series of symmetric traceless tensors [49], while for higher $d$ they are much harder to construct, see e.g. the brief note [59] and the references therein.

When $\Delta \neq \frac{d}{2} + is$ leaves off the unitary principal series, the representations are called non-unitary principal series. They are reducible for certain $\Delta$, and the inner product (D.16) is not invariant under the group action due to $\Delta + \Delta^* \neq d$.

**Shadow transform.** The shadow transform is defined by

$$\mathcal{S}: f(P) \in \mathcal{E}_\Delta \mapsto c_{\mathrm{sd}}(\Delta) \int_{\mathrm{LC}_{d+1}} d'P' \, (-P \cdot P')^{-\widetilde{\Delta}} f(P') \in \mathcal{E}_{\widetilde{\Delta}}, \tag{D.17}$$

and the double shadow transform is proportional to the identity map[13],

$$\mathcal{S}^2 = \frac{(2\pi)^d \Gamma(\Delta - \frac{d}{2}) \Gamma(\widetilde{\Delta} - \frac{d}{2})}{\Gamma(\Delta)\Gamma(\widetilde{\Delta})} c_{\mathrm{sd}}(\Delta) c_{\mathrm{sd}}(\widetilde{\Delta}) \, \mathrm{id} \,. \tag{D.18}$$

The integral kernel $(-P \cdot P')^{-\widetilde{\Delta}}$ should be understood as a tempered distribution and has simple poles at $\Delta = \frac{d}{2} - N$, $N \in \mathbb{N}$, see Appendix B. To ensure the shadow transform free from divergence, the adjustable prefactor $c_{\mathrm{sd}}$ must contain $\frac{1}{\Gamma(\Delta - \frac{d}{2})}$. Then the principal/complementary series representation $\mathcal{E}_\Delta$ and its shadow $\mathcal{E}_{\widetilde{\Delta}}$ are isomorphic by $\mathcal{S}$.

**Complementary series.** When $\Delta \in \mathbb{R}$, the homogeneity degrees of $f^*(P)$ and $f(P)$ are equal, and the problem of inner product can be fixed by introducing a shadow transform

$$(f_1, f_2) = c_{\mathrm{cp}}(\Delta) \int_{\mathrm{LC}_{d+1}} d'P \int_{\mathrm{LC}_{d+1}} d'P' \, (-P \cdot P')^{-\widetilde{\Delta}} f_1^*(P) f_2(P'). \tag{D.19}$$

The positive-definiteness condition requires that $\Delta \in (0, d)$. The group action is the same as (D.15). This type of unitary irreps is called the complementary series.

**Exceptional series.** For $\Delta = -N$ or $\Delta = d + N$, $N \in \mathbb{N}$, the double shadow transform vanishes due to the factor $\frac{1}{\Gamma(\Delta)\Gamma(\widetilde{\Delta})}$. This implies the shadow transform has a non-trivial kernel and coimage, and the representation $\mathcal{E}_\Delta$ is reducible. The image $\mathcal{V}_{d+N} := \mathcal{E}_{-N} / \ker \mathcal{S} \subset \mathcal{E}_{d+N}$ is irreducible for $d \geqslant 2$. The inner product of $\mathcal{V}_{d+N}$ turns out to be the residue of (D.19),

$$(f_1, f_2) = \mathrm{Res}_{\Delta=-N} \, c_{\mathrm{ep}}(\Delta) \int_{\mathrm{LC}_{d+1}} d'P \int_{\mathrm{LC}_{d+1}} d'P' \, (-P \cdot P')^{-\widetilde{\Delta}} f_1^*(P) f_2(P'). \tag{D.20}$$

The group action is the same as (D.15). The resulting unitary irrep $\mathcal{V}_{d+N}$ is called the scalar exceptional series.

For $d = 1$, the scalar exceptional series split into the direct sum of holomorphic and antiholomorphic discrete series: $\mathcal{V}_{N+1} = \mathcal{D}_{N+1} \oplus \overline{\mathcal{D}}_{N+1}$. For $d = 2$, the unitary irreps are labeled by $(h, \overline{h})$ with $\Delta = h + \overline{h}$, $J = |h - \overline{h}| \in \frac{1}{2}\mathbb{N}$[14]. By the following isomorphisms

$$\mathcal{E}_{(-N_1, -N_2)} / \mathcal{E}'_{(-N_1, -N_2)} \simeq \mathcal{E}_{(1+N_1, -N_2)} \simeq \mathcal{E}_{(-N_1, 1+N_2)}, \ N_1, N_2 \in \frac{1}{2}\mathbb{N}, \tag{D.21}$$

---

[13]Notice that the coefficient of id is different from (3.16) in [51] due to $-P_1 \cdot P_2 = \frac{1}{2}|x_{12}|^2$.

the scalar exceptional series $\mathcal{V}_{N+2}$ are isomorphic to another spinning principal series with $\Delta = 1, J = N + 1$.

## D.3 Harmonic analysis on EAdS

In this subsection we rederive the split representation on the Euclidean AdS space [10, 12–14] from the perspective of harmonic analysis. The reader can consider in mind the following intuitive comparison to $S^2 = SO(3)/SO(2)$: the analogs of the angular momentum $j$, the magnetic number $m$ and the spherical coordinates $(\theta, \varphi)$ are the weight $(\Delta, J = 0)$, the boundary coordinates $P \in LC_{d+1}$ and the bulk coordinates $X \in H_{d+1}$ respectively.

**Spherical function.** Following the discussion in Appendix C, for $H = SO(d + 1)$ we determine the $H$-fixed vectors in the principal series $\mathcal{E}_\Delta$ and the spherical functions in $L^2(H_{d+1})$. Choosing a reference point $X_0 = (1, 0, \ldots, 0)$ as the origin of $dS_{d+1}$, the stabilizer subgroup at $X_0$ is $H \simeq SO(d + 1)$. Then the only normalized $H$-invariant function on $S^d$ is $f_\Delta(1, \widehat{P}) = 1$, and by the homogeneity the unique $H$-fixed vector is

$$f_\Delta(P) = (P^0)^{-\Delta}, \text{ for } P \in LC_{d+1}. \tag{D.22}$$

To obtain the spherical functions, we need an explicit expression of the matrix element $\pi(X)f_\Delta(P)$, $X \in H_{d+1}$. Due to the $SO(d + 1)$-invariance, we can choose a representative of $P$ such that $X$ and $P$ are in the same plane through the origin: $P = (z, 0, \ldots, 0, z)$ and $X = (\cosh t, 0, \ldots, 0, \sinh t)$. Then by picking a boost $B = \left( \begin{smallmatrix} \cosh t & \sinh t \\ \sinh t & \cosh t \end{smallmatrix} \right)$ such that $X = B \cdot X_0$, the matrix element is $\pi(X)f_\Delta(P) = f_\Delta(B^{-1} \cdot P) = (z \cosh t - z \sinh t)^{-\Delta} = (-X \cdot P)^{-\Delta}$.

The matrix element $\pi(X)f_\Delta(P) = (-X \cdot P)^{-\Delta}$ is exactly the bulk-to-boundary propagator. This can be understood that $\pi(X)f_\Delta(P)$ should transform covariantly and simultaneously under the actions of the bulk isometric and the boundary conformal transformations. And $(-X \cdot P)^{-\Delta}$ is the only covariant quantity that can be built from bulk and boundary points and has mass dimension $\Delta$. This fixes $\pi(X)f_\Delta(P)$ to be the bulk-to-boundary propagator uniquely.

For the principal series $\mathcal{E}_{\frac{d}{2}+is}$, $s \in \mathbb{R}$, the two-point spherical functions are derived in Appendix D.6.1,

$$\phi_\Delta(X_1, X_2) = \int_{LC_{d+1}} d'P \, (-X_1 \cdot P)^{-\widetilde{\Delta}}(-X_2 \cdot P)^{-\Delta} = \text{vol} \, S^d \, {}_2F_1(\Delta, \widetilde{\Delta}, \frac{d+1}{2}, \frac{1-T}{2}), \tag{D.23}$$

where $T = -X_1 \cdot X_2$. The spherical functions satisfy $\phi_\Delta(T) = \phi_{\widetilde{\Delta}}(T)$, which reflects the shadow symmetry $\mathcal{E}_\Delta \simeq \mathcal{E}_{\widetilde{\Delta}}$.

---

[14] The physics notation is related to [56] by $h = \frac{1-n_1}{2}, \bar{h} = \frac{1-n_2}{2}$.

**Completeness from Laplacian.** The spherical function (D.23) satisfies the eigenequation of the radial Laplacian (D.12), even for non-unitary principal series $\Delta \in \mathbb{C}$. Using the Sturm-Liouville theory, in Appendix D.5.1 we show that the spherical functions of principal series provide a complete basis for the radial Laplacian. The orthogonality and completeness relations of are

$$2\pi N_{\mathrm{H}}(\Delta_1)\delta(s_1 - s_2) = \int_0^\infty dt \, \sinh^d t \, \phi_{\Delta_1}^*(t)\phi_{\Delta_2}(t), \tag{D.24}$$

$$\sinh^{-d} t \, \delta(t_1 - t_2) = \int_0^\infty \frac{ds}{2\pi N_{\mathrm{H}}(\Delta)} \, \phi_\Delta^*(t_1)\phi_\Delta(t_2), \tag{D.25}$$

where $\Delta_i = \frac{d}{2} + is_i$, $s_i > 0$, and the density $N_{\mathrm{H}}(\Delta)$ will be derived later.

The integration over $s$ can be rewritten as $\frac{1}{2\pi i} \int_\Gamma \frac{d\Delta}{N_{\mathrm{H}}(\Delta)}$, where the contour is half of the principal series from $\Delta = \frac{d}{2}$ to $\Delta = \frac{d}{2} + i\infty$. Then for normalizable functions we have the decomposition and inversion formulas

$$\widetilde{f}(\Delta) = \int_0^\infty dt \, \sinh^d t \, \phi_\Delta^*(t)f(t), \;\; \text{and} \;\; f(t) = \frac{1}{2\pi i} \int_\Gamma \frac{d\Delta}{N_{\mathrm{H}}(\Delta)} \, \widetilde{f}(\Delta)\phi_\Delta(t). \tag{D.26}$$

They can be interpreted geometrically on the EAdS space $\mathrm{H}_{d+1}$. The Hilbert space with respect to the radial Laplacian contains normalizable $SO(d+1)$-invariant functions on $\mathrm{H}_{d+1}$. The spectral decomposition of the radial Laplacian ensures that the $SO(d+1)$-invariant functions can be decomposed and inverted by the spherical functions, which are called the spherical transforms in the theory of Gelfand pairs.

**Plancherel measure.** The density $N_{\mathrm{H}}(\Delta)$ can be read off by the following localization technique. The integrand $I(t, \Delta_1, \Delta_2)$ of the right side of (D.24) has the asymptotic form as $t \to \infty$,

$$I(t, \Delta_1, \Delta_2) \sim e^{-i(s_1-s_2)t}c_0 c^*(\Delta_1)c(\Delta_2) + e^{i(s_1-s_2)t}c_0 c^*(\widetilde{\Delta}_1)c(\widetilde{\Delta}_2) + \text{terms with } e^{\pm i(s_1+s_2)t}. \tag{D.27}$$

where $c(\Delta)$ is the leading coefficient of the spherical function (D.70) and the factor $c_0 = 2^{-d}$ is the leading coefficient of the measure $\sinh^d t$.

Using the substitution $t \to \lambda t$ and taking the limit $\lambda \to \infty$, the terms with $e^{\pm i(s_1+s_2)t}$ do not contribute due to $s_1, s_2 > 0$, and the first two terms combine into $\int_{\mathbb{R}} dt \, e^{-i(s_1-s_2)t} = 2\pi\delta(s_1 - s_2)$. Hence the measure of the inversion formula (D.26) is

$$N_{\mathrm{H}}(\Delta) = c_0|c(\Delta)|^2 = c_0 c(\Delta)c(\widetilde{\Delta}) = \frac{(2\pi)^d \Gamma(\Delta - \frac{d}{2})\Gamma(\widetilde{\Delta} - \frac{d}{2})}{\Gamma(\Delta)\Gamma(\widetilde{\Delta})}. \tag{D.28}$$

In history [60, 61], Harish-Chandra was the first to establish the connection between the Plancherel measure and the asymptotic behaviour of the spherical functions, and for this

reason the leading coefficient is called the Harish-Chandra $c$-function. In the EAdS case, the Plancherel measure of $\mathrm{EAdS}_{d+1}$ coincides with the prefactor of double shadow transform. This fact is not accidental and holds for other semisimple groups, see [62, 63].

**Fourier transform and inversion formula.** The Poisson transform (C.15), the Fourier transform (C.16) and the projector (C.17) on $\mathrm{H}_{d+1}$ are

$$\mathcal{F}_\Delta^\dagger : f(P) \mapsto \int_{\mathrm{LC}_{d+1}} d'P \, (-X \cdot P)^{-\widetilde{\Delta}} f(P) \in L^2(\mathrm{H}_{d+1}), \tag{D.29}$$

$$\mathcal{F}_\Delta : f(X) \mapsto \int_{\mathrm{H}_{d+1}} dX \, (-X \cdot P)^{-\Delta} f(X) \in \mathcal{E}_\Delta, \tag{D.30}$$

$$\mathcal{P}_\Delta : f(X) \mapsto \int_{\mathrm{H}_{d+1}} dX' \, \phi_\Delta(X, X') f(X'), \tag{D.31}$$

where the two-variable spherical function is

$$\phi_\Delta(X, X') = \int_{\mathrm{LC}_{d+1}} d'P \, (-X \cdot P)^{-\widetilde{\Delta}} (-X' \cdot P)^{-\Delta}. \tag{D.32}$$

From (C.22), the inversion formula reads as

$$f(X) = \frac{1}{2\pi i} \int_\Gamma \frac{d\Delta}{N_{\mathrm{H}}(\Delta)} \, \mathcal{P}_\Delta[f(X)], \tag{D.33}$$

where the contour $\Gamma$ is half of the principal series.

For the Dirac delta distribution we have the resolution of identity, *i.e.* the split representation on the Euclidean AdS space

$$\delta(X, X') = \frac{1}{2\pi i} \int_\Gamma \frac{d\Delta}{\mu(\Delta)} \int_{\mathrm{LC}_{d+1}} d'P \, (-2X \cdot P)^{-\widetilde{\Delta}} (-2X' \cdot P)^{-\Delta}, \tag{D.34}$$

where for later convenience we have introduced the factor

$$\mu(\Delta) = 2^{-d} N_{\mathrm{H}}(\Delta) = \frac{\pi^d \Gamma(\Delta - \frac{d}{2}) \Gamma(\widetilde{\Delta} - \frac{d}{2})}{\Gamma(\Delta) \Gamma(\widetilde{\Delta})}. \tag{D.35}$$

## D.4   Harmonic analysis on dS

In this subsection we provide a derivation of the harmonic analysis on $\mathrm{dS}_{d+1}$, $d \geqslant 2$. The main difference from the EAdS case is that the bulk-to-boundary covariant quantity $X \cdot P$ is indefinite in the dS spacetime, and this allows the existence of two independent spherical functions associated with the same unitary irrep $\mathcal{E}_\Delta$, $\Delta = \frac{d}{2} + is$, $s \in \mathbb{R}$. For $d = 1$, the discussion is slightly more complicated than that of $d \geqslant 2$ since the exceptional series split

into two discrete series, but the resulting Fourier transform and inversion formula share the same forms with $d \geqslant 2$.

**Spherical function.** The derivation of spherical functions is almost the same as that on EAdS space. We choose a reference point $X_0 = (0, \ldots, 0, 1) \in \mathrm{dS}_{d+1}$ and $H \simeq \mathrm{SO}(d, 1)$ is the stabilizer subgroup of $X_0$. In the EAdS case, the lightcone is foliated into orbits of the $\mathrm{SO}(d+1)$-action, $P^0 = \text{const.}$, and homogeneous functions are completely determined by values on one of the orbits $P^0 = 1$. While in the dS case, the $\mathrm{SO}(d, 1)$-orbits are $P^{d+1} = \text{const.}$, and the homogeneous functions are determined by values on the two orbits $P^{d+1} = \pm 1$. Hence there are two $H$-fixed homogeneous functions on $\mathrm{LC}_{d+1}$:

$$f_{\Delta, \epsilon}(P) = |P^{d+1}|^{-\Delta, \epsilon} = |P^{d+1}|^{-\Delta} \operatorname{sgn}^{\epsilon}(P^{d+1}), \quad \text{for} \ \ P \in \mathrm{LC}_{d+1}. \tag{D.36}$$

where the superscript notation $\epsilon = 0, 1$ follows from Appendix B and labels the parity of $f_{\Delta, \epsilon}(P)$ under $P \to -P$. The matrix element $\pi(X) f_{\Delta}(P)$, $X \in \mathrm{dS}_{d+1}$ can be deduced similarly: $\pi(X) f_{\Delta, \epsilon}(P) = |X \cdot P|^{-\Delta, \epsilon}$.

The functions $f_{\Delta, \epsilon}(P)$ have singularities along the hypersurface $P^{d+1} = 0$, and are not $H$-fixed vectors but $H$-fixed distributions associated with $\mathcal{E}_{\Delta}$. For the principal series $\mathcal{E}_{\frac{d}{2}+is}$, $s \in \mathbb{R}$, The corresponding spherical functions are

$$\phi_{\Delta, \epsilon}(T) = \phi_{\Delta, \epsilon}(X_1, X_2) = \int_{\mathrm{LC}_{d+1}} d'P \, |X_1 \cdot P|^{-\widetilde{\Delta}, \epsilon} |X_2 \cdot P|^{-\Delta, \epsilon}, \tag{D.37}$$

where $T = X_1 \cdot X_2$, and the derivations can be found in Appendix D.6.2. They respect the parity symmetry and the shadow symmetry,

$$\phi_{\Delta, \epsilon}(T) = (-1)^{\epsilon} \phi_{\Delta, \epsilon}(-T) = \phi_{\widetilde{\Delta}, \epsilon}(T). \tag{D.38}$$

In the third hyperbolic chart $T \in (1, \infty)$, the spherical functions are

$$\phi_{\Delta, \epsilon}(T) = 2^{\frac{d+1}{2}} \pi^{\frac{d-1}{2}} \Gamma(1-\Delta) \left[ \Gamma(-d+1+\Delta) \left( (-1)^{\epsilon} \sin \frac{d\pi}{2} + \sin \frac{(d-2\Delta)\pi}{2} \right) \phi_1(T) + 2\phi_2(T) \right], \tag{D.39}$$

and in the second hyperbolic chart $T \in (-1, 1)$, they are

$$\phi_{\Delta, \epsilon}(T) = 2^{\frac{d+1}{2}} \pi^{\frac{d-1}{2}} \Gamma(1-\Delta) \Gamma(-d+\Delta+1) \left[ \phi_1(T) \left( (-1)^{\epsilon} - \cos \pi(d-\Delta) \right) + \phi_2(T) \frac{2 \sin \pi(d-\Delta)}{\pi} \right], \tag{D.40}$$

where $\phi_{1,2}$-s are proportional to the Legendre functions, see (D.60), (D.61), (D.62) and (D.63), and we have set $a_1 = 1, a_2 = \frac{e^{-\frac{1}{2} i \pi(d-1)}}{\Gamma(d-\Delta)}, a_1' = 1, a_2' = 1$ therein.

As explained in Appendix B, the $H$-fixed distributions $f_{\Delta, \epsilon}(P)$ are meromorphic functions of $\Delta$ with simple poles at positive integers, and the regularized ones $\frac{1}{\Gamma(\frac{-\Delta+\epsilon+1}{2})} f_{\Delta, \epsilon}(P)$ are

holomorphic with respect to $\Delta$. We introduce the regularized spherical functions $\varphi_{\Delta,\epsilon}(T)$, which are well-defined for integer $\Delta$-s,

$$\varphi_{\Delta,\epsilon}(T) = A^{-1}(\Delta,\epsilon)\phi_{\Delta,\epsilon}(T), \quad \text{where} \quad A(\Delta,\epsilon) = \Gamma(\frac{-\Delta+\epsilon+1}{2})\Gamma(\frac{-\widetilde{\Delta}+\epsilon+1}{2}). \quad \text{(D.41)}$$

**Completeness and Plancherel measure.** Similar to the EAdS case, in Appendix D.5.2 we show that the (regularized) spherical functions of principal, complementary and exceptional series representations provide a complete basis for the radial Laplacian (D.13), including the continuous part $\{\varphi_{\frac{d}{2}+is,\epsilon} : s > 0, \epsilon = 0, 1\}$ and the discrete part $\{\varphi_{\Delta,\epsilon} : (\Delta,\epsilon) \in D_{\mathrm{dS}}^-\}$, where

$$D_{\mathrm{dS}}^- = \{(\Delta \in \mathbb{Z}, \epsilon) : \Delta < \frac{d}{2} \text{ and } \epsilon = 1 + \Delta \bmod \mathbb{Z}_2\}. \quad \text{(D.42)}$$

By the Sturm-Liouville theory, the nonvanishing orthogonality relations are

$$2\pi N_{\mathrm{dS}}^c(\Delta_1,\epsilon_1)\delta(s_1-s_2)\delta_{\epsilon_1,\epsilon_2} = (\varphi_{\Delta_1,\epsilon_1}, \varphi_{\Delta_2,\epsilon_2}), \quad \text{(D.43)}$$

$$2\pi N_{\mathrm{dS}}^d(\Delta_1)\delta_{\Delta_1,\Delta_2} = (\varphi_{\Delta_1}, \varphi_{\Delta_2}), \quad \text{(D.44)}$$

where the inner product in the right side is

$$(f,g) = \int_{\mathbb{R}} dT |T^2-1|^{\frac{d-1}{2}} f^*(T)g(T). \quad \text{(D.45)}$$

And the completeness relation is

$$|T^2-1|^{\frac{d-1}{2}}\delta(T_1-T_2)$$
$$= \frac{1}{2\pi i}\sum_{\epsilon=0,1}\int_{\Gamma}\frac{d\Delta}{N_{\mathrm{dS}}^c(\Delta,\epsilon)}\varphi_{\Delta,\epsilon}^*(T_1)\varphi_{\Delta,\epsilon}(T_2) + \frac{1}{2\pi}\sum_{(\Delta,\epsilon)\in D_{\mathrm{dS}}^-}\frac{1}{N_{\mathrm{dS}}^d(\Delta)}\varphi_{\Delta}^*(T_1)\varphi_{\Delta}(T_2). \quad \text{(D.46)}$$

Similarly, the density $N_{\mathrm{dS}}^c(\Delta,\epsilon)$ is determined by the asymptotic behaviour of the spherical functions

$$N_{\mathrm{dS}}^c(\Delta,\epsilon) = 2c(\Delta,\epsilon)c(\widetilde{\Delta},\epsilon), \quad \text{(D.47)}$$

where $c(\Delta,\epsilon)$ is the $c$-function of dS spacetime (D.72), and the factor 2 is due to the two boundary $T \to \pm\infty$. We find and check numerically that the density $N_{\mathrm{dS}}^d(\Delta)$ is related to $N_{\mathrm{dS}}^c(\Delta,\epsilon)$ by

$$\frac{1}{N_{\mathrm{dS}}^d(\Delta)} = -\alpha_d \operatorname{Res}_{\Delta \in D_{\mathrm{dS}}^-}\frac{1}{N_{\mathrm{dS}}^c(\Delta,\epsilon)}, \quad \text{(D.48)}$$

where the constant $\alpha_d = 1$ for odd $d$ and $\alpha_d = \frac{1}{2}$ for even $d$.

**Fourier transform and inversion formula.** For later convenience, we turn back to the unregularized spherical functions $\varphi_{\Delta,\epsilon}(T)$ by recovering the regularization factor $A(\Delta,\epsilon)$. The Poisson transform and the Fourier transform are

$$\mathcal{F}_{\Delta,\epsilon}^{\dagger} : f(P) \mapsto \int_{\text{LC}_{d+1}} d'P \, |X \cdot P|^{-\Delta,\epsilon} f(P) \in L^2(\text{dS}_{d+1}), \tag{D.49}$$

$$\mathcal{F}_{\Delta,\epsilon} : f(X) \mapsto \int_{\text{dS}_{d+1}} dX \, |X \cdot P|^{-\widetilde{\Delta},\epsilon} f(X) \in \mathcal{E}_{\Delta}, \tag{D.50}$$

$$\mathcal{P}_{\Delta,\epsilon} : f(X) \mapsto \int_{\text{dS}_{d+1}} dX' \, \phi_{\Delta,\epsilon}(X,X') f(X'). \tag{D.51}$$

Similar to the spherical functions, the Fourier transform of $f(X)$ has poles at $D_{\text{dS}}^-$, and the residue is proportional to the Fourier transform with respect to $\varphi_{\Delta,\epsilon}(X)$. The inversion formula is

$$f(X) = \frac{1}{2\pi i} \sum_{\epsilon=0,1} \int_{\Gamma} \frac{d\Delta}{2^{d+1}\mu(\Delta)} \mathcal{P}_{\Delta,\epsilon}[f(X)] - \alpha_d \sum_{(\Delta,\epsilon)\in D_{\text{dS}}^-} \text{Res} \frac{1}{2^{d+1}\mu(\Delta)} \mathcal{P}_{\Delta,\epsilon}[f(X)], \tag{D.52}$$

where we have rewritten the density $N_{\text{dS}}^d(\Delta)$ by

$$\mu(\Delta) = 2^{-d} A^{-1}(\Delta,\epsilon) N_{\text{dS}}^d(\Delta) = \frac{\pi^d \Gamma(\Delta - \frac{d}{2})\Gamma(\widetilde{\Delta} - \frac{d}{2})}{\Gamma(\Delta)\Gamma(\widetilde{\Delta})}. \tag{D.53}$$

For the Dirac delta distribution we have the split representation on the dS spacetime

$$\begin{aligned}
\delta(X,X') = &\frac{1}{2\pi i} \sum_{\epsilon=0,1} \int_{\Gamma} \frac{d\Delta}{2\mu(\Delta)} \int_{\text{LC}_{d+1}} d'P \, |2X \cdot P|^{-\widetilde{\Delta},\epsilon} |2X' \cdot P|^{-\Delta,\epsilon} \\
&- \alpha_d \sum_{(\Delta,\epsilon)\in D_{\text{dS}}^-} \text{Res} \int_{\text{LC}_{d+1}} d'P \, \frac{1}{2\mu(\Delta)} |2X \cdot P|^{-\widetilde{\Delta},\epsilon} |2X' \cdot P|^{-\Delta,\epsilon},
\end{aligned} \tag{D.54}$$

By the shadow symmetry of the spherical functions, the discrete part of (D.54) flips sign and with $D_{\text{dS}}^-$ changing to $D_{\text{dS}}$,

$$D_{\text{dS}} = \{(\Delta \in \mathbb{Z}, \epsilon) : \Delta > \frac{d}{2} \text{ and } \epsilon = 1 + d + \Delta \bmod \mathbb{Z}_2\}. \tag{D.55}$$

## D.5 Sturm-Liouville problem and boundary conditions

In this appendix we analyse the Sturm-Liouville equation (D.12) with different boundary conditions,

$$\left(z^2 - 1\right) f''(z) + (d+1)z f'(z) - \Delta(\Delta - d) f(z) = 0. \tag{D.56}$$

This is a singular[15] Sturm-Liouville problem and can be solved by the following steps:

- find all solutions and determine the inner product;

- choose suitable boundary conditions;

- select the complete basis from the solutions.

The allowed boundary conditions are usually not unique. They lead to different set of complete basis and correspond to inequivalent self-adjoint extensions of the same differential operator. This phenomenon has already occurred in the discussion of CFT inversion formulas [43, 64–66].

The Sturm-Liouville standard form of (D.56) is

$$\mathcal{L}[f](x) = -|z^2 - 1|^{\frac{1-d}{2}} \frac{d}{dz} \left[ \mathrm{sgn}(z-1)|z^2 - 1|^{\frac{1+d}{2}} \frac{d}{dz} f(z) \right] = \Delta(\Delta - d)f(x), \qquad \text{(D.57)}$$

where we keep the weight function $w(z) = |z^2 - 1|^{\frac{1-d}{2}}$ positive and absorb the sign into $p(z) = \mathrm{sgn}(z-1)|z^2 - 1|^{\frac{1+d}{2}}$. Then the Sturm-Liouville inner product agrees with that in the global coordinates of $H_{d+1}$ and in the hyperbolic coordinates of $dS_{d+1}$,

$$(f, g) = \int dz |z^2 - 1|^{\frac{d-1}{2}} f^*(z)g(z), \qquad \text{(D.58)}$$

and the differential operator $\mathcal{L}$ is formally self-adjoint upto the boundary terms

$$(f, \mathcal{L}g) - (\mathcal{L}f, g) = \mathrm{sgn}(z-1)|z^2 - 1|^{\frac{1+d}{2}} \left[ \frac{d}{dz} f(z)g^*(z) - f(z)\frac{d}{dz} g^*(z) \right]_{z=-\infty}^{z=\infty}. \qquad \text{(D.59)}$$

The region $z \in (-\infty, -1)$ is related to $z \in (1, \infty)$ by $z \to -z$. For $z \in (1, \infty)$ using the substitution $f(z) = (z^2 - 1)^{\frac{1-d}{4}} g(z)$, $g(z)$ satisfies the standard Legendre equation, hence the two independent solutions of (D.56) are

$$\phi_1(z) = a_1 \left( z^2 - 1 \right)^{\frac{1-d}{4}} P_{\frac{1}{2}(d-2\Delta-1)}^{\frac{d-1}{2}}(z), \qquad \text{(D.60)}$$

$$\phi_2(z) = a_2 \left( z^2 - 1 \right)^{\frac{1-d}{4}} Q_{\frac{1}{2}(d-2\Delta-1)}^{\frac{d-1}{2}}(z), \qquad \text{(D.61)}$$

where the Legendre functions $P_b^a(z), Q_b^a(z)$ have a branch cut along $z \in (-\infty, 1)$, and in Mathematica they are `LegendreP[b,a,3,z]`, `LegendreQ[b,a,3,z]` respectively. For example, if choosing $a_2 = \frac{e^{-\frac{1}{2}i\pi(d-1)}}{\Gamma(d-\Delta)}$, then $f_2(z)$ is well-defined and real-valued for any $d, \Delta \in \mathbb{R}$ and $z \in (1, \infty)$.

---

[15]The term "singular" means the interval is infinite and the function $(z^2 - 1)^{-\frac{1+d}{2}}$ is singular at the boundary $z = \pm 1$. In this case the spectrum may contain both continuous and discrete parts, see e.g. [67].

For $z \in (-1, 1)$ we choose the following solutions

$$\phi_1(z) = a_1' \left(1 - z^2\right)^{\frac{1-d}{4}} \widetilde{P}_{\frac{1}{2}(d-2\Delta-1)}^{\frac{d-1}{2}}(z), \tag{D.62}$$

$$\phi_2(z) = a_2' \left(1 - z^2\right)^{\frac{1-d}{4}} \widetilde{Q}_{\frac{1}{2}(d-2\Delta-1)}^{\frac{d-1}{2}}(z), \tag{D.63}$$

where $\widetilde{P}, \widetilde{Q}$ have a branch cut along $z \in (-\infty, -1) \cup (1, \infty)$, and in Mathematica they are `LegendreP[b,a,2,z]`, `LegendreQ[b,a,2,z]` respectively.

### D.5.1  EAdS case

The radial direction of the EAdS space corresponds to the region $z \in (1, \infty)$. For convenience we work in the $t$-coordinate with $z = \cosh t$, $t > 0$. The Sturm–Liouville equation is

$$\sinh^{-d} t \frac{d}{dt} \left[\sinh^d t \frac{df}{dt}\right] = \Delta(\Delta - d) f. \tag{D.64}$$

Then the inner product is

$$(f, g) = \int_0^\infty dt \, \sinh^d t \, f^*(t) g(t), \tag{D.65}$$

and the boundary terms are

$$\sinh^d t \left[\frac{d}{dt} f_1(t) f_2^*(t) - f_1(t) \frac{d}{dt} f_2^*(t)\right]_{t=0}^{t=\infty}. \tag{D.66}$$

The spherical function is a linear combination of Legendre functions,

$$\phi_\Delta(t) = \frac{(2\pi)^{\frac{d+1}{2}} \Gamma(1 - \Delta)}{\Gamma(d - \Delta)} \left(\phi_1(\cosh t) + \frac{2}{(\tan \frac{\pi d}{2} - i)\pi} \phi_2(\cosh t)\right), \tag{D.67}$$

where we have set $a_1 = a_2 = 1$ in (D.60) and (D.61). By the theory of ODE, the other solution of (D.64) besides $\phi_\Delta(t)$ is singular at $t = 0$, since the index equation of (D.64) at $t = 0$ is $x(x - 1) + xd = 0$ and the difference of roots $x_1 - x_2 = d - 1$ is an integer.

To establish the completeness of the spherical functions, we require that the boundary conditions should eliminate the boundary terms (D.66) and exclude the non-spherical singular solutions. The correct choice is

$$f(0) < \infty, \ f'(0) = 0 \ \text{at} \ t = 0, \tag{D.68}$$

$$f(t) \sim O(e^{-\frac{d}{2}t}) \ \text{as} \ t \to \infty, \tag{D.69}$$

and this determines the Hilbert space $\mathcal{H}_{\mathcal{L}}$ of the Sturm-Liouville problem.

The reality condition of the eigenvalues $\Delta(\Delta - d)$ implies: 1) $\Delta = \frac{d}{2} + is$, $s \geqslant 0$; 2) $\Delta = \frac{d}{2} + s$, $s < 0$, where the redundancy from the shadow symmetry $\phi_\Delta(t) = \phi_{\widetilde{\Delta}}(t)$ has been reduced. The asymptotic behaviour of $\phi_\Delta(t)$ as $t \to \infty$ is

$$\phi_\Delta(t) \sim e^{-\widetilde{\Delta}t}c(\Delta) + e^{-\Delta t}c(\widetilde{\Delta}), \quad \text{where} \ \ c(\Delta) = \frac{2^d \pi^{\frac{d}{2}} \Gamma(\Delta - \frac{d}{2})}{\Gamma(\Delta)}. \tag{D.70}$$

For the first case, $\phi_\Delta(t) \sim e^{-\frac{d}{2}t}(e^{ist}c(\Delta) + e^{-ist}c(\widetilde{\Delta}))$ saturates the boundary condition and oscillates like plane-wave, hence the spherical functions $\phi_{\frac{d}{2}+is}(t)$, $s \geqslant 0$ of unitary principal series belong to the continuous spectrum of the Laplacian. For the second case, the term $\phi_\Delta(t) \sim e^{-\frac{d}{2}t}e^{-st}c(\widetilde{\Delta})$ dominates the growth and exceeds the boundary condition unless the coefficient vanishes $c(\widetilde{\Delta}) = 0$. The equation $c(\widetilde{\Delta}) = 0$ does not have solutions for $s < 0$, hence all the eigenfunctions $\phi_{\frac{d}{2}+s}(t)$, $s < 0$ are ruled out and there is no discrete spectrum.

Hence the spherical functions $\phi_{\frac{d}{2}+is}(t)$, $s \geqslant 0$ associated with the unitary principal series representations provide a orthogonal complete basis of the radial Laplacian (D.64).

### D.5.2 dS case

For the dS case, there are several differences from the EAdS case: 1) the region is $z \in \mathbb{R}$ and the joint points $z = \pm 1$ are singular; 2) the well-defined eigenfunctions are the regularized spherical functions $\varphi_{\Delta,\epsilon}(z)$ and they exhaust all the solutions of the Sturm-Liouville equation.

Since the dS spherical functions are even/odd functions of $z$, we can restrict to $z > 0$ and the boundary terms at $z = 0$ get canceled by the parity symmetry. As $z \to 1^{\pm}$ the asymptotic behaviours of $\varphi_{\Delta,\epsilon}(z)$ take the same form, the boundary terms at $z = 1$ cancel with each other due to the factor $\text{sgn}(z - 1)$. Similar to the EAdS case, to cancel the boundary term at $z = \infty$ we choose the following boundary condition

$$f(z) \sim O(z^{-\frac{d}{2}}) \ \text{ as } \ z \to \infty, \tag{D.71}$$

and this determines the Hilbert space $\mathcal{H}_{\mathcal{L}}$.

The reality condition of eigenvalues is the same as before: 1) $\Delta = \frac{d}{2} + is$, $s \geqslant 0$; 2) $\Delta = \frac{d}{2} + s$, $s < 0$. The asymptotic behaviours of the spherical functions as $z \to \infty$ are

$$\varphi_{\Delta,\epsilon}(z) \sim z^{-\widetilde{\Delta}}c(\Delta, \epsilon) + z^{-\Delta}c(\widetilde{\Delta}, \epsilon), \tag{D.72}$$

where the $c$-function is

$$c(\Delta, \epsilon) = \frac{2\pi^{\frac{d-1}{2}} \Gamma(\Delta - \frac{d}{2}) \cos \frac{\pi}{2}(d - \Delta + \epsilon)}{\Gamma(\frac{\Delta-d+\epsilon+1}{2})\Gamma(\frac{\Delta+\epsilon}{2})}. \tag{D.73}$$

For the first case, all the spherical functions associated with the principal series saturate the boundary condition, hence they are in the continuum spectrum of the Laplacian. For the second case, the dominate term is $z^{-\Delta}c(\widetilde{\Delta}, \epsilon)$. But unlike the EAdS case, for $s < 0$ the $c$-function $c(\widetilde{\Delta}, \epsilon) \sim \cos\frac{\pi}{2}(\Delta + \epsilon) = 0$ has solutions at the set

$$D_{\text{dS}}^- = \{(\Delta \in \mathbb{Z}, \epsilon) : \Delta < \frac{d}{2} \text{ and } \epsilon = 1 + \Delta \mod \mathbb{Z}_2\}. \tag{D.74}$$

Namely, $\Delta$ is an integer at the left of the principal series and the parity $\epsilon$ is determined by $\Delta$. For $(\Delta, \epsilon) \in D_{\text{dS}}^-$, the spherical functions $\varphi_{\Delta, \epsilon}(z)$ are square integrable due to the estimate,

$$\int_\Lambda^\infty (z^2 - 1)^{\frac{d-1}{2}} dz \, |\phi_{\Delta, \epsilon}(t)|^2 \sim \int_\Lambda^\infty dz \, z^{2s-1} < \infty, \text{ for } s < 0. \tag{D.75}$$

Hence the corresponding representations are in the discrete part of $L^2(\text{dS}_{d+1})$. By comparing the Casimir eigenvalues, they are the exceptional series representations for $\Delta > d$ and the complementary series representations for $\frac{d}{2} < \Delta \leqslant d$.

## D.6  EAdS/dS spherical functions

In this section we derive the $H$-spherical functions on the EAdS and dS spacetimes. We need the following conformal integrals: for timelike $X$, $-X^2 > 0$,

$$\int_{\text{LC}_{d+1}} d'P \, (-X \cdot P)^{-d} = \text{vol} \, S^d (-X^2)^{-\frac{d}{2}}, \tag{D.76}$$

and for spacelike $X$, $X^2 > 0$,

$$\int_{\text{LC}_{d+1}} d'P \, |X \cdot P|^{-d} = 2\pi^{\frac{d-1}{2}} \Gamma(\frac{1-d}{2})(X^2)^{-\frac{d}{2}} = \frac{1}{\cos\frac{d}{2}\pi} \text{vol} \, S^d (X^2)^{-\frac{d}{2}}. \tag{D.77}$$

They can be computed in the global coordinates, and the remaining integrals are

$$\text{vol} \, S^{d-1} \int_0^\pi d\theta \, \sin^{d-1}\theta = \text{vol} \, S^d, \tag{D.78}$$

$$\text{vol} \, S^{d-1} \int_0^\pi d\theta \, \sin^{d-1}\theta |\cos^{-d}\theta| = 2\pi^{\frac{d-1}{2}} \Gamma(\frac{1-d}{2}). \tag{D.79}$$

The prefactor $\Gamma(\frac{1-d}{2})$ in (D.77) is exactly the regularization factor of the distribution $|X \cdot P|^{-d}$. By the relations (B.4), the conformal integral (D.77) is equivalent to

$$\int_{\text{LC}_{d+1}} d'P \, (X \cdot P \pm i\epsilon)^{-d} = e^{\mp\frac{d}{2}\pi i} \text{vol} \, S^d (X^2)^{-\frac{d}{2}}. \tag{D.80}$$

The Feynman-Schwinger parameterization is

$$\prod_{i=1}^{n} A_i^{-a_i} = \frac{\Gamma(d)}{\prod_{i=1}^{n} \Gamma(a_i)} \int_0^\infty \prod_{i=2}^{n} ds_i s_i^{a_i-1} \left( A_1 + \sum_{i=2}^{n} s_i A_i \right)^{-d} \tag{D.81}$$

$$= \frac{\Gamma(d)}{\prod_{i=1}^{n} \Gamma(a_i)} \int_0^1 \prod_{i=1}^{n} du_i u_i^{a_i-1} \, \delta(1 - \sum_{i=1}^{n} u_i)(\sum_{i=1}^{n} u_i A_i)^{-d}, \tag{D.82}$$

where $\Re(a_i) > 0$ and $d = \sum_i a_i$.

The Gegenbauer integral is related to the hypergeometric function by

$$I_G(z) = C_{-\Delta}^{\frac{d}{2}}(z)\frac{\pi}{\sin \pi z} = \int_0^\infty ds s^{\Delta-1}\left(1 + s^2 + 2sz\right)^{-\frac{d}{2}} \tag{D.83}$$

$$= \frac{\Gamma(\Delta)\Gamma(d-\Delta)}{\Gamma(d)}{}_2F_1(\Delta, d-\Delta, \frac{d+1}{2}, \frac{1-z}{2}). \tag{D.84}$$

where $C_{-\Delta}^{\frac{d}{2}}(z)$ is the Gegenbauer function, see section 3.15.2 of [68].

### D.6.1   EAdS spherical functions

In the global coordinates $X = (\cosh t, \widehat{X} \sinh t)$, $P = (1, \widehat{P})$ and $\cos \theta = \widehat{X} \cdot \widehat{P}$, the spherical function (D.23) is

$$\phi_\Delta(t) = \int_{S^d} d\widehat{P}\,(-X \cdot P)^{-\widetilde{\Delta}} \tag{D.85}$$

$$= \text{vol}\, S^{d-1} \int_0^\pi d\theta\, \sin^{d-1}\theta(\cosh t - \sinh t \cos \theta)^{-\widetilde{\Delta}} \tag{D.86}$$

$$= \text{vol}\, S^{d-1} 2^{d-1} e^{-\Delta t} \int_0^1 dz\, z^{\frac{d}{2}-1}(1-z)^{\frac{d}{2}-1}(1 - (1 - e^{-2t})z)^{-\widetilde{\Delta}} \tag{D.87}$$

$$= \text{vol}\, S^d\, e^{-\widetilde{\Delta}t}\, {}_2F_1(\frac{d}{2}, \widetilde{\Delta}, d, 1 - e^{-2t}), \tag{D.88}$$

where in the third line the $\theta$-integral is substituted by $z = \frac{1}{2}(1 + \cos \theta)$. As a crosscheck, we re-derive the spherical function (D.23) by the Feynman-Schwinger parameterization: with $T = -X_1 \cdot X_2 \geqslant 1$ the two-point spherical function is

$$\phi_\Delta(X_1, X_2) = \int_{\text{LC}_{d+1}} d'P\,(-X_1 \cdot P)^{-\widetilde{\Delta}}(-X_2 \cdot P)^{-\Delta} \tag{D.89}$$

$$= \frac{\Gamma(d)}{\Gamma(\Delta)\Gamma(\widetilde{\Delta})} \int_0^\infty ds s^{\Delta-1} \int_{\text{LC}_{d+1}} d'P\,(-(X_1 + sX_2) \cdot P)^{-d} \tag{D.90}$$

$$= \text{vol}\, S^d \frac{\Gamma(d)}{\Gamma(\Delta)\Gamma(\widetilde{\Delta})} \int_0^\infty ds s^{\Delta-1}\left(1 + s^2 + 2sT\right)^{-\frac{d}{2}} \tag{D.91}$$

$$= \text{vol}\, S^d{}_2F_1(\Delta, \widetilde{\Delta}, \frac{d+1}{2}, \frac{1-T}{2}), \tag{D.92}$$

where in the third line, the $P$-integral is done by (D.76), and in the last line the $s$-integral is done by (D.83). The two results (D.88) and (D.92) are related by the quadratic transformations of hypergeometric functions.

We show that the spherical functions of complementary series are analytic continuation of $\Delta$ from that of principal series, although they do not enter into the complete basis. In this case $\Delta$ is real and the inner product follows from (D.19). The spherical function is

$$c_{\text{cp}} \int d'P_1 d'P_2 \, (-X_1 \cdot P_1)^{-\Delta}(-P_1 \cdot P_2)^{-\widetilde{\Delta}}(-X_2 \cdot P_2)^{-\Delta} \tag{D.93}$$

$$= c_{\text{cp}} \frac{\Gamma(d)}{\Gamma(\Delta)\Gamma(\widetilde{\Delta})} \int d'P_1 d'P_2 \int_0^\infty d\alpha\, \alpha^{\widetilde{\Delta}-1}(-(X_1 + \alpha P_2) \cdot P_1)^{-d}(-X_2 \cdot P_2)^{-\Delta} \tag{D.94}$$

$$= c_{\text{cp}} \frac{\text{vol}\, S^d \Gamma(d)}{\Gamma(\Delta)\Gamma(\widetilde{\Delta})} \int d'P_2 \int_0^\infty d\alpha\, \alpha^{\widetilde{\Delta}-1}(1 - 2\alpha X_1 \cdot P_2)^{-\frac{d}{2}}(-X_2 \cdot P_2)^{-\Delta} \tag{D.95}$$

$$= c_{\text{cp}} \frac{2^\Delta \pi^{\frac{d}{2}}\Gamma(\Delta - \frac{d}{2})}{\Gamma(\Delta)} \int_{\text{LC}_{d+1}} d'P_2 \, (-X_1 \cdot P_2)^{-\widetilde{\Delta}}(-X_2 \cdot P_2)^{-\Delta} \tag{D.96}$$

$$\sim \phi_{\Delta, \text{principal}}(X_1, X_2). \tag{D.97}$$

In the second line the Feynman-Schwinger parameterization is used to merge the factors, and in the third line the integral over $P_1$ is done by (D.76).

### D.6.2   dS spherical functions

The dS two-point spherical funcions (D.37) take different forms in the three hyperbolic charts $T = X_1 \cdot X_2 \in (-\infty, -1)$, $(-1, 1)$ and $(1, \infty)$,

$$\phi_{\Delta, \epsilon}(T) = \int_{\text{LC}_{d+1}} d'P\, |X_1 \cdot P|^{-\widetilde{\Delta}, \epsilon}|X_2 \cdot P|^{-\Delta, \epsilon}, \tag{D.98}$$

and the result of $T \in (-\infty, -1)$ is related to that of $T \in (1, \infty)$ by the parity symmetry

$$\phi_{\Delta, \epsilon}(T) = (-1)^\epsilon \phi_{\Delta, \epsilon}(-T). \tag{D.99}$$

As a crosscheck, we also evaluate the spherical functions by two different methods.

**First method.** We fix $X_2$ to the reference point $X_0 = (0, \ldots, 1)$ and parametrize $X_1$ and $P$ by the hyperbolic coordinates (D.4),

$$\phi_{\Delta, \epsilon}(T) = \int_{\text{LC}_{d+1}} d'P\, |X_1 \cdot P|^{-\widetilde{\Delta}, \epsilon}|P^{d+1}|^{-\Delta, \epsilon} \tag{D.100}$$

$$= \int_{\mathrm{H}_d} d\check{P}\, |X_1 \cdot (\check{P}, 1)|^{-\tilde{\Delta},\epsilon} + \int_{\mathrm{H}_d} d\check{P}\, |X_1 \cdot (\check{P}, -1)|^{-\tilde{\Delta},\epsilon}(-1)^\epsilon. \tag{D.101}$$

In the third chart of the hyperbolic coordinates, with $X_1 = (\check{X}\sinh t, \cosh t)$, $T = \cosh t$, $t \geqslant 0$, $P = (\check{P}, \pm 1)$ and $\cosh s = -\check{X}\cdot\check{P}$, $s \geqslant 0$, we have

$$\phi_{\Delta,\epsilon}(t) = \phi_{\Delta,\epsilon}^+(t) + \phi_{\Delta,\epsilon}^-(t), \tag{D.102}$$

where $\pm$ labels the contributions from the two terms in (D.101). For the minus term, the factor $\cosh t + \sinh t \cosh s > 0$ is positive, and

$$\phi_{\Delta,\epsilon}^-(t) = \mathrm{vol}\,\mathrm{S}^{d-1}(-1)^\epsilon \int_0^\infty ds\,\sinh^{d-1} s\, |-\cosh t - \sinh t \cosh s|^{\Delta-d,\epsilon} \tag{D.103}$$

$$= \mathrm{vol}\,\mathrm{S}^{d-1} \int_0^\infty ds\,\sinh^{d-1} s\,(\cosh t + \sinh t \cosh s)^{\Delta-d} \tag{D.104}$$

$$= 2^{d-1}e^{(d-\Delta)t}\int_0^1 dz\, z^{\frac{d}{2}-1}(1-z)^{-\Delta}(e^{2t}-z)^{\Delta-d} \tag{D.105}$$

$$= \frac{2^d \pi^{d/2}\Gamma(1-\Delta)}{\Gamma(\frac{d}{2}-\Delta+1)}e^{(\Delta-d)t}{}_2F_1(\frac{d}{2},d-\Delta,\frac{d}{2}-\Delta+1,e^{-2t}), \tag{D.106}$$

where in the third line $s$ is substituted by $z = \tanh^2 \frac{s}{2}$. The plus term can be separated into two parts at $\cosh s = \coth t$ according to the sign of $X_1 \cdot P$, denoted as $I^{p/n}$,

$$\phi_{\Delta,\epsilon}^+(t) = \mathrm{vol}\,\mathrm{S}^{d-1}\int_0^\infty ds\,\sinh^{d-1} s\, |\cosh t - \sinh t \cosh s|^{\Delta-d,\epsilon} = I^p(t) + (-1)^\epsilon I^n(t). \tag{D.107}$$

For the positive part $I^p(t)$ with $\cosh s \in (1, \coth t)$, we use the substitution $z = \frac{\cosh s-1}{\coth t-1}$,

$$I^p(t) = \mathrm{vol}\,\mathrm{S}^{d-1}e^{(d-\Delta)t}(\coth t-1)^{d-1}\int_0^1 dz\,(e^{2t}+z-1)^{\frac{d-2}{2}}z^{\frac{d-2}{2}}(1-z)^{\Delta-d} \tag{D.108}$$

$$= \frac{2^d\pi^{d/2}\Gamma(-d+\Delta+1)}{\Gamma(-\frac{d}{2}+\Delta+1)}e^{(d-\Delta)t}(e^{2t}-1)^{-d/2}\,{}_2F_1(1-\frac{d}{2},\frac{d}{2},-\frac{d}{2}+\Delta+1,\frac{1}{1-e^{2t}}). $$

And for the negative part $I^n(t)$ with $\cosh s \in (\coth t, \infty)$, we use the substitution $z = \frac{\coth t-1}{\cosh s-1}$,

$$I^n(t) = \mathrm{vol}\,\mathrm{S}^{d-1}e^{(d-\Delta)t}(\coth t-1)^{d-1}\int_0^1 dz\,\left((e^{2t}-1)z+1\right)^{\frac{d-2}{2}}z^{-\Delta}(z-1)^{\Delta-d} \tag{D.109}$$

$$= \frac{2^d\pi^{d/2}\Gamma(1-\Delta)\Gamma(-d+\Delta+1)}{\Gamma(\frac{d}{2})\Gamma(2-d)}e^{(d-\Delta)t}(e^{2t}-1)^{1-d}\,{}_2F_1(1-\frac{d}{2},1-\Delta,2-d,1-e^{2t}). $$

Then the total contributions to the spherical functions are

$$\phi_{\Delta,\epsilon}(T) = 2^{\frac{d+1}{2}}\pi^{\frac{d-1}{2}}\Gamma(1-\Delta)\left[\Gamma(-d+1+\Delta)\left((-1)^\epsilon \sin\frac{d\pi}{2} + \sin\frac{(d-2\Delta)\pi}{2}\right)\phi_1(T) + 2\phi_2(T)\right], \tag{D.110}$$

where we have set $a_1 = 1, a_2 = \frac{e^{-\frac{1}{2}i\pi(d-1)}}{\Gamma(d-\Delta)}$ in (D.60) and (D.61).

For the second chart $T \in (-1, 1)$, with $X_1 = (\widetilde{X}\sin\theta, \cos\theta)$, $P = (\check{P}, \pm 1)$ and $T = \cos\theta$, the integral also contains two terms from (D.101),

$$\phi_{\Delta,\epsilon}(\theta) = \phi_{\Delta,\epsilon}^+(\theta) + \phi_{\Delta,\epsilon}^-(\theta), \tag{D.111}$$

and each term is of the form $\phi_{\Delta,\epsilon}^\pm(\theta) = \int_{H_d} d\check{P} f^\pm(\widetilde{X} \cdot \check{P}, \theta)$. For the plus term $\phi_{\Delta,\epsilon}^+(\theta)$, using the symmetry $SO(d,1)$ of $H_d$ and $dS_d$, we can fix $\widetilde{X} = (0, \ldots, 1)$ and parametrize $\check{P}$ as $\check{P} = (\cosh s, \ldots, \sinh s \cos\varphi)$, then

$$\phi_{\Delta,\epsilon}^+(\theta) = \text{vol}\, S^{d-2} \int_0^\pi d\varphi \int_0^\infty ds \, \sin^{d-2}\varphi \sinh^{d-1} s |\cos\theta + \sin\theta \sinh s \cos\varphi|^{\Delta-d,\epsilon} \tag{D.112}$$

$$= \text{vol}\, S^{d-2} \int_0^\infty dy \int_{\mathbb{R}} dx \, \frac{y^{d-2}}{\sqrt{1+x^2+y^2}} |\cos\theta + x\sin\theta|^{\Delta-d,\epsilon} \tag{D.113}$$

$$= \pi^{\frac{d}{2}-1}\Gamma(1 - \frac{d}{2}) \int_{\mathbb{R}} dx \, (1+x^2)^{\frac{d}{2}-1} |\cos\theta + x\sin\theta|^{\Delta-d,\epsilon} \tag{D.114}$$

$$= J^p(\theta) + (-1)^\epsilon J^n(\theta), \tag{D.115}$$

where we have substituted $x = \sinh s \cos\varphi, y = \sinh s \sin\varphi$ and separated the integral into two parts according to the sign of $\cos\theta + x\sin\theta$. The minus term $\phi_{\Delta,\epsilon}^-(\theta)$ is equal to the plus term by

$$\phi_{\Delta,\epsilon}^-(\theta) = (-1)^\epsilon \left( J^p(\pi-\theta) + (-1)^\epsilon J^n(\pi-\theta) \right) = (-1)^\epsilon \left( J^n(\theta) + (-1)^\epsilon J^p(\theta) \right) = \phi_{\Delta,\epsilon}^+(\theta). \tag{D.116}$$

For the postive part $J^p(\theta)$ with $x \in (-\cot\theta, \infty)$, by the substitution $z = \cos\theta + x\sin\theta \in (0, \infty)$, we have

$$J^p(\theta) = \pi^{\frac{d}{2}-1}\Gamma(1 - \frac{d}{2})\sin^{1-d}\theta \int_0^\infty dz \, z^{\Delta-d}(1+z^2 - 2z\cos\theta)^{\frac{d}{2}-1} \tag{D.117}$$

$$= \frac{2^{d-1}\pi^{\frac{d-1}{2}}\Gamma(1-\Delta)\Gamma(-d+\Delta+1)}{\Gamma(\frac{3}{2} - \frac{d}{2})}\sin^{1-d}\theta \,_2F_1(1-\Delta, 1-d+\Delta, \frac{3-d}{2}, \frac{1+\cos\theta}{2}), \tag{D.118}$$

and for the minus part $J^n(\theta)$ with $x \in (-\infty, -\cot\theta)$, by the substitution $z = -\cos\theta - x\sin\theta \in (0, \infty)$, we have

$$J^n(\theta) = \pi^{\frac{d}{2}-1}\Gamma(1 - \frac{d}{2})\sin^{1-d}\theta \int_0^\infty dz \, z^{\Delta-d}(1+z^2 + 2z\cos\theta)^{\frac{d}{2}-1} \tag{D.119}$$

$$= \frac{2^{d-1}\pi^{\frac{d-1}{2}}\Gamma(1-\Delta)\Gamma(-d+\Delta+1)}{\Gamma(\frac{3}{2} - \frac{d}{2})}\sin^{1-d}\theta \,_2F_1(1-\Delta, 1-d+\Delta, \frac{3-d}{2}, \frac{1-\cos\theta}{2}). \tag{D.120}$$

Then the spherical functions are

$$\phi_{\Delta,\epsilon}(T) = 2^{\frac{d+1}{2}}\pi^{\frac{d-1}{2}}\Gamma(1-\Delta)\Gamma(-d+\Delta+1)\left[\phi_1(T)\left((-1)^\epsilon - \cos\pi(d-\Delta)\right) + \phi_2(T)\frac{2\sin\pi(d-\Delta)}{\pi}\right],$$

(D.121)

where we have set $a_1' = 1, a_2' = 1$ in (D.62) and (D.63).

**Second method.** We evaluate the two-point spherical functions by the Feynman-Schwinger parameterization. By the relations (B.6), the integrand $|X_1 \cdot P|^{-\widetilde{\Delta},\epsilon}|X_2 \cdot P|^{-\Delta,\epsilon}$ can be rewritten as linear combinations of

$$h_{\pm,\pm} = (X_1 \cdot P \pm i\epsilon)^{-\widetilde{\Delta}}(X_2 \cdot P \pm i\epsilon)^{-\Delta}.$$

(D.122)

Explicitly, they are

$$\phi_{\Delta,0}(T) \ni \frac{e^{2i\pi\Delta}h_{-,+}}{(1+e^{i\pi\Delta})(e^{i\pi d}+e^{i\pi\Delta})} + \frac{h_{+,-}}{(1+e^{i\pi\Delta})(1+e^{i\pi(\Delta-d)})}$$

(D.123)

$$+ \frac{e^{i\pi\Delta}h_{-,-}}{(1+e^{i\pi\Delta})(e^{i\pi d}+e^{i\pi\Delta})} + \frac{h_{+,+}}{(1+e^{-i\pi\Delta})(1+e^{i\pi(\Delta-d)})},$$

(D.124)

$$\phi_{\Delta,1}(T) \ni \frac{e^{2i\pi\Delta}h_{-,+}}{(-1+e^{i\pi\Delta})(e^{i\pi\Delta}-e^{i\pi d})} + \frac{h_{+,-}}{(-1+e^{i\pi\Delta})(-1+e^{i\pi(\Delta-d)})}$$

(D.125)

$$+ \frac{e^{i\pi\Delta}h_{-,-}}{(-1+e^{i\pi\Delta})(e^{i\pi d}-e^{i\pi\Delta})} + \frac{h_{+,+}}{(-1+e^{-i\pi\Delta})(-1+e^{i\pi(\Delta-d)})}.$$

(D.126)

Then similar to (D.92) we combine the two factor $(X_1 \cdot P \pm i\epsilon)^{-\widetilde{\Delta}}$ and $(X_2 \cdot P \pm i\epsilon)^{-\Delta}$ by the Feynman-Schwinger parameterization and do the $P$-integral by the conformal integral (D.80).

For $T \in (-1, 1)$, $s \geqslant 0$, the factor $1 + s^2 \pm 2sT > 0$ is positive, and the $s$-integrand can be done by (D.83). The four terms contribute to

$$h_{+,+} \to e^{-\frac{d}{2}\pi i}\operatorname{vol}S^d I_{G,0}(T),$$

(D.127)

$$h_{-,-} \to e^{d\pi i}e^{-\frac{d}{2}\pi i}\operatorname{vol}S^d I_{G,0}(T),$$

(D.128)

$$h_{+,-} \to e^{\Delta\pi i}e^{-\frac{d}{2}\pi i}\operatorname{vol}S^d I_{G,0}(-T),$$

(D.129)

$$h_{-,+} \to e^{(d-\Delta)\pi i}e^{-\frac{d}{2}\pi i}\operatorname{vol}S^d I_{G,0}(-T),$$

(D.130)

where we have relabeled $I_G$ to $I_{G,0}$,

$$I_{G,0}(T) = \frac{\Gamma(\Delta)\Gamma(\widetilde{\Delta})}{\Gamma(d)}\,_2F_1(\Delta, d - \Delta, \frac{d+1}{2}, \frac{1-T}{2}).$$

(D.131)

Then the spherical functions are

$$\phi_{\Delta,0}(T) = 2\pi^{\frac{d}{2}-1}\Gamma(\frac{1}{2} - \frac{\Delta}{2})\Gamma(\frac{-d+\Delta+1}{2})\,_2F_1(\frac{d-\Delta}{2}, \frac{\Delta}{2}, \frac{1}{2}, T^2),$$

(D.132)

$$\phi_{\Delta,1}(T) = 4\pi^{\frac{d}{2}-1} T\Gamma(1-\frac{\Delta}{2})\Gamma(\frac{-d+\Delta+2}{2})_2F_1(\frac{d-\Delta+1}{2}, \frac{\Delta+1}{2}, \frac{3}{2}, T^2), \qquad (D.133)$$

and they match with (D.121).

For $T \in (1,\infty), s \geqslant 0$, the factor $1 + s^2 \pm 2sT$ is not definite and the integrals should be divided into three parts according to the signs. The four terms contribute to

$$h_{+,+} \to e^{-\frac{d}{2}\pi i} \operatorname{vol} S^d I_{G,0}(T), \qquad (D.134)$$

$$h_{-,-} \to e^{d\pi i} e^{-\frac{d}{2}\pi i} \operatorname{vol} S^d I_{G,0}(T), \qquad (D.135)$$

$$h_{+,-} \to e^{\Delta\pi i} \operatorname{vol} S^d \left( e^{-\frac{d}{2}\pi i} I_{G,1}(T) + e^{-\frac{d}{2}\pi i} I_{G,3}(T) + I_{G,2}(T) \right), \qquad (D.136)$$

$$h_{-,+} \to e^{(d-\Delta)\pi i} \operatorname{vol} S^d \left( e^{-\frac{d}{2}\pi i} I_{G,1}(T) + e^{-\frac{d}{2}\pi i} I_{G,3}(T) + I_{G,2}(T) \right), \qquad (D.137)$$

where $T = \cosh t$ and

$$I_{G,1}(T) = \frac{\Gamma(1-\frac{d}{2})\Gamma(\Delta)}{\Gamma(-\frac{d}{2}+\Delta+1)} e^{-\Delta t} {}_2F_1(\frac{d}{2}, \Delta, -\frac{d}{2}+\Delta+1, e^{-2t}), \qquad (D.138)$$

$$I_{G,2}(T) = \frac{\Gamma(1-\frac{d}{2})^2}{\Gamma(2-d)} (e^{2t}-1)^{1-d} e^{dt-\Delta t} {}_2F_1(1-\frac{d}{2}, 1-\Delta, 2-d, 1-e^{2t}), \qquad (D.139)$$

$$I_{G,3}(T) = \frac{\Gamma(1-\frac{d}{2})\Gamma(d-\Delta)}{\Gamma(\frac{d-2\Delta+2}{2})} e^{t(d-\Delta)} {}_2F_1(\frac{d}{2}, d-\Delta, \frac{d-2\Delta+2}{2}, e^{2t}). \qquad (D.140)$$

Then by the relations between hypergeometric functions and Legendre functions, the spherical functions match with (D.110).

# E  Massless scalar-massless scalar-tachyonic scalar

In this appendix, we compute the three-point celestial amplitude $\mathcal{A}^{\Delta_i}_{1^0_0 \to 2^0_0 + 3^0_{iM,\epsilon}}$ which involves one incoming massless scalar, one outgoing massless scalar and one outgoing tachyonic scalar with imaginary mass $iM$. In terms of the conformal primary basis, $\mathcal{A}^{\Delta_i}_{1^0_0 \to 2^0_0 + 3^0_{iM,\epsilon}}$ takes the form as

$$\mathcal{A}^{\Delta_i}_{1^0_0 \to 2^0_0 + 3^0_{iM,\epsilon}} = \int_0^\infty d\omega_1 d\omega_2 \omega_1^{\Delta_1-1} \omega_2^{\Delta_2-1} \int \frac{d^{d+1}\widehat{k}}{|\widehat{k}^+|} \frac{1}{|\widehat{q}_3 \cdot \widehat{k}|^{\Delta_3}} \operatorname{sgn}^\epsilon(\widehat{q}_3 \cdot \widehat{k}) \delta^{(d+2)}(q_1 - q_2 - k) , \qquad (E.1)$$

where $k^2 = M^2$. The region with $k^2 = M^2$ can be divided into two regions corresponding to expanding and contracting patches of the dS hypersurfaces

$$\begin{aligned} \mathcal{D}^+ : \qquad & k^2 = M^2 , \qquad \widehat{k}^+ \equiv \widehat{k}^0 + \widehat{k}^{d+1} > 0 , \\ \mathcal{D}^- : \qquad & k^2 = M^2 , \qquad \widehat{k}^+ \equiv \widehat{k}^0 + \widehat{k}^{d+1} < 0 , \end{aligned} \qquad (E.2)$$

respectively. We can thus define the celestial coordinates in $\mathcal{D}^+$ as

$$k = M\widehat{k} = \frac{M}{2y}\left(1 - y^2 + |\vec{w}|^2, 2\vec{w}, 1 + y^2 - |\vec{w}|^2\right), \tag{E.3}$$

and the celestial coordinates in $\mathcal{D}^-$ as

$$k = M\widehat{k} = -\frac{M}{2y}\left(1 - y^2 + |\vec{w}|^2, 2\vec{w}, 1 + y^2 - |\vec{w}|^2\right), \tag{E.4}$$

with $y \in [0, \infty)$ in both regions. As a result, $\mathcal{A}^{\Delta_i}_{1^0_0 \to 2^0_0 + 3^0_{iM,\epsilon}}$ can be written as

$$\mathcal{A}^{\Delta_i}_{1^0_0 \to 2^0_0 + 3^0_{iM,\epsilon}} = \mathcal{A}^{+,\Delta_i}_{1^0_0 \to 2^0_0 + 3^0_{iM,\epsilon}} + \mathcal{A}^{-,\Delta_i}_{1^0_0 \to 2^0_0 + 3^0_{iM,\epsilon}}, \tag{E.5}$$

where $\mathcal{A}^{\pm,\Delta_i}_{1^0_0 \to 2^0_0 + 3^0_{iM,\epsilon}}$ are obtained by integral $\widehat{k}$ over region $\mathcal{D}^\pm$. We first focus on $\mathcal{A}^{+,\Delta_i}_{1^0_0 \to 2^0_0 + 3^0_{iM,\epsilon}}$. Using the fact that

$$\int_{\mathcal{D}^+} \frac{d^{d+1}\widehat{k}}{|k^+|} \frac{1}{|\widehat{q}_3 \cdot \widehat{k}|^{\Delta_3}} = \int_0^\infty \frac{dy}{y^{d+1}} \int d^d\vec{w} \left|\frac{y}{|x_3 - w|^2 - y^2}\right|^{\Delta_3}, \tag{E.6}$$

we find that

$$\mathcal{A}^{+,\Delta_i}_{1^0_0 \to 2^0_0 + 3^0_{iM,\epsilon}} = \int_0^\infty d\omega_1 d\omega_2 \omega_1^{\Delta_1 - 1} \omega_2^{\Delta_2 - 1} \int_0^\infty \frac{dy}{y^{d+1}} \int d^d\vec{w} \left|\frac{y}{|x_3 - w|^2 - y^2}\right|^{\Delta_3} \delta^{(d+2)}(q_1 - q_2 - k). \tag{E.7}$$

The support of the momentum-conserving delta-function is

$$\omega_2 = \frac{M^2}{4|x_{12}|^2\omega_1}, \qquad y = \frac{2M|x_{12}|^2\omega_1}{4|x_{12}|^2\omega_1^2 - M^2}, \qquad \vec{w} = \frac{M^2\vec{x}_2 - 4\vec{x}_1|x_{12}|^2\omega_1^2}{M^2 - 4|x_{12}|^2\omega_1^2} \tag{E.8}$$

with the Jacobian

$$|J| = \left|\frac{M^{d+1}(y^2 - |x_2 - w|^2)}{y^{d+2}}\right|. \tag{E.9}$$

We note that $\omega_1 \geqslant M/(2|x_{12}|)$ since $y \geqslant 0$. The integrand in (E.7) then becomes

$$\begin{aligned}
&\frac{\omega_1^{\Delta_1 - 1}\omega_2^{\Delta_2 - 1}}{y^{d+1}} \left|\frac{y}{|x_3 - w|^2 - y^2}\right|^{\Delta_3} \delta^{(d+2)}(q_1 - q_2 - k) \\
&= \frac{2^{1-2\Delta_2 + \Delta_3} M^{2\Delta_2 + \Delta_3 - d - 2}\omega_1^{\Delta_1 - \Delta_2 + \Delta_3 - 1}}{|x_{12}|^{2\Delta_2 - 2\Delta_3}} \left|\frac{1}{-M^2|x_{23}|^2 + 4|x_{12}|^2|x_{13}|^2\omega_1^2}\right|^{\Delta_3} \\
&\quad \times \delta\left(y - \frac{2M|x_{12}|^2\omega_1}{4|x_{12}|^2\omega_1^2 - M^2}\right) \delta\left(\omega_2 - \frac{M^2}{4|x_{12}|^2\omega_1}\right) \delta^{(d)}\left(\vec{w} - \frac{M^2\vec{x}_2 - 4\vec{x}_1|x_{12}|^2\omega_1^2}{M^2 - 4|x_{12}|^2\omega_1^2}\right).
\end{aligned} \tag{E.10}$$

Performing the integral over $y$, $\omega_2$, and $\vec{w}$ leads to

$$\mathcal{A}^{+,\Delta_i}_{1^0_0\to2^0_0+3^0_{iM,\epsilon}}=\frac{2^{1-2\Delta_2+\Delta_3}M^{2\Delta_2+\Delta_3-d-2}}{|x_{12}|^{2\Delta_2-2\Delta_3}}\int_{\frac{M}{2|x_{12}|}}^{\infty}d\omega_1\frac{\omega_1^{\Delta_1-\Delta_2+\Delta_3-1}\mathrm{sgn}^{\epsilon}(-M^2|x_{23}|^2+4|x_{12}|^2|x_{13})}{\big|-M^2|x_{23}|^2+4|x_{12}|^2|x_{13}|^2\omega_1^2\big|^{\Delta_3}}\,.$$
(E.11)

Now, we turn to the computation of $\mathcal{A}^{-,\Delta_i}_{1^0_0\to2^0_0+3^0_{iM,\epsilon}}$. In the region $\mathcal{D}^-$, we have

$$\int_{\mathcal{D}^-}\frac{d^{d+1}\widehat{k}}{|k^+|}\frac{1}{|\widehat{q}_3\cdot\widehat{k}|^{\Delta_3}}=\int_0^{\infty}\frac{dy}{y^{d+1}}\int d^d\vec{w}\left|\frac{-y}{|x_3-w|^2-y^2}\right|^{\Delta_3},$$
(E.12)

leading to

$$\mathcal{A}^{-,\Delta_i}_{1^0_0\to2^0_0+3^0_{iM,\epsilon}}=\int_0^{\infty}d\omega_1d\omega_2\omega_1^{\Delta_1-1}\omega_2^{\Delta_2-1}\int_0^{\infty}\frac{dy}{y^{d+1}}\int d^d\vec{w}\left|\frac{-y}{|x_3-w|^2-y^2}\right|^{\Delta_3}\delta^{(d)}(q_1-q_2-k)\,.$$
(E.13)

The support of the momentum-conserving delta-function is

$$\omega_2=\frac{M^2}{4|x_{12}|^2\omega_1}\,,\qquad y=\frac{2M|x_{12}|^2\omega_1}{M^2-4|x_{12}|^2\omega_1^2}\,,\qquad\vec{w}=\frac{M^2\vec{x}_2-4\vec{x}_1|x_{12}|^2\omega_1^2}{M^2-4|x_{12}|^2\omega_1^2}$$
(E.14)

with the Jacobian

$$|J|=\left|\frac{M^{d+1}(y^2-|x_2-w|^2)}{y^{d+2}}\right|\,.$$
(E.15)

We note that in the region $\mathcal{D}^-$, $y\geqslant0$ demands that $0\leqslant\omega_1\leqslant M/(2|x_{12}|)$. The integrand in (E.13) then becomes

$$\begin{aligned}&\frac{\omega_1^{\Delta_1-1}\omega_2^{\Delta_2-1}}{y^{d+1}}\left|\frac{-y}{|x_3-w|^2-y^2}\right|^{\Delta_3}\delta^{(d+2)}(q_1-q_2-k)\\[2mm]&=\frac{2^{1-2\Delta_2+\Delta_3}M^{2\Delta_2+\Delta_3-d-2}\omega_1^{\Delta_1-\Delta_2+\Delta_3-1}}{|x_{12}|^{2\Delta_2-2\Delta_3}}\left|\frac{1}{-M^2|x_{23}|^2+4|x_{12}|^2|x_{13}|^2\omega_1^2}\right|^{\Delta_3}\\[2mm]&\times\delta\left(y-\frac{2M|x_{12}|^2\omega_1}{M^2-4|x_{12}|^2\omega_1^2}\right)\delta\left(\omega_2-\frac{M^2}{4|x_{12}|^2\omega_1}\right)\delta^{(d)}\left(\vec{w}-\frac{M^2\vec{x}_2-4\vec{x}_1|x_{12}|^2\omega_1^2}{M^2-4|x_{12}|^2\omega_1^2}\right)\,.\end{aligned}$$
(E.16)

Performing the integral over $y$, $\omega_2$, and $\vec{w}$ leads to

$$\mathcal{A}^{-,\Delta_i}_{1^0_0\to2^0_0+3^0_{iM,\epsilon}}=\frac{2^{1-2\Delta_2+\Delta_3}M^{2\Delta_2+\Delta_3-d-2}}{|x_{12}|^{2\Delta_2-2\Delta_3}}\int_0^{\frac{M}{2|x_{12}|}}d\omega_1\frac{\omega_1^{\Delta_1-\Delta_2+\Delta_3-1}\mathrm{sgn}^{\epsilon}(-M^2|x_{23}|^2+4|x_{12}|^2|x_{13}|^2)}{\big|-M^2|x_{23}|^2+4|x_{12}|^2|x_{13}|^2\omega_1^2\big|^{\Delta_3}}\,.$$
(E.17)

Thus we have

$$
\begin{aligned}
\mathcal{A}^{\Delta_i}_{1_0^0 \to 2_0^0 + 3_{iM,\epsilon}^0} &= \frac{2^{1-2\Delta_2+\Delta_3} M^{2\Delta_2+\Delta_3-d-2}}{|x_{12}|^{2\Delta_2-2\Delta_3}} \int_0^\infty d\omega_1 \frac{\omega_1^{\Delta_1-\Delta_2+\Delta_3-1}\mathrm{sgn}^\epsilon(-M^2|x_{23}|^2 + 4|x_{12}|^2|x_{13})}{\big| - M^2|x_{23}|^2 + 4|x_{12}|^2|x_{13}|^2\omega_1^2\big|^{\Delta_3}} \\
&= \frac{2^{1-2\Delta_2+\Delta_3} M^{2\Delta_2+\Delta_3-d-2}}{|x_{12}|^{2\Delta_2-2\Delta_3}} \left( \int_0^{\frac{M|x_{23}|}{2|x_{12}||x_{13}|}} d\omega_1 \frac{\omega_1^{\Delta_1-\Delta_2+\Delta_3-1}(-1)^\epsilon}{(M^2|x_{23}|^2 - 4|x_{12}|^2|x_{13}|^2\omega_1^2)^{\Delta_3}} \right. \\
&\left. \quad + \int_{\frac{M|x_{23}|}{2|x_{12}||x_{13}|}}^\infty d\omega_1 \frac{\omega_1^{\Delta_1-\Delta_2+\Delta_3-1}}{(-M^2|x_{23}|^2 + 4|x_{12}|^2|x_{13}|^2\omega_1^2)^{\Delta_3}} \right) \\
&= \frac{C^{\Delta_i}_{1_0^0 \to 2_0^0 + 3_{iM,\epsilon}^0}}{|x_{12}|^{\Delta_1+\Delta_2-\Delta_3}|x_{13}|^{\Delta_1-\Delta_2+\Delta_3}|x_{23}|^{\Delta_2+\Delta_3-\Delta_1}} \, ,
\end{aligned}
$$

$$(E.18)$$

where $C^{\Delta_i}_{1_0^0 \to 2_0^0 + 3_{iM,\epsilon}^0}$ is given by

$$
C^{\Delta_i}_{1_0^0 \to 2_0^0 + 3_{iM,\epsilon}^0} = \frac{M^{\Delta_1+\Delta_2-d-2}\Gamma[1-\Delta_3]}{2^{\Delta_1+\Delta_2}} \left( \frac{\Gamma[\frac{\Delta_3-\Delta_1+\Delta_2}{2}]}{\Gamma[\frac{2-\Delta_1+\Delta_2-\Delta_3}{2}]} + (-1)^\epsilon \frac{\Gamma[\frac{\Delta_3+\Delta_1-\Delta_2}{2}]}{\Gamma[\frac{2+\Delta_1-\Delta_2-\Delta_3}{2}]} \right) . \quad (E.19)
$$

Using the identity

$$
\Gamma[x]\Gamma[1-x] = \frac{\pi}{\sin[\pi x]} \, , \tag{E.20}
$$

we get

$$
\begin{aligned}
C^{\Delta_i}_{1_0^0 \to 2_0^0 + 3_{iM,\epsilon}^0} &= \frac{M^{\Delta_1+\Delta_2-d-2}\Gamma[1-\Delta_3]\Gamma[\frac{\Delta_3-\Delta_1+\Delta_2}{2}]\Gamma[\frac{\Delta_3+\Delta_1-\Delta_2}{2}]}{2^{\Delta_1+\Delta_2}} \\
&\times \left( 2\cos[\frac{\Delta_3}{2}\pi]\sin[\frac{(\Delta_1-\Delta_2)}{2}\pi] \right)^\epsilon \left( 2\sin[\frac{\Delta_3}{2}\pi]\cos[\frac{(\Delta_1-\Delta_2)}{2}\pi] \right)^{1-\epsilon} .
\end{aligned}
$$

$$(E.21)$$

Moreover, using the symmetric property $\delta^{(4)}(x) = \delta^{(4)}(-x)$, we find that $\mathcal{A}^{\Delta_i}_{2_0^0+3_{iM,\epsilon}^0 \to 1_0^0} = \mathcal{A}^{\Delta_i}_{1_0^0 \to 2_0^0+3_{iM,\epsilon}^0}$ .

# F $\quad$ $t$-channel conformal block expansion from alpha space approach

In this section we derive the $t$-channel conformal block expansion of the $1d$ four-point massless scalar $t$-channel celestial amplitudes by the alpha space approach. The four-point massless scalar $t$-channel celestial amplitudes in one dimensional CCFT is

$$
_t\mathcal{A}^{\Delta_i}_{1_0^0+2_0^0 \to 3_0^0+4_0^0} = \frac{\pi m^{\beta-1}}{2^{\beta+4}} \sec[\frac{\pi\beta}{2}]I_{13-24}f(\chi) \, , \tag{F.1}
$$

where $f(\chi) = \chi(1-\chi)^{\frac{1}{2}(-1+\Delta_{13}-\Delta_{24})}$. In the following, we will first consider the conformal block expansion of $\mathcal{G}(\chi) = \chi^a(1-\chi)^b$ and then take the special value $a = 1, b = \frac{1}{2}(-1 + \Delta_{13} - \Delta_{24})$.

Due to the different boundary condition of the conformal Casimir equation, in the alpha space approach the integral region is $x \in (0,1)$ and there are no contributions from the discrete series representations, in contrast to the Euclidean inversion formula [64, 65], see also the application in celestial CFT [7].

In the alpha space approach, the stripped conformal blocks and partial waves are

$$G_\Delta^{\Delta_{13},\Delta_{24}}(\chi) = \chi^\Delta {}_2F_1(\Delta - \Delta_{13}, \Delta + \Delta_{24}, 2\Delta, \chi), \tag{F.2}$$

$$\psi_\Delta(\chi) = \frac{1}{2}(Q(\Delta)G_\Delta^{\Delta_{13},\Delta_{24}}(\chi) + Q(1-\Delta)G_{1-\Delta}^{\Delta_{13},\Delta_{24}}(\chi)) \tag{F.3}$$

$$= \chi^{\Delta_{13}} {}_2F_1(\Delta - \Delta_{13}, 1 - \Delta - \Delta_{13}, 1 - \Delta_{13} + \Delta_{24}, 1 - \frac{1}{\chi}), \tag{F.4}$$

with the boundary condition $\psi_\Delta(1) = 1$. The inversion function and formula are

$$I(\Delta) = \int_0^1 d\chi\, \chi^{-2}(1-\chi)^{-\Delta_{13}+\Delta_{24}}\mathcal{G}(\chi)\psi_\Delta^*(\chi), \tag{F.5}$$

$$\mathcal{G}(\chi) = \frac{1}{2\pi i}\int_\Gamma \frac{d\Delta}{N(\Delta)}\,I(\Delta)\psi_\Delta(\chi), \tag{F.6}$$

where $\Gamma$ is the principal series, and

$$Q(\Delta) = \frac{2\Gamma(1-2\Delta)\Gamma(1-\Delta_{13}+\Delta_{24})}{\Gamma(1-\Delta-\Delta_{13})\Gamma(1-\Delta+\Delta_{24})}, \quad \text{and} \quad N(\Delta) = \frac{1}{2}Q(\Delta)Q(1-\Delta). \tag{F.7}$$

For the four-point function $\mathcal{G}(\chi) = \chi^a(1-\chi)^b$ using the Mellin-Barnes integral and changing the order of integration, we have

$$I(\Delta) = \int_0^1 d\chi\, \chi^{a-2}(1-\chi)^{b-\Delta_{13}+\Delta_{24}}\psi_\Delta(\chi) \tag{F.8}$$

$$= \int_{-i\infty}^{+i\infty} \frac{ds}{2\pi i} \frac{(\Delta - \Delta_{13})_s(1-\Delta-\Delta_{13})_s}{(1-\Delta_{13}+\Delta_{24})_s}\Gamma(-s) \cdot \int_0^1 dx\, x^{a+\Delta_{13}-s-2}(1-x)^{b+s-\Delta_{13}+\Delta_{24}}$$

$$= \int_{-i\infty}^{+i\infty} \frac{ds}{2\pi i} \frac{(\Delta - \Delta_{13})_s(1-\Delta-\Delta_{13})_s}{(1-\Delta_{13}+\Delta_{24})_s} \frac{\Gamma(b+s-\Delta_{13}+\Delta_{24}+1)}{\Gamma(a+b+\Delta_{24})}\Gamma(-s)\Gamma(a+\Delta_{13}-s-1).$$

There are two series of poles in the right half-plane, $s = n$ and $s = a + \Delta_{13} + n - 1$ for $n \in \mathbb{N}$, and their contributions are

$$I(\Delta) = -a_1\, {}_3F_2\left(\begin{matrix} -\Delta - \Delta_{13} + 1, \Delta - \Delta_{13}, b - \Delta_{13} + \Delta_{24} + 1 \\ 2 - a - \Delta_{13}, -\Delta_{13} + \Delta_{24} + 1 \end{matrix}; 1\right) \tag{F.9}$$

$$- a_2 \, {}_3F_2 \left( \begin{matrix} a - \Delta, a + \Delta - 1, a + b + \Delta_{24} \\ a + \Delta_{13}, a + \Delta_{24} \end{matrix} ; 1 \right) \tag{F.10}$$

where the minus sign is due to the orientation of contour, and the coefficients are

$$a_1 = \frac{\Gamma \left( a + \Delta_{13} - 1 \right) \Gamma \left( b - \Delta_{13} + \Delta_{24} + 1 \right)}{\Gamma \left( a + b + \Delta_{24} \right)}, \tag{F.11}$$

$$a_2 = \frac{\Gamma \left( -\Delta_{13} + \Delta_{24} + 1 \right) \Gamma (a - \Delta) \Gamma (a + \Delta - 1) \Gamma \left( -a - \Delta_{13} + 1 \right)}{\Gamma \left( -\Delta - \Delta_{13} + 1 \right) \Gamma \left( \Delta - \Delta_{13} \right) \Gamma \left( a + \Delta_{24} \right)}. \tag{F.12}$$

As a first check, for four identical external operators $\Delta_{13} = \Delta_{24} = 0$, this result matches with (2.58) in [64] with the conventions $(\Delta, a, b) = (\frac{1}{2} + \alpha, p, -q)$.

Now we can derive the conformal block expansion by deforming the $\Delta$-contour to the right infinity. The first term is analytic in $\Delta$, and the second term contains simple poles $\Delta_n = a + n$, $n \in \mathbb{N}$ at the right side of principal series. Hence the four-point function can be expanded as

$$\mathcal{G}(\chi) = \chi^a (1 - \chi)^b = \sum_n \mathcal{C}_{1+n}^{\Delta_i} G_{1+n}^{\Delta_{13}, \Delta_{24}}(\chi) = -2 \sum_n \frac{\mathrm{Res}_{\Delta = \Delta_n} I(\Delta)}{Q(1 - \Delta_n)} G_{1+n}^{\Delta_{13}, \Delta_{24}}(\chi), \tag{F.13}$$

where the orientation of contour contributes to factor $-1$, and the shadow term contributes to factor 2. The block coefficients are

$$\mathcal{C}_{1+n}^{\Delta_i} = \frac{(a + \Delta_{13})_n \, (a + \Delta_{24})_n}{n! (2a + n - 1)_n} \, {}_3F_2 \left( \begin{matrix} -n, 2a + n - 1, a + b + \Delta_{24} \\ a + \Delta_{13}, a + \Delta_{24} \end{matrix} ; 1 \right) \tag{F.14}$$

and we can match the block expansion with the four-point function order by order.

Back to the original four-point function (F.1), taking $a = 1, b = \frac{1}{2}(-1 + \Delta_{13} - \Delta_{24})$ we have

$$\mathcal{C}_{1+n}^{\Delta_i} = \frac{\pi m^{\beta - 1}}{2^{\beta + 4}} \sec[\frac{\pi \beta}{2}] \frac{(-1)^n 2^{2n} \Gamma \left( \frac{n - \Delta_{13} + 1}{2} \right) \Gamma \left( \frac{n + \Delta_{24} + 1}{2} \right)}{\Gamma(2n + 1) \Gamma \left( \frac{-n - \Delta_{13} + 1}{2} \right) \Gamma \left( \frac{-n + \Delta_{24} + 1}{2} \right)}, \tag{F.15}$$

which agrees with (5.38) computed by using the split representation.

# G    Expansion of conformal block around its poles

In this appendix, we will derive the expansion of $1D$ and $2D$ conformal block around its poles. We start with the $1D$ conformal block $G_\Delta^{\Delta_{13}, \Delta_{24}}$ which takes the following form

$$G_\Delta^{\Delta_{13}, \Delta_{24}}(\chi) = \chi^\Delta \sum_{k=0}^\infty \frac{(\Delta - \Delta_{13})_k (\Delta + \Delta_{24})_k}{\Gamma[k + 1](2\Delta)_k} \chi^k = \Gamma[2\Delta] H_\Delta^{\Delta_{13}, \Delta_{24}}(\chi), \tag{G.1}$$

where we defined $H_\Delta^{\Delta_{13},\Delta_{24}}$ as

$$H_\Delta^{\Delta_{13},\Delta_{24}}(\chi) = \chi^\Delta \sum_{k=0}^\infty \frac{(\Delta - \Delta_{13})_k(\Delta + \Delta_{24})_k}{\Gamma[k+1]\Gamma[2\Delta + k]} \chi^k . \tag{G.2}$$

We note that $G_\Delta^{\Delta_{13},\Delta_{24}}$ has simple poles at $2\Delta = -n$ with $n = 0, 1, 2, \cdots$. This admits the following expansion of $_tF_\Delta^{\Delta_i}$ around $2\Delta = -n$

$$G_\Delta^{\Delta_{13},\Delta_{24}}(\chi) = \frac{a_{-1}^{\Delta_i,n}(\chi)}{2\Delta + n} + a_0^{\Delta_i,n}(\chi) + O^1(2\Delta + n) . \tag{G.3}$$

The coefficient $a_{-1}^{\Delta_i,n}(\chi)$ can be computed from

$$a_{-1}^{\Delta_i,n}(\chi) = \left( (2\Delta + n)G_\Delta^{\Delta_{13},\Delta_{24}}(\chi) \right)\Big|_{\Delta=-\frac{n}{2}} = \frac{(-1)^n}{n!} G_{-\frac{n}{2}}^{\Delta_{13},\Delta_{24}}(\chi) . \tag{G.4}$$

Using the definition (G.2) of $H_\Delta^{\Delta_{13},\Delta_{24}}(\chi)$, we find that

$$H_{-\frac{n}{2}}^{\Delta_{13},\Delta_{24}}(\chi) = \sum_{k=0}^\infty \chi^{-\frac{n}{2}} \frac{(-\frac{n}{2} - \Delta_{13})_k(-\frac{n}{2} + \Delta_{24})_k}{\Gamma[k+1]\Gamma[k-n]} \chi^k . \tag{G.5}$$

Shifting $k$ by $k \to k + n + 1$ leads to

$$\begin{aligned} H_{-\frac{n}{2}}^{\Delta_{13},\Delta_{24}}(\chi) &= \sum_{k=0}^\infty \chi^{\frac{n+2}{2}} \frac{(-\frac{n}{2} - \Delta_{13})_{k+n+1}(-\frac{n}{2} + \Delta_{24})_{k+n+1}}{\Gamma[k+n+2]\Gamma[k+1]} \chi^k \\ &= (-\frac{n}{2} - \Delta_{13})_{n+1}(-\frac{n}{2} + \Delta_{24})_{n+1} H_{\frac{n+2}{2}}^{\Delta_{13},\Delta_{24}}(\chi) . \end{aligned} \tag{G.6}$$

This leads to

$$a_{-1}^{\Delta_i,n}(\chi) = \frac{(-1)^n(-\frac{n}{2} - \Delta_{13})_{n+1}(-\frac{n}{2} + \Delta_{24})_{n+1}}{\Gamma[n+1]\Gamma[n+2]} G_{\frac{n+2}{2}}^{\Delta_{13},\Delta_{24}}(\chi) . \tag{G.7}$$

The coefficient $a_0^{\Delta_i,n}(\chi)$ can be computed from

$$\begin{aligned} a_0^{\Delta_i,n}(\chi) &= \frac{1}{2}\frac{\partial}{\partial\Delta}\left( (2\Delta + n)G_\Delta^{\Delta_{13},\Delta_{24}}(\chi) \right)\Big|_{\Delta=-\frac{n}{2}} \\ &= \frac{(-1)^n}{n!}\psi(n+1)H_{-\frac{n}{2}}^{\Delta_{13},\Delta_{24}}(\chi) + \frac{(-1)^n}{2n!}\frac{\partial H_\Delta^{\Delta_{13},\Delta_{24}}(\chi)}{\partial\Delta}\Big|_{\Delta=-\frac{n}{2}} \\ &= \frac{(-1)^n(-\frac{n}{2} - \Delta_{13})_{n+1}(-\frac{n}{2} + \Delta_{24})_{n+1}}{\Gamma[n+1]\Gamma[n+2]}\psi(n+1)G_{\frac{n+2}{2}}^{\Delta_{13},\Delta_{24}} + \frac{(-1)^n}{2n!}\frac{\partial H_\Delta^{\Delta_{13},\Delta_{24}}}{\partial\Delta}\Big|_{\Delta=-\frac{n}{2}} , \end{aligned} \tag{G.8}$$

where we used (G.6). To compute $\frac{1}{2} \frac{\partial H_\Delta^{\Delta_{13},\Delta_{24}}(\chi)}{\partial \Delta}\Big|_{\Delta=-\frac{n}{2}}$, we note that

$$\frac{1}{2} \frac{\partial H_\Delta^{\Delta_{13},\Delta_{24}}(\chi)}{\partial \Delta}\Big|_{\Delta=-\frac{n}{2}} = {}_1 S_{13-24}^{\Delta_i,-\frac{n}{2}}(\chi) + {}_2 S_{13-24}^{\Delta_i,-\frac{n}{2}}(\chi) + {}_3 S_{13-24}^{\Delta_i,-\frac{n}{2}}(\chi) + {}_4 S_{13-24}^{\Delta_i,-\frac{n}{2}}(\chi) , \qquad \text{(G.9)}$$

where ${}_i S_{13-24}^{\Delta_i,-\frac{n}{2}}(z)$ are defined as

$$ {}_1 S_{13-24}^{\Delta_i,-\frac{n}{2}}(\chi) \equiv \frac{1}{2} \frac{\partial \chi^\Delta}{\partial \Delta}\Big|_{\Delta=-\frac{n}{2}} \chi^{\frac{n}{2}} H_{-\frac{n}{2}}^{\Delta_{13},\Delta_{24}}(\chi) , \qquad \text{(G.10)}$$

and

$$\begin{aligned}
{}_2 S_{13-24}^{\Delta_i,-\frac{n}{2}}(\chi) &\equiv \frac{1}{2} \sum_{k=0}^\infty \chi^{\frac{-n}{2}} \frac{(-\frac{n}{2}+\Delta_{24})_k}{\Gamma[k+1]\Gamma[k-n]} \frac{\partial(\Delta-\Delta_{13})_k}{\partial\Delta}\Big|_{\Delta=-\frac{n}{2}} \chi^k \\
&= \sum_{k=0}^\infty \chi^{\frac{-n}{2}} \frac{(-\frac{n}{2}-\Delta_{13})_k(-\frac{n}{2}+\Delta_{24})_k}{\Gamma[k+1]\Gamma[k-n]} \frac{\psi(-\frac{n}{2}-\Delta_{13}+k)-\psi(-\frac{n}{2}-\Delta_{13})}{2} \chi^k ,
\end{aligned}$$

$$\text{(G.11)}$$

and

$$\begin{aligned}
{}_3 S_{13-24}^{\Delta_i,-\frac{n}{2}}(\chi) &\equiv \frac{1}{2} \sum_{k=0}^\infty \chi^{\frac{-n}{2}} \frac{(-\frac{n}{2}-\Delta_{13})_k}{\Gamma[k+1]\Gamma[k-n]} \frac{\partial(\Delta+\Delta_{24})_k}{\partial\Delta}\Big|_{\Delta=-\frac{n}{2}} \chi^k \\
&= \sum_{k=0}^\infty \chi^{\frac{-n}{2}} \frac{(-\frac{n}{2}-\Delta_{13})_k(-\frac{n}{2}+\Delta_{24})_k}{\Gamma[k+1]\Gamma[k-n]} \frac{\psi(-\frac{n}{2}+\Delta_{24}+k)-\psi(-\frac{n}{2}+\Delta_{24})}{2} \chi^k ,
\end{aligned}$$

$$\text{(G.12)}$$

and

$$\begin{aligned}
{}_4 S_{13-24}^{\Delta_i,-\frac{n}{2}}(\chi) &\equiv \frac{1}{2} \sum_{k=0}^\infty \chi^{\frac{-n}{2}} \frac{(-\frac{n}{2}-\Delta_{13})_k(-\frac{n}{2}+\Delta_{24})_k}{\Gamma[k+1]} \left( \frac{\partial}{\partial\Delta} \frac{1}{\Gamma[k+2\Delta]} \right)\Big|_{\Delta=-\frac{n}{2}} \chi^k \\
&= - \sum_{k=0}^\infty \chi^{\frac{-n}{2}} \frac{-(\frac{n}{2}-\Delta_{13})_k(-\frac{n}{2}+\Delta_{24})_k}{\Gamma[k+1]\Gamma[k-n]} \psi(k-n)\chi^k .
\end{aligned}$$

$$\text{(G.13)}$$

Using (G.6), we can re-write ${}_1 S_{13-24}^{\Delta_i,-\frac{n}{2}}(\chi)$ as

$$\begin{aligned}
{}_1 S_{13-24}^{\Delta_i,-\frac{n}{2}}(\chi) &= \frac{1}{2} \log(\chi) \, {}_t H_{-\frac{n}{2}}^{\Delta_i}(\chi) \\
&= \frac{1}{2} \log(\chi)(-\frac{n}{2}-\Delta_{13})_{n+1}(-\frac{n}{2}+\Delta_{24})_{n+1} H_{\frac{n+2}{2}}^{\Delta_{13},\Delta_{24}}(\chi) \qquad \text{(G.14)} \\
&= (-\frac{n}{2}-\Delta_{13})_{n+1}(-\frac{n}{2}+\Delta_{24})_{n+1} \, {}_1 S_{13-24}^{\Delta_i,\frac{n+2}{2}}(\chi) .
\end{aligned}$$

Moreover, after shifting the dummy index $k$ by $k \to k+n+1$, we can re-write $_2S_{13-24}^{\Delta_i,-\frac{n}{2}}(z)$ ast

$$_2S_{13-24}^{\Delta_i,-\frac{n}{2}} = \sum_{k=0}^{\infty} \chi^{\frac{n+2}{2}} \frac{(-\frac{n}{2}-\Delta_{13})_{k+n+1}(-\frac{n}{2}+\Delta_{24})_{k+n+1}}{\Gamma[k+n+2]\Gamma[k+1]} \frac{\psi(\frac{n+2}{2}-\Delta_{13}+k)-\psi(-\frac{n}{2}-\Delta_{13})}{2}\chi^k$$

$$=(-\frac{n}{2}-\Delta_{13})_{n+1}(-\frac{n}{2}+\Delta_{24})_{n+1}\left[_2S_{13-24}^{\Delta_i,\frac{n+2}{2}} + \frac{\psi(\frac{2+n}{2}-\Delta_{13})-\psi(-\frac{n}{2}-\Delta_{13})}{2\Gamma[n+2]} G_{\frac{n+2}{2}}^{\Delta_{13},\Delta_{24}}\right] .$$
$$\text{(G.15)}$$

Similarly, we get

$$_3S_{13,24}^{\Delta_i,-\frac{n}{2}} =(-\frac{n}{2}-\Delta_{13})_{n+1}(-\frac{n}{2}+\Delta_{24})_{n+1}\left[_3S_{13,24}^{\Delta_i,\frac{n+2}{2}} + \frac{\psi(\frac{2+n}{2}+\Delta_{24})-\psi(-\frac{n}{2}+\Delta_{24})}{2\Gamma[n+2]} G_{\frac{n+2}{2}}^{\Delta_{13},\Delta_{24}}\right] .$$
$$\text{(G.16)}$$

To compute $_4S_{13-24}^{\Delta_i,-\frac{n}{2}}$, we split the sum over $k \geqslant 0$ into two sums, $i.e.$,

$$_4S_{13-24}^{\Delta_i,-\frac{n}{2}} = -\left(\sum_{k=0}^{n}+\sum_{k=n+1}^{\infty}\right)\chi^{\frac{-n}{2}}\frac{(-\frac{n}{2}-\Delta_{13})_k(-\frac{n}{2}+\Delta_{24})_k}{\Gamma[k+1]\Gamma[k-n]}\psi(k-n)\chi^k . \qquad \text{(G.17)}$$

The sum with $k \geqslant n+1$ can be computed by shifting $k$ to $k+n+1$, giving

$$-\sum_{k=0}^{\infty}\chi^{\frac{n+2}{2}}\frac{(-\frac{n}{2}-\Delta_{13})_{k+n+1}(-\frac{n}{2}+\Delta_{24})_{k+n+1}}{\Gamma[k+n+2]\Gamma[k+1]}\psi(k+1)\chi^k . \qquad \text{(G.18)}$$

Armed with the identity

$$\psi(k+n+2) = \psi(k+1) + \sum_{s=0}^{n}\frac{1}{k+1+s} , \qquad \text{(G.19)}$$

It can be written as

$$-\sum_{k=0}^{\infty}\chi^{\frac{n+2}{2}}\frac{(-\frac{n}{2}-\Delta_{13})_{k+n+1}(-\frac{n}{2}+\Delta_{24})_{k+n+1}}{\Gamma[k+n+2]\Gamma[k+1]}\psi(k+1)\chi^k$$

$$= (-\frac{n}{2}-\Delta_{13})_{n+1}(-\frac{n}{2}+\Delta_{24})_{n+1}\,_4S_{13-24}^{\Delta_i,\frac{n+2}{2}} + \sum_{s=0}^{n}\sum_{k=0}^{\infty}\chi^{\frac{n+2}{2}}\frac{(-\frac{n}{2}-\Delta_{13})_{k+n+1}(-\frac{n}{2}+\Delta_{24})_{k+n+1}}{\Gamma[k+n+2]\Gamma[k+1](k+1+s)}\chi^k .$$
$$\text{(G.20)}$$

Using the fact that $\psi(-n)/\Gamma[-n] = (-1)^{n+1}n!$, we find that

$$-\sum_{k=0}^{n}\frac{(-\frac{n}{2}-\Delta_{13})_k(-\frac{n}{2}+\Delta_{24})_k\psi(k-n)}{\Gamma[k+1]\Gamma[k-n]}\chi^{k-\frac{n}{2}} + \sum_{s=0}^{n}\sum_{k=0}^{\infty}\frac{(-\frac{n}{2}-\Delta_{13})_{k+n+1}(-\frac{n}{2}+\Delta_{24})_{k+n+1}}{\Gamma[k+n+2]\Gamma[k+1](k+1+s)}\chi^{k+\frac{n+2}{2}}$$
$$\text{(G.21)}$$

is equal to $(-1)^n n! G_{\text{sta},-\frac{n}{2}}^{\Delta_{13},\Delta_{24}}(\chi)$ where $G_{\text{sta},-\frac{n}{2}}^{\Delta_{13},\Delta_{24}}(\chi)$ is the conformal block associated with the staggered module (A.3). Thus we find that

$$_4 S_{13-24}^{\Delta_i,-\frac{n}{2}} = (-\frac{n}{2} - \Delta_{13})_{n+1}(-\frac{n}{2} + \Delta_{24})_{n+1}\, _4 S_{13-24}^{\Delta_i,\frac{n+2}{2}} + (-1)^n n! G_{\text{sta},-\frac{n}{2}}^{\Delta_{13},\Delta_{24}} \,. \tag{G.22}$$

Combining (G.14), (G.15), (G.16), and (G.22), we get

$$\frac{1}{2}\frac{\partial H_\Delta^{\Delta_{13},\Delta_{24}}(\chi)}{\partial \Delta}\bigg|_{\Delta=-\frac{n}{2}}$$
$$= \frac{(-\frac{n}{2} - \Delta_{13})_{n+1}(-\frac{n}{2} + \Delta_{24})_{n+1}}{2\Gamma[n+2]}\left( b_n^{\Delta_i} G_{\frac{n+2}{2}}^{\Delta_{13},\Delta_{24}} + \frac{\partial G_\Delta^{\Delta_{13},\Delta_{24}}}{\partial \Delta}\bigg|_{\Delta=\frac{n+2}{2}}\right) + (-1)^n n! G_{\text{sta},-\frac{n}{2}}^{\Delta_{13},\Delta_{24}} \,, \tag{G.23}$$

where we defined $b_n^{\Delta_i}$ as

$$b_n^{\Delta_i} \equiv \psi(\frac{2+n}{2} - \Delta_{13}) - \psi(\frac{-n}{2} - \Delta_{13}) + \psi(\frac{2+n}{2} + \Delta_{24}) - \psi(\frac{-n}{2} + \Delta_{24}) - 2\psi(n+2) \,. \tag{G.24}$$

We conclude that

$$G_\Delta^{\Delta_{13},\Delta_{24}}(\chi) = \frac{a_{-1}^{\Delta_i,n}(\chi)}{2\Delta + n} + a_0^{\Delta_i,n}(\chi) + O^1(2\Delta + n) \,, \tag{G.25}$$

where $a_{-1}^{\Delta_i,n}(\chi)$ is given by

$$a_{-1}^{\Delta_i,n}(\chi) = \frac{(-1)^n(-\frac{n}{2} - \Delta_{13})_{n+1}(-\frac{n}{2} + \Delta_{24})_{n+1}}{\Gamma[n+1]\Gamma[n+2]}\, _t F_{\frac{n+2}{2}}^{\Delta_i}(\chi) \,, \tag{G.26}$$

and $a_0^{\Delta_i,n}(\chi)$ is given by

$$a_0^{\Delta_i,n} = \frac{(-1)^n(-\frac{n}{2} - \Delta_{13})_{n+1}(-\frac{n}{2} + \Delta_{24})_{n+1}}{2\Gamma[n+1]\Gamma[n+2]}\left( B_n^{\Delta_i}\, _t F_{\frac{n+2}{2}}^{\Delta_i} + \frac{\partial_t F_\Delta^{\Delta_i}}{\partial \Delta}\bigg|_{\Delta=\frac{n+2}{2}}\right) + (-1)^n n!\, _t F_{\text{sta},-\frac{n}{2}}^{\Delta_i} \tag{G.27}$$

with

$$B_n^{\Delta_i} \equiv \psi(\frac{2+n}{2} - \Delta_{13}) - \psi(\frac{-n}{2} - \Delta_{13}) + \psi(\frac{2+n}{2} + \Delta_{24}) - \psi(\frac{-n}{2} + \Delta_{24}) - \frac{2}{n+1} \,. \tag{G.28}$$

Since the two dimensional global conformal block is a product of two one-dimensional conformal blocks, *i.e.*,

$$G_{h,\bar{h}}^{h_{13},h_{24};\bar{h}_{13},\bar{h}_{24}}(\chi,\overline{\chi}) = G_h^{h_{13},h_{24}}(\chi)G_{h,\bar{h}}^{\bar{h}_{13},\bar{h}_{24}}(\overline{\chi}) \,, \tag{G.29}$$

we find that $G_{h,\overline{h}}^{h_{13},h_{24};\overline{h}_{13},\overline{h}_{24}}(\chi,\overline{\chi})$ admits the following expansion around the pole $2h = -n$ and $2\overline{h} = -\overline{n}$ or $\Delta = (-n - \overline{n})/2$ and $\ell = (-n + \overline{n})/2$

$$G_{h,\overline{h}}^{h_{13},h_{24};\overline{h}_{13},\overline{h}_{24}}(\chi,\overline{\chi}) = \frac{a_{-1}^{h_i,n}(\chi)a_{-1}^{\overline{h}_i,\overline{n}}(\overline{\chi})}{(2h+n)(2\overline{h}+\overline{n})} + \frac{a_{-1}^{h_i,n}(\chi)a_0^{\overline{h}_i,\overline{n}}(\overline{\chi})}{2h+n} + \frac{a_0^{h_i,n}(\chi)a_{-1}^{\overline{h}_i,\overline{n}}(\overline{\chi})}{2\overline{h}+\overline{n}} + \cdots . \quad \text{(G.30)}$$

Especially, when $h_i = \overline{h}_i = \Delta_i/2$ and $h = \overline{h} = \Delta/2$, we have

$$G_{h,\overline{h}}^{\Delta_{13},\Delta_{24}}(\chi,\overline{\chi})$$
$$= \frac{a_{-1}^{h_i,n}(\chi)a_{-1}^{h_i,n}(\overline{\chi})}{(2h+n)^2} + \frac{a_{-1}^{h_i,n}(\chi)a_0^{h_i,n}(\overline{\chi}) + a_0^{h_i,n}(\chi)a_{-1}^{h_i,n}(\overline{\chi})}{2h+n} + \cdots$$
$$= \left[\frac{(-\frac{n}{2} - h_{13})_{n+1}(-\frac{n}{2} + h_{24})_{n+1}}{\Gamma[n+1]\Gamma[n+2]}\right]^2 \left[\frac{G_{\frac{n+2}{2},\frac{n+2}{2}}^{\Delta_{13},\Delta_{24}}}{(\Delta+n)^2} + \frac{1}{\Delta+n}\left(B_n^{h_i}G_{\frac{n+2}{2},\frac{n+2}{2}}^{\Delta_{13},\Delta_{24}}\right.\right.$$
$$\left.\left. + \frac{\partial G_{h,\overline{h}}^{\Delta_{13},\Delta_{24}}}{\partial \Delta}\bigg|_{\Delta=n+2}\right)\right] + \frac{(-\frac{n}{2} - h_{13})_{n+1}(-\frac{n}{2} + h_{24})_{n+1}}{\Gamma[n+2](\Delta+n)}\left(G_{\text{sta},-\frac{n}{2},\frac{n}{2}+1}^{\Delta_{13},\Delta_{24}} + G_{\text{sta},\frac{n}{2}+1,-\frac{n}{2}}^{\Delta_{13},\Delta_{24}}\right) + \cdots ,$$
$$\text{(G.31)}$$

where $G_{\text{sta},-\frac{n}{2},\frac{n}{2}+1}^{\Delta_{13},\Delta_{24}}$ and $G_{\text{sta},\frac{n}{2}+1,-\frac{n}{2}}^{\Delta_{13},\Delta_{24}}$ are conformal blocks associated with chiral staggered module given in (A.6).

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
