# Peer review of "Split representation in celestial holography"

_SciPost Physics_

## Round 1 · Referee Report · Anonymous (Referee 1) · 2024-4-2

Strengths

interesting result, clean presentation, detailed appendices

Weaknesses

n/a

Report

The authors are developing a more systematic way to compute conformal partial waves for celestial amplitudes. I think this is an interesting result and I'm happy to endorse its publication. I also appreciate the efficiency in presenting the result of the paper paired with the extended pedagogy in the appendices.

Requested changes

The statements about helicity vs spin at the beginning of section 5.2 are a little confusing. I'd recommend using helicity for the 4D quantities and spin for the 2D ones if everything is massless. I don't like the sentence that the shadow flips the helicity -- I think you just mean 2D spin, hence my comment above.

---

## Round 1 · Referee Report · Prahar Mitra (Referee 2) · 2024-5-3

Strengths

  1. The split representation of the AdS and dS propagators to obtain a new representation of celestial amplitudes in terms of lower point amplitudes (analogous to the conformal block expansion in a CFT)

  2. The result may help us learn more about the locality and unitarity properties of CCFTs and is, therefore, interesting.

  3. The presentation of the paper is clear and several examples are explored (with lengthy calculations).

Weaknesses

The split representation seems to apply to individual Feynman diagrams, not to the bulk amplitude as a whole. It is not clear what this new representation would teach us about the complete amplitude. For example, the tree-level four gluon amplitude in Yang-Mills contains 4 Feynman diagrams, contributing over 1000 terms (the five gluon amplitude has around 10000 terms), but the total amplitude is incredibly simple.

How can the split representation of the Feynman propagator be useful in this case?

Report

The paper is well-written and accompanied by a rather detailed Appendix that will be useful in future developments in celestial holography. It can potentially give new insight into the structure of celestial amplitudes.

I recommend that it be published.

Requested changes

  1. Factors of $i\epsilon$ should be included in equations (4.2), (4.3), (4.4), (5.1) etc. to ensure that the relevant integrals are well-defined.

  2. There is an extra "[spin?]" in the first paragraph of section 5.2. Is that a typo?

Recommendation

Publish (easily meets expectations and criteria for this Journal; among top 50%)

---

## Editorial Decision

awaiting_resubmission